# Logical quantum processor based on reconfigurable atom arrays

Dolev Bluvstein[1], Simon J. Evered[1], Alexandra A. Geim[1], Sophie H. Li[1], Hengyun Zhou[1,2], Tom Manovitz[1], Sepehr Ebadi[1], Madelyn Cain[1], Marcin Kalinowski[1], Dominik Hangleiter[3], J. Pablo Bonilla Ataides[1], Nishad Maskara[1], Iris Cong[1], Xun Gao[1], Pedro Sales Rodriguez[2], Thomas Karolyshyn[2], Giulia Semeghini[4], Michael J. Gullans[3], Markus Greiner[1], Vladan Vuletić[5] & Mikhail D. Lukin[1✉]

Suppressing errors is the central challenge for useful quantum computing[1], requiring quantum error correction (QEC)[2–6] for large-scale processing. However, the overhead in the realization of error-corrected 'logical' qubits, in which information is encoded across many physical qubits for redundancy[2–4], poses substantial challenges to large-scale logical quantum computing. Here we report the realization of a programmable quantum processor based on encoded logical qubits operating with up to 280 physical qubits. Using logical-level control and a zoned architecture in reconfigurable neutral-atom arrays[7], our system combines high two-qubit gate fidelities[8], arbitrary connectivity[7,9], as well as fully programmable single-qubit rotations and mid-circuit readout[10–15]. Operating this logical processor with various types of encoding, we demonstrate improvement of a two-qubit logic gate by scaling surface-code[6] distance from $d = 3$ to $d = 7$, preparation of colour-code qubits with break-even fidelities[5], fault-tolerant creation of logical Greenberger–Horne–Zeilinger (GHZ) states and feedforward entanglement teleportation, as well as operation of 40 colour-code qubits. Finally, using 3D [[8,3,2]] code blocks[16,17], we realize computationally complex sampling circuits[18] with up to 48 logical qubits entangled with hypercube connectivity[19] with 228 logical two-qubit gates and 48 logical CCZ gates[20]. We find that this logical encoding substantially improves algorithmic performance with error detection, outperforming physical-qubit fidelities at both cross-entropy benchmarking and quantum simulations of fast scrambling[21,22]. These results herald the advent of early error-corrected quantum computation and chart a path towards large-scale logical processors.

Quantum computers have the potential to substantially outperform their classical counterparts for solving certain problems[1]. However, executing large-scale, useful algorithms on quantum processors requires very low gate error rates (generally below about $10^{-10}$)[23], far below those that will probably ever be achievable with any physical device[2]. The landmark development of QEC theory provides a conceptual solution to this challenge[2–4]. The key idea is to use entanglement to delocalize a logical qubit degree of freedom across many redundant physical qubits, such that, if any given physical qubit fails, it does not corrupt the underlying logical information. In principle, with sufficiently low physical error rates and sufficiently many qubits, a logical qubit can be made to operate with extremely high fidelity, providing a path to realizing large-scale algorithms[4]. However, in practice, useful QEC poses many challenges, ranging from large overhead in physical qubit numbers[23] to highly complex gate operations between the delocalized logical degrees of freedom[24].

Recent experiments have achieved milestone demonstrations of two logical qubits and one entangling gate[5,6] and explorations of new encodings[25–28].

One specific challenge for realizing large-scale logical processors involves efficient control. Unlike modern classical processors that can efficiently access and manipulate many bits of information[29], quantum devices are typically built such that each physical qubit requires several classical control lines. Although suitable for the implementation of physical qubit processors, this approach poses a substantial obstacle to the control of logical qubits redundantly encoded over many physical qubits.

Here we describe the realization of a programmable quantum processor based on hardware-efficient control over logical qubits in reconfigurable neutral-atom arrays[7]. We use this logical processor to demonstrate key building blocks of QEC and realize programmable logical algorithms. In particular, we explore important features of

[1]Department of Physics, Harvard University, Cambridge, MA, USA. [2]QuEra Computing Inc., Boston, MA, USA. [3]Joint Center for Quantum Information and Computer Science, NIST/University of Maryland, College Park, MD, USA. [4]John A. Paulson School of Engineering and Applied Sciences, Harvard University, Cambridge, MA, USA. [5]Department of Physics and Research Laboratory of Electronics, Massachusetts Institute of Technology, Cambridge, MA, USA. ✉e-mail: lukin@physics.harvard.edu

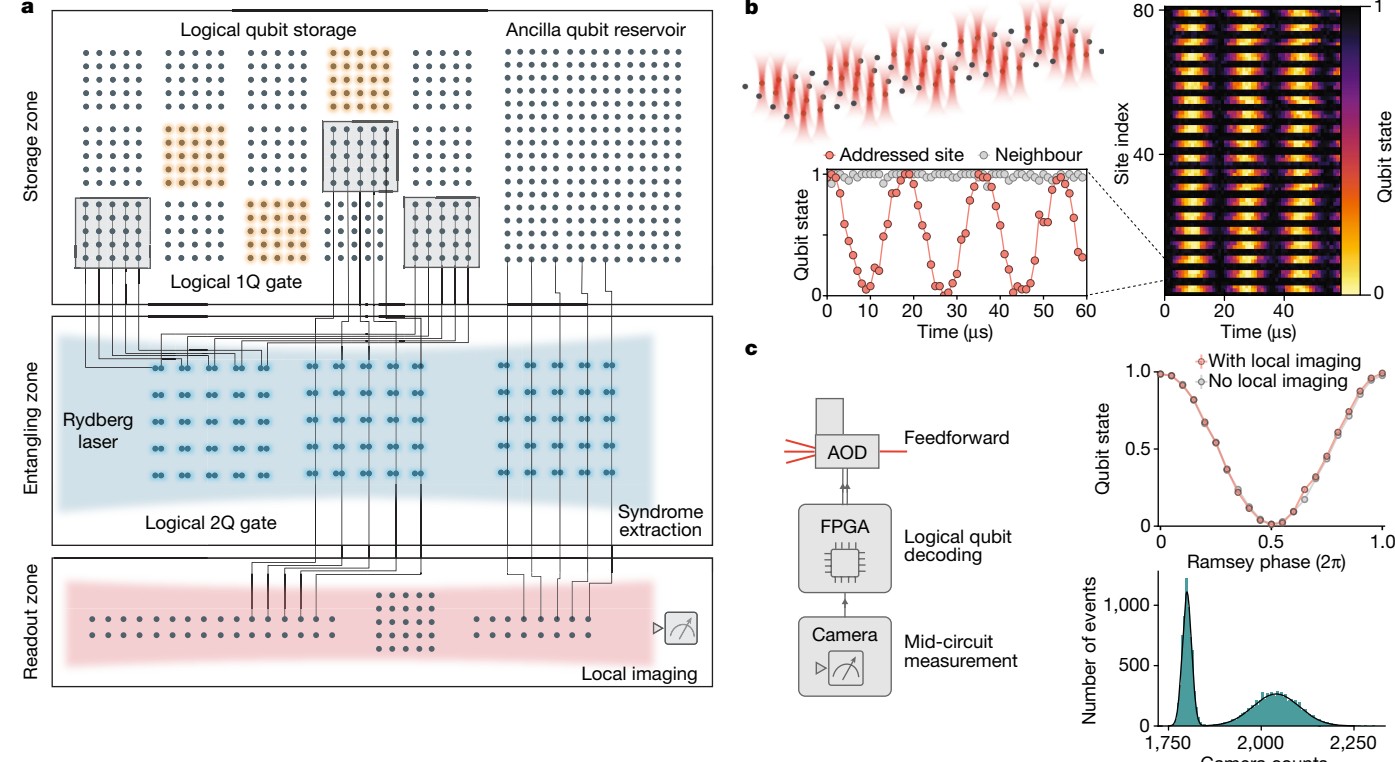

**Fig. 1 | A programmable logical processor based on reconfigurable atom arrays. a**, Schematic of the logical processor, split into three zones: storage, entangling and readout (see Extended Data Fig. 1 for detailed layout). Logical single-qubit and two-qubit operations are realized transversally with efficient, parallel operations. Transversal CNOTs are realized by interlacing two logical qubit grids and performing a single global entangling pulse that excites atoms to Rydberg states. Physical qubits are encoded in hyperfine ground states of [87]Rb atoms trapped in optical tweezers. **b**, Fully programmable single-qubit rotations are implemented using Raman excitation through a 2D AOD; parallel grid illumination delivers the same instruction to multiple atomic qubits. **c**, Mid-circuit readout and feedforward. The imaging histogram shows high-fidelity state discrimination (500 μs imaging time, readout fidelity approximately 99.8%; Methods) and the Ramsey fringe shows that qubit coherence is unaffected by measuring other qubits in the readout zone (error probability $p \approx 10^{-3}$; Methods). The FPGA performs real-time image processing, state decoding and feedforward (Fig. 4).

logical operations and circuits, including scaling to large codes, fault tolerance and complex non-Clifford circuits.

## Logical processor based on atom arrays

Our logical processor architecture, illustrated in Fig. 1a, is split into three zones (see also Extended Data Fig. 1). The storage zone is used for dense qubit storage, free from entangling-gate errors and featuring long coherence times. The entangling zone is used for parallel logical qubit encoding, stabilizer measurements and logical gate operations. Finally, the readout zone enables mid-circuit readout of desired logical or physical qubits, without disturbing the coherence of the computation qubits still in operation. This architecture is implemented using arrays of individual [87]Rb atoms trapped in optical tweezers, which can be dynamically reconfigured in the middle of the computation while preserving qubit coherence[7,9].

Our experiments make use of the apparatus described previously in refs. 7,8,30, with key upgrades enabling universal digital operation. Physical qubits are encoded in clock states within the ground-state hyperfine manifold ($T_2 > 1s$ (ref. 7)) and stored in optical tweezer arrays created by a spatial light modulator (SLM)[30,31]. We use systems of up to 280 atomic qubits, combining high-fidelity two-qubit gates[8], enabled by fast excitation into atomic Rydberg states interacting through robust Rydberg blockade[32], with arbitrary connectivity enabled by atom transport by means of 2D acousto-optic deflectors (AODs)[7]. Central to our approach of scalable control, AODs[10-15,31,33] use frequency multiplexing to take in just two voltage waveforms (one for each axis) to create large, dynamically programmable grids of light. Fully programmable local single-qubit rotations are realized through qubit-specific, parallel Raman excitation through an additional 2D AOD (ref. 34) (Fig. 1b and Extended Data Fig. 2). Mid-circuit readout is enabled by moving selected qubits about 100 μm away to a readout zone and illuminating with a focused imaging beam[7,35], resulting in high-fidelity imaging, as well as negligible decoherence on stored qubits (Fig. 1c and Extended Data Fig. 3). The mid-circuit[10-15] image is collected with a CMOS camera and sent to a field-programmable gate array (FPGA) for real-time decoding and feedforward.

The central aspect of our logical processor is the control of individual logical qubits as the fundamental units, instead of individual physical qubits. To this end, we observe that, during most error-corrected operations, the physical qubits of a logical block are supposed to realize the same operation, and this instruction can be delivered in parallel with only a few control lines. This approach naturally multiplexes with optical techniques. For example, to realize a logical single-qubit gate[2], we use the Raman 2D AOD (Fig. 1b) to create a grid of light beams and simultaneously illuminate the physical qubits of the logical block with the same instruction. Such a gate is transversal[2], meaning that operations act on physical qubits of the code block independently. This transversal property further implies that the gate is inherently fault-tolerant[2], meaning that errors cannot spread within the code block (see Methods), thereby preventing a physical error from spreading into a logical fault. Crucially, a similar approach can realize logical entangling gates[2,4]. Specifically, we use the grids generated by our moving 2D AOD to pick up two logical qubits, interlace them in the entangling

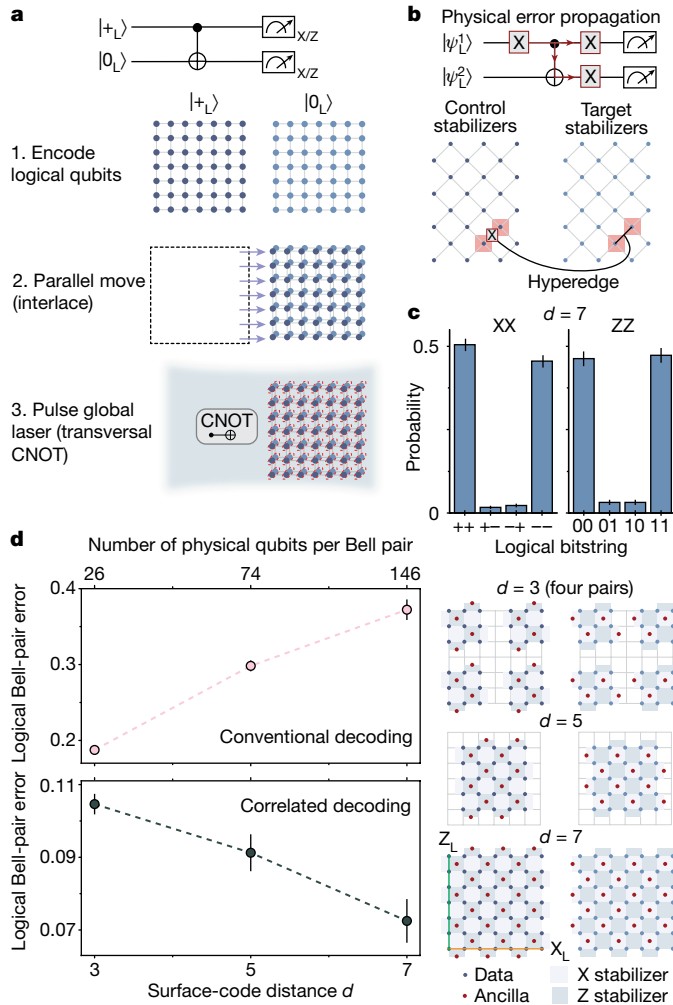

**Fig. 2 | Transversal entangling gates between two surface codes. a**, Illustration of transversal CNOT between two $d=7$ surface codes based on parallel atom transport. **b**, The concept of correlated decoding. Physical errors propagate between physical qubit pairs during transversal CNOT gates, creating correlations that can be used for improved decoding. We account for these correlations, arising from deterministic error propagation (as opposed to correlated error events), by adding edges and hyperedges that connect the decoding graphs of the two logical qubits. **c**, Populations of entangled $d=7$ surface codes measured in the XX and ZZ basis. **d**, Measured Bell-pair error as a function of code distance, for both conventional (top) and correlated (bottom) decoding. We estimate Bell error with the average of the ZZ populations and the XX parities (Methods). To reduce code distance, we simply remove selected atoms from the grid, as shown on the right, ensuring unchanged experimental conditions (for $d=3$, four logical Bell pairs are generated in parallel). Error bars represent the standard error of the mean. See Extended Data Figs. 4 and 5 for further surface-code data.

zone and then pulse our single global Rydberg excitation laser to realize a physical entangling gate on each twin pair of the blocks (Figs. 1a and 2a). This process realizes a high-fidelity, fault-tolerant transversal CNOT in a single parallel step.

## Improving entangling gates with code distance

A key property of QEC codes is that, for error rates below some threshold, the performance should improve with system size, associated with a so-called code distance[4,24]. Recently, this property has been experimentally verified by reducing idling errors of a code[6]. Neutral-atom qubits can be idly stored for long times with low errors, and the central

challenge is to improve entangling operations with code distance. Thus motivated, we realize a transversal CNOT gate using logical qubits encoded in two surface codes (Fig. 2). Surface codes have stabilizers that are used for detecting and correcting errors without disrupting the logical state[4,24]. The stabilizers form a 2D lattice of four-body plaquettes of X and Z operators, which commute with the $X_L$ ($Z_L$) logical operators that run horizontally (vertically) along the lattice (Fig. 2d). By measuring stabilizers, we can detect the presence of physical qubit errors, decode (infer what error occurred) and correct the error simply by applying a software $Z_L/X_L$ correction[24]. Such a code can detect and correct a certain number of errors determined by the linear dimension of the system (the code distance $d$).

To test the performance of our logical entangling gate, we first initialize the logical qubits by preparing physical qubits of two blocks in $|+\rangle$ and $|0\rangle$ states, respectively, and performing a single round of stabilizer measurements with parallel operations[7]. Although this state preparation is non-fault-tolerant (nFT) beyond $d=3$, we are still able to study error suppression of the transversal CNOT (Methods). Specifically, we prepare the two logicals in state $|+_L\rangle$ and $|0_L\rangle$, perform the transversal CNOT and then projectively measure to evaluate the logical Bell-state stabilizers $X_L^1 X_L^2$ and $Z_L^1 Z_L^2$ (Fig. 2c). For decoding and correcting the logical state, we observe that there are strong correlations between the stabilizers of the two blocks (Extended Data Figs. 4 and 5) owing to propagation of physical errors between the codes during the transversal CNOT (ref. 36) (Fig. 2b). We use these correlations to improve performance by decoding the logical qubits jointly, realized by a joint decoding graph that includes edges and hyperedges connecting the stabilizers of the two logical qubits (Fig. 2b, Methods). Using this correlated decoding procedure, we measure roughly 0.95 populations in the $X_L X_L$ and $Z_L Z_L$ bases (Fig. 2c), showing entanglement between the $d=7$ logical qubits.

Studying the performance as a function of code size (Fig. 2d) reveals that the logical Bell pair improves with larger code distance, demonstrating improvement of the entangling operation. By contrast, we note that, when conventional decoding, that is, independent minimum-weight perfect matching within both codes[4], is used, the fidelity decreases with code distance. This is in part because of the nFT state preparation, whose effect is partially mitigated by the correlated decoding (Methods).

We emphasize that, although these results demonstrate surpassing an effective threshold for the entire circuit (implying that we surpass the threshold of the transversal CNOT), such a threshold is higher owing to projective readout after the transversal CNOT. In practice, the transversal CNOT should be used in combination with many repeated rounds of noisy syndrome extraction[6], which is expected to have a lower threshold and is an important goal for future research.

## Fault-tolerant logical algorithms

All logical algorithms we perform in this work are built from transversal gates, which are intrinsically fault-tolerant[2]. We now also use fault-tolerant state preparation to explore programmable logical algorithms. We use 2D $d=3$ colour codes[3,37], which are topological codes akin to the surface code, but with the useful capability of transversal operations of the full Clifford group: Hadamard (H), π/2 phase (S) gate and CNOT (ref. 37). This transversal gate set can realize any Clifford circuit fault-tolerantly. As a test case, here we create a logical GHZ state. Figure 3a shows the implementation of a ten-logical-qubit algorithm, in which all ten qubits are first encoded by a nFT encoding circuit (Methods). Then, five of the codes are used as ancilla logicals, performing parallel transversal CNOTs to fault-tolerantly detect errors on the computation logicals[38], and are then moved into the storage zone, in which they are safely kept. Subsequently, four computation logicals are used to prepare the GHZ state and logical Clifford rotations

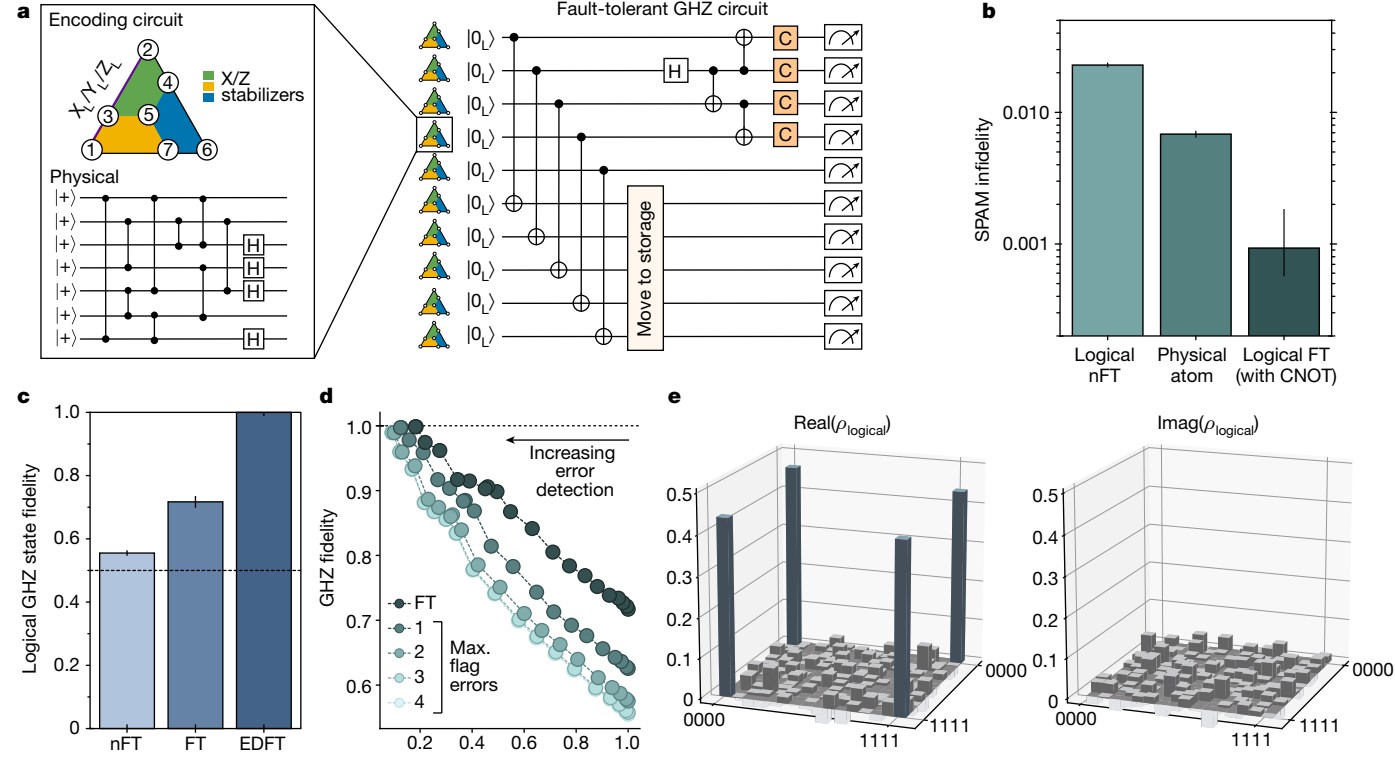

**Fig. 3 | Fault-tolerant logical algorithms. a**, Circuit for preparation of logical GHZ state. Ten colour codes are encoded non-fault-tolerantly and then parallel transversal CNOTs between computation and ancilla logical qubits perform fault-tolerant initialization. The ancilla logical qubits are moved to storage and a four-logical-qubit GHZ state is created between the computation qubits. Logical Clifford operations are applied before readout to examine the GHZ state. **b**, SPAM infidelity of the logical qubits without (nFT) and with (FT) the transversal-CNOT-based flagged preparation, compared with physical

qubit SPAM. **c**, Logical GHZ fidelity without postselecting on flags (nFT), postselecting on flags (FT) and postselecting on flags and stabilizers of the computation logical qubits, corresponding to error detection (EDFT). **d**, GHZ fidelity as a function of sliding-scale error-detection threshold (converted into the probability of accepted repetitions) and of the number of successful flags in the circuit. **e**, Density matrix of the four-logical-qubit GHZ state (with at most three flag errors) measured by means of full-state tomography involving all 256 logical Pauli strings.

are used at the end of the circuit for direct fidelity estimation[39] and full logical state tomography.

We first benchmark our state initialization[5,40,41] (Fig. 3b). Averaged over the five computation logicals, we find that, by using the fault-tolerant initialization (postselecting on the ancilla logical flag not detecting errors) our $|0_L\rangle$ initialization fidelity is $99.91^{+0.04}_{-0.09}\%$, exceeding both our physical qubit $|0\rangle$ initialization fidelity (99.32(4)% (ref. 8)) and physical two-qubit gate fidelity (99.5% (ref. 8)). Then, Fig. 3c shows that the resulting GHZ state fidelity obtained using the fault-tolerant algorithm is 72(2)% (again using correlated decoding), demonstrating genuine multipartite entanglement. Furthermore, we can postselect on all stabilizers of our computation logicals being correct; using this error-detection approach, the GHZ fidelity increases to $99.85^{+0.1}_{-1.0}\%$, at the cost of postselection overhead.

Because not all nontrivial syndromes are equally likely to cause algorithmic failure, we can perform a partial postselection, in which syndrome events most likely to have caused algorithmic failure are discarded, given by the weight of the correlated matching in the whole algorithm. Figure 3d shows the measured GHZ fidelity as a function of this sliding threshold converted into a fraction of accepted experimental repetitions, continuously tuning the trade-off between the success probability of the algorithm and its fidelity; for example, discarding just 50% of the data improves GHZ fidelity to approximately 90%. (As discussed below, for certain applications, purifying samples can be advantageous in improving algorithmic performance.) Finally, fault-tolerantly measuring all 256 logical Pauli strings, we perform full GHZ state tomography (Fig. 3e).

The use of the zoned architecture directly allows scaling circuits to larger numbers, without increasing the number of controls, by encoding and operating on logical qubits, moving them to storage and then accessing storage as appropriate. This process is illustrated in Fig. 4a,b, in which ten colour codes are made and operated on with parallel transversal CNOTs, moved to storage and then more qubits are accessed from storage. Repeating this process four times, we create 40 colour codes with 280 physical qubits, at the cost of slow idling errors of roughly 1% logical decoherence per additional encoding step (Fig. 4c). These storage idling errors primarily originate from global Raman π pulses applied for dynamical decoupling of atoms in the entangling zone, which could be greatly reduced with zone-specific Raman controls.

Because mid-circuit readout[10-15] is an important component of logical algorithms, we next demonstrate a fault-tolerant entanglement teleportation circuit. We first create a three-logical-qubit GHZ state $|0_L0_L0_L\rangle + |1_L1_L1_L\rangle$ (Fig. 4d,e) from fault-tolerantly prepared colour codes. Mid-circuit X-basis measurement of the middle logical creates $|0_L0_L\rangle + |1_L1_L\rangle$ if measured as $|+_L\rangle$ and $|0_L0_L\rangle - |1_L1_L\rangle$ if measured as $|-_L\rangle$. We recover $|0_L0_L\rangle + |1_L1_L\rangle$ by applying a logical S gate to the first and third logicals conditioned in real time on the state of the middle logical, akin to the magic-state-teleportation circuit[24]. Measurements in Fig. 4e indicate that, although $\langle X_LX_L\rangle$ and $\langle Y_LY_L\rangle$ indeed vanish without the feedforward step, by applying the feedforward correction, we recover a Bell-state fidelity of 77(2)%, limited by imperfections in the original underlying GHZ state. By repeating this experiment without mid-circuit readout and instead postselecting on the middle logical being in $|+_L\rangle$,

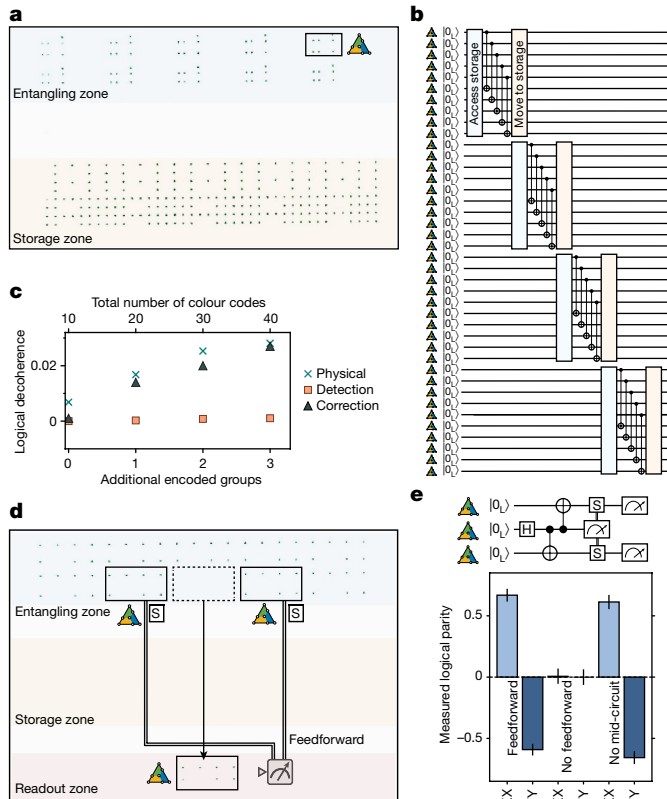

**Fig. 4 | Zoned logical processor: scaling and mid-circuit feedforward.**
**a**, Atoms in storage and entangling zones and approach for creating and entangling 40 colour codes with 280 physical qubits. **b,c**, 40 colour codes are prepared with a nFT circuit and then 20 transversal CNOTs are used to fault-tolerantly prepare 20 of the 40 codes, whose fidelity is plotted. Logical decoherence is smaller than the physical idling decoherence experienced during the encoding steps. **d**, Mid-circuit measurement and feedforward for logical entanglement teleportation. The middle of three logical qubits is measured in the X basis and, by applying a mid-circuit conditional, locally pulsed logical S rotation on the other two logical qubits, the state $|0_L0_L\rangle + |1_L1_L\rangle$ is prepared. **e**, Measured logical qubit parity with and without feedforward, showing that feedforward recovers the intended state with Bell fidelity of 77(2)% (ZZ parities of 83(4)% not plotted; Methods). No mid-circuit refers to turning off the mid-circuit readout and postselecting on the middle logical being in state $|+_L\rangle$ in the final readout. By postselecting on perfect stabilizers of only the two computation logicals (error detection in the final measurement), the feedforward Bell fidelity is 92(2)% (not plotted). In **d**, three of the extra blocks are flag qubits and the other four are prepared but unused for this circuit.

we find a similar Bell fidelity of 75(2)%, indicating high-fidelity performance of the readout and feedforward operations.

## Complex logical circuits using 3D codes

One important challenge in realizing complex algorithms with logical qubits is that universal computation cannot be implemented transversally[42]. For instance, when using 2D codes such as the surface code, non-Clifford operations cannot be easily performed[37], and relatively expensive techniques are required for nontrivial computation[24,43], as Clifford circuits can be easily simulated[44]. By contrast, 3D codes can transversally realize non-Clifford operations, but lose the transversal H (ref. 37). However, these constraints do not imply that classically hard or useful quantum circuits cannot be realized transversally or efficiently. Motivated by these considerations, we explore efficient realization of classically hard algorithms that are co-designed with a particular error-correcting code. Specifically, we implement fast

scrambling circuits using small 3D codes, which are used for native non-Clifford operations (CCZ).

We focus on small 3D [[8,3,2]] codes[16,17,26,27] (Fig. 5a), which have various appealing features. They encode three logicals per block, feature $d = 2$ ($d = 4$) in the Z basis (X basis), implying error-detection (error-correction) capabilities for Z (X) errors and can realize a transversal CNOT between blocks. Most importantly, by using physical {T, S} rotations (T is π/4 phase gate), we can realize transversal {CCZ, CZ, Z} gates on the logical qubits encoded within each block, as well as intra-block CNOTs by physical permutation[26,27] (Methods). This gate set allows us to transversally realize the circuits illustrated in Fig. 5a,c, alternating between layers of {CCZ, CZ, Z} within blocks and layers of CNOTs between blocks. Although transversal H is forbidden, initialization and measurement in either the X or the Z basis effectively allows H at the beginning and end of the circuit.

We use these transversal operations to realize logical algorithms that are difficult to simulate classically[45,46]. More specifically, these circuits can be mapped to instantaneous quantum polynomial (IQP) circuits[20,45,46]. Sampling from the output distribution of such circuits is known to be classically hard in certain instances[20], implying that a quantum device can be exponentially faster than a classical computer for this task.

Figure 5b shows an example implementation of a 12-logical-qubit sampling circuit. Here we prepare all logical blocks in $|+_L\rangle$, implement a scrambling circuit with 28 logical entangling gates and then measure all logicals in the X basis. Figure 5b shows the probability of observing each of the $2^{12} = 4,096$ possible logical bitstring outcomes, showing that, as we progressively apply more error detection (that is, postselection) in post-processing, the distribution more closely reproduces the ideal theoretical distribution. To characterize the distribution overlap, we use the cross-entropy benchmark (XEB)[18,47], which is a weighted sum between the measured probability distribution and the ideal calculated distribution, normalized such that XEB = 1 corresponds to perfectly reproducing the ideal distribution and XEB = 0 corresponds to the uniform distribution, which occurs when circuits are overwhelmed by noise. Consistent with Fig. 5b, the 12-logical-qubit circuit XEB increases from 0.156(2) to 0.616(7) when applying error detection (Fig. 5e). We note that the XEB should be a good fidelity benchmark for IQP circuits (Methods).

We next explore scaling to larger systems and circuit depths. To ensure high complexity of our logical circuits, we use nonlocal connections to entangle the logical triplets on up to 4D hypercube graphs (Extended Data Fig. 6 and Supplementary Video 3), which results in fast scrambling[19]. Exploring entangled systems of 3, 6, 12, 24 and 48 logical qubits, in all cases, we find a finite XEB score, which improves with increased error detection (Fig. 5e,f). The finite XEB indicates successful sampling and the improvement with error detection shows the benefit of using logical qubits. Although this improvement comes at the cost of measurement time owing to error detection, improving the sample quality cannot be replaced by simply generating more samples. Thus, improving the XEB score yields substantial practical gains. We obtain an XEB of approximately 0.1 for 48 logical qubits and hundreds of nonlocal logical entangling gates, up to roughly an order of magnitude higher than previous physical qubit implementations of digital circuits of similar complexity[18,48], showing the benefits of a logical encoding for this application.

Assuming our best measured physical fidelities, the estimated upper bound for an optimized physical qubit implementation in our system is also greatly below the measured logical XEB (blue line in Fig. 5f; Methods). In attempting to run these complex physical circuits, in practice, we find that realizing non-vanishing XEB is much more challenging; we confirm with small physical instances that we measure values well below this upper bound (Methods). As well as the error-detecting benefits, it seems that the logical circuit is substantially more tolerant to coherent errors, exhibiting operation that is inherently digital,

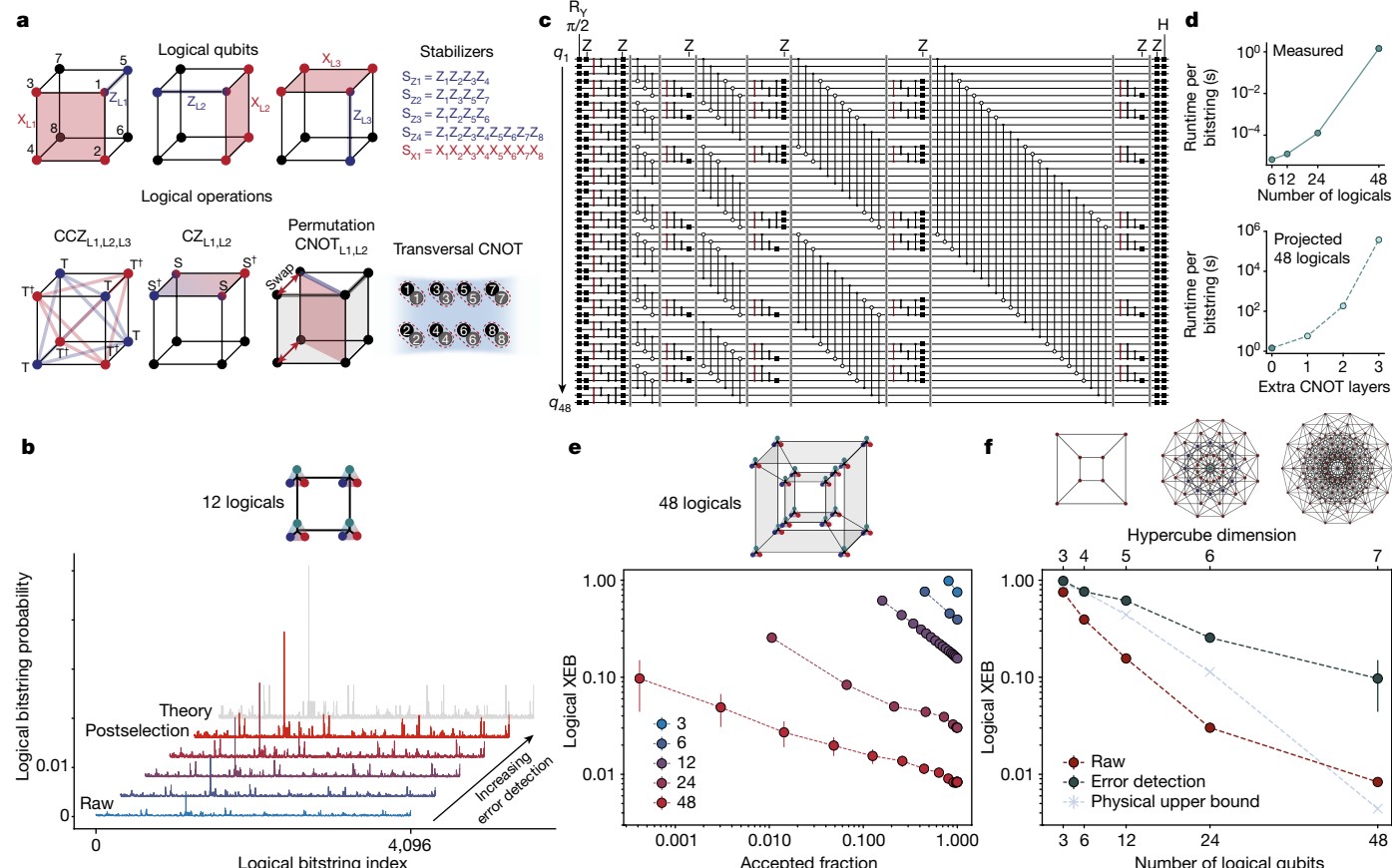

**Fig. 5 | Complex logical circuits using 3D codes. a**, [[8,3,2]] block codes can transversally realize {CCZ, CZ, Z, CNOT} gates within each block and transversal CNOTs between blocks. By preparing logical qubits in |+$_L$⟩, performing layers of {CCZ, CZ, Z} alternated with inter-block CNOTs and measuring in the X basis, we realize classically hard sampling circuits with logical qubits. **b**, Measured sampling outcomes for a circuit with 12 logical qubits, eight logical CZs, 12 logical CNOTs and eight logical CCZs. By increasing error detection, the measured distribution converges towards the ideal distribution. **c**, Circuit involving 48 logical qubits with 228 logical CZ/CNOT gates and 48 logical CCZs. **d**, Classical simulation runtime for calculating an individual bitstring probability; bottom plot is estimated on the basis of matrix multiplication complexity. **e**, Measured normalized XEB as a function of sliding-scale error detection for 3, 6, 12, 24 and 48 logical qubits. For all sizes, we observe a finite XEB score that improves with increased error detection. Diagram shows 48-logical connectivity, with logical triplets entangled on a 4D hypercube. **f**, Scaling of raw (red) and fully error-detected (black) XEB from **e**. Physical upper-bound fidelity (blue) is calculated using best measured physical gate fidelities (see Methods and Extended Data Fig. 7 for scaling discussion). Diagrams show physical connectivity. [[8,3,2]] cubes are entangled on 4D hypercubes, realizing physical connectivity of 7D hypercubes.

---

just with imperfect fidelity (see, for example, Extended Data Fig. 7a), consistent with theoretical predictions[49]. We also note that, for the logical algorithms, we optimize performance by optimizing the stabilizer expectation values (rather than the complex sampling output), providing further advantage for logical implementations.

Our 48-logical circuit, corresponding to a physical qubit connectivity of a 7D hypercube, contains up to 228 logical two-qubit gates and 48 logical CCZ gates. Simulation of such logical circuits is challenging because of the high connectivity (rendering tensor networks inefficient) and large numbers of non-Cliffords[50]. To benchmark our circuits, we structure them such that we can use an efficient simulation method (Methods), which takes about 2 s to calculate the probability of each bitstring (Fig. 5d and Extended Data Fig. 8). Modelling noise in our logical circuits is even more complicated, as they are composed of 128 physical qubits and 384 T gates, thereby making experimentation with logical algorithms necessary to understand and optimize performance.

## Quantum simulations with logical qubits

Finally, we explore the use of logical qubits as a tool in quantum simulation, probing entanglement properties of our fast scrambling circuits, potentially related to complex systems such as black holes[19,51].

In particular, we use a Bell-basis measurement made on two copies of the quantum state (Fig. 6a), which is a powerful tool that can efficiently extract many properties of an unknown state[21,22,52]. With this two-copy technique, in Fig. 6b, we plot the measured entanglement entropy in the scrambled system. We observe a characteristic Page curve[51] associated with a maximally entangled, highly scrambled, but globally pure state. These measurements also reveal a final state purity of 0.74(3), compared with the measured XEB of 0.616(7) in Fig. 5f, consistent with the XEB being a good proxy for the final state fidelity. Despite postselection overhead, we find that error detection greatly improves signal to noise here, as near-zero entropies are exponentially faster to measure (Extended Data Fig. 9).

Two-copy measurements can also be used to simultaneously extract information about all 4$^N$ Pauli strings[22]. Using this property and an analysis technique known as Bell difference sampling[53], we experimentally evaluate and directly verify the amount of additive Bell magic[53] in our circuits as a function of the number of applied logical CCZs (Fig. 6c). This measurement of magic, associated with non-Clifford operations, quantifies the number of T gates (assuming decomposition into T) required to realize the quantum state by observing the probability that sampled Pauli strings commute with each other (see Methods). Moreover, combining encoded qubits and two-copy measurement

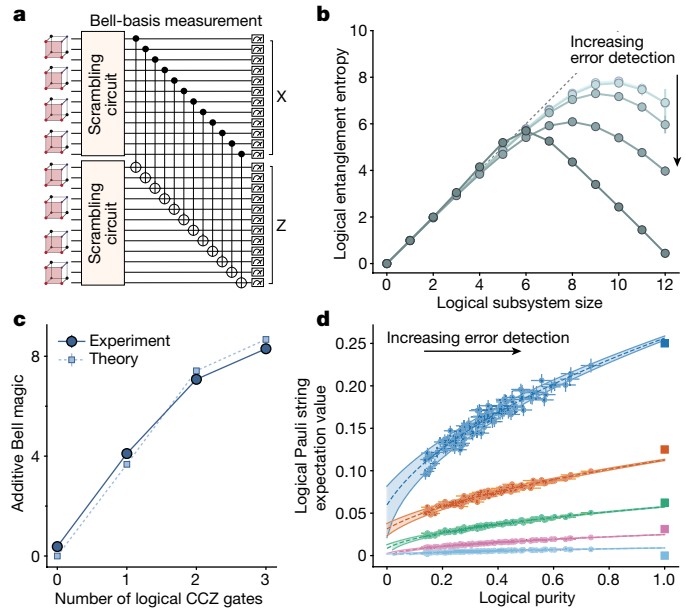

**Fig. 6 | Logical two-copy measurement. a**, Identical scrambling circuits are performed on two copies of 12 logical qubits and then measured in the Bell basis to extract information about the state. Z-basis measurements are corrected with an [[8,3,2]] decoder (when full error detection is not applied). **b**, Measured entanglement entropy as a function of subsystem size, showing expected Page-curve behaviour[19] for the highly scrambled state, improving with increased error detection. **c**, Measured and simulated magic (associated with non-Clifford operations) as a function of the number of CCZ gates applied, performed on two copies of scrambled six-logical-qubit systems. **d**, Pauli string measurement and zero-noise extrapolation using logical qubits. Plot shows the absolute values of all $4^{12}$ Pauli string expectation values, which only have five discrete values for our digital circuit; Pauli strings with the same theory value are grouped. By analysing with sliding-scale error detection, we improve towards the theoretical expectation values (squares) while also improving towards a purity of 1. By extrapolating to perfect purity, we extrapolate the expectation values and better approximate the ideal values (shaded regions are statistical fit uncertainty).

allows for further error-mitigation techniques. As an example, Fig. 6d shows the measured absolute expectation values of all $4^{12}$ logical Pauli strings with sliding-scale error detection. Because in the two-copy measurements for each error-detection threshold we also measure the overall system purity, we can extrapolate our expectation values to the case of unit purity (zero noise)[54]. This procedure evaluates the averaged Pauli expectation values to about 10% relative precision of the ideal theoretical values spanning several orders of magnitude (Methods).

## Outlook

These experiments demonstrate key ingredients of scalable error correction and quantum information processing with logical qubits. As well as implementing the key elements of logical processing, our approach demonstrates practical utility of encoding methods for improving sampling and quantum simulations of complex scrambling circuits. Future work can explore whether these methods can be generalized, for example, to more robust, higher-distance codes and if such highly entangled, non-Clifford states could be used in practical algorithms. We note that the demonstrated logical circuits are approaching the edge of exact simulation methods (Fig. 5d) and can readily be used for exploring error-corrected quantum advantage. These examples demonstrate that the use of new encoding schemes, co-designed with efficient implementations, can allow the implementation of particular logical algorithms at reduced cost.

Our observations open the door for exploration of large-scale logical qubit devices. A key future milestone would be to perform repetitive error correction[6] during a logical quantum algorithm to greatly extend its accessible depth. This repetitive correction can be directly realized using the tools demonstrated here by repeating the stabilizer measurement (Fig. 2) in combination with mid-circuit readout (Fig. 4). The use of the zoned architecture and logical-level control should allow our techniques to be readily scaled to more than 10,000 physical qubits by increasing laser power and optimizing control methods, whereas QEC efficiency can be improved by reducing two-qubit gate errors to 0.1% (ref. 8). Deep computation will further require continuous reloading of atoms from a reservoir source[11,15]. Continued scaling will benefit from improving encoding efficiency, for example, by using quantum low-density-parity-check codes[55,56], using erasure conversion[13,33,57] or noise bias[35] and optimizing the choice of (possibly several) atomic species[11,14,47], as well as advanced optical controls[34]. Further advances could be enabled by connecting processors together in a modular fashion using photonic links or transport[10,58] or more power-efficient trapping schemes such as optical lattices[59]. Although we do not expect clock speed to limit medium-scale logical systems, approaches to speed up processing in hardware[60] or with nonlocal connectivity[61] should also be explored. We expect that such experiments with early-generation logical devices will enable experimental and theoretical advances that greatly reduce anticipated costs of large-scale error-corrected systems, accelerating the development of practical applications of quantum computers.

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

## Methods

### System overview

Our experimental apparatus (Extended Data Fig. 1a) is described previously in refs. 7,8,30. To carry out these experiments, several key upgrades have been made enabling programmable quantum circuits on both physical and logical qubits. A cloud containing millions of cold [87]Rb atoms is loaded in a magneto-optical trap inside a glass vacuum cell, which are then loaded stochastically into programmable, static arrangements of 852-nm traps generated with a SLM and rearranged with a set of 850-nm moving traps generated by a pair of crossed AODs (DTSX-400, AA Opto-Electronic) to realize defect-free arrays[30,31,62]. Atoms are imaged with a 0.65-NA objective (Special Optics) onto a CMOS camera (Hamamatsu ORCA-Quest C15550-20UP), chosen for fast electronic readout times. The qubit state is encoded in $m_F = 0$ hyperfine clock states in the [87]Rb ground-state manifold, with $T_2 > 1s$ (ref. 7), and fast, high-fidelity single-qubit control is executed by two-photon Raman excitation[7,63] (Extended Data Fig. 1b). A global Raman path illuminating the entire array is used for global rotations (Rabi frequency roughly 1 MHz, resulting in approximately 1-μs rotations with composite pulse techniques[7]), as well as for dynamical decoupling throughout the entire circuit (typically one global π pulse per movement). Fully programmable local single-qubit rotations are realized with the same Raman light but redirected through a local path, which is focused onto targeted atoms by an additional set of 2D AODs. Entangling gates (270-ns duration) between clock qubits are performed with fast two-photon excitation using 420-nm and 1,013-nm Rydberg beams to $n = 53$ Rydberg states, using a time-optimal two-qubit gate pulse[64,65], detailed in ref. 8. During the computation, atoms are rearranged with the AOD traps to enable arbitrary connectivity[7,66,67]. Mid-circuit readout is carried out by illuminating from the side with a locally focused 780-nm imaging beam, with scattered photons collected on the CMOS camera and processed in real time by a FPGA (Xilinx ZCU102), with feedforward control signal outputs.

The quantum circuits are programmed with a control infrastructure consisting of five arbitrary waveform generators (AWGs) (Spectrum Instrumentation), as illustrated in Extended Data Fig. 1c, synchronized to <10 ns jitter. The two-channel 'Rearrangement AWG' is used for rearranging into defect-free arrangements[30] before the circuit, the one channel of the 'Rydberg AWG' is used for entangling-gate pulses, the four channels of the 'Raman AWG' are used for IQ (in-phase and quadrature) control of a 6.8-GHz source[7,63] (the global phase reference for all qubits) and pulse-shaping of the global and local Raman driving, the two channels of the 'Raman AOD AWG' are used for generating tones that create the programmable light grids for local single-qubit control and the two channels of the 'Moving AOD AWG' are used for controlling the positions of all atoms during the circuit. AODs are central to our methods of efficient control[62], in which the two voltage waveforms (one for the $x$ axis and one for the $y$ axis) control many physical or logical qubits in parallel: each row and column of the grid simply corresponds to a single frequency tone, and these tones are then superimposed in the waveform delivered to the AOD (amplified by Mini Circuits ZHL-5W-1+). The phase relationship between tones is chosen to minimize interference.

### Programming circuits

Most of the system parameters used in our approach do not have hard limits but instead result from possible trade-offs. Next, we detail some design decisions made for the circuits used in this work.

**Zone parameter choices.** For simplicity, we keep the entangling zone fixed for all experiments. This conveniently allows us to switch between, for example, surface code and [[8,3,2]] code experiments, without further calibrations. We choose our entangling-zone profile, realized by 420-nm and 1,013-nm Rydberg 'top-hat' beams generated by SLM phase

profiles[30], to be homogeneous over a 35-μm-tall region. As the Rydberg beams propagate longitudinally, the entangling zone is longer than it is tall. We optimize top hats to be homogeneous over roughly 250-μm horizontal extent. Taller regions are also achievable, with a trade-off with reduced laser intensity and greater challenge in homogenization. The 250-μm width of the zones used here is set by the bandwidth of our AOD deflection efficiency. We position the readout zone on the other side of the storage zone to further minimize decoherence on entangling-zone atoms.

Our two-qubit gate parameters are similar to our previous work in ref. 8. During two-qubit Rydberg ($n = 53$) gates, we place atoms ≲2 μm apart within a 'gate site', resulting in ≥450 MHz interaction strength between pairs, much larger than the Rabi frequency of 4.6 MHz. Notably, owing to the use of the Rydberg blockade[32,68], the gate is largely independent of the exact distance between atoms. Hence, precise inter-atom positioning is not required. Gate sites are separated such that atoms in different gate sites are no closer than 10 μm during the gate, resulting in negligible long-range interactions. Throughout this work, we use four gate sites vertically (five for the surface-code experiment) and 20 horizontally, performing gates on as many as 160 qubits simultaneously (see Extended Data Fig. 1d). Under various conditions, with proper calibration, we measure two-qubit gate fidelities in the range $F = 99.3$–$99.5\%$. We do not observe any error on storage-zone atoms when Rydberg gates are executed in the entangling zone. Even though the tail of the top-hat Rydberg excitation beams is only suppressed to about 10% intensity, the two-photon drive is far off-resonant owing to the approximately 20 MHz 1013 light shift detuning that is present for the entangling-zone atoms[8]. We natively realize physical CZ gates; when implementing CNOTs, we add physical H gates. We find minimal two-qubit cross-talk between gate sites, as examined with long benchmarking sequences in ref. 8. Although ref. 8 seems to find some small cross-talk seemingly originating from decay into Rydberg P states, this should be considerably suppressed in the practical operation here owing to the approximately 200 μs duration between gates, during which time Rydberg atoms should either fly away or decay back to the ground state.

**Shuttling and transfers.** The SLM tweezers can have arbitrary positions but are static. The AOD tweezers are mobile but have several constraints[7,69]. In particular, the AOD array creates rectangular grids (but not all sites need to be filled). During the atom-moving operations, they are only used for stretches, compressions and translations of the AOD trap array, that is, atoms move in rows and columns, and rows and columns never cross[7,69]. Arbitrary qubit motions and permutation is achieved by shuttling atoms around in AOD tweezers and then transferring atoms between AOD and SLM tweezers as appropriate. We perform gates on pairs of atoms in both AOD–AOD traps and AOD–SLM traps, with no observed difference for gate performance as measured by randomized benchmarking[8].

We find that free-space shuttling of atoms (that is, no transfers) in AOD tweezers comes essentially with no fidelity cost (other than time overhead), consistent with our previous work[7]. Two further improvements here are the use of a photodiode to calibrate and homogenize the 2D deflection efficiency of our 2D AODs to percent-level homogeneity across our used region and engineering atomic trajectories and echo sequences to cancel out residual-path-dependent inhomogeneities. For example, we move an atom 100 μm away to realize a distant entangling gate and then, before returning the atom, we perform a Raman π pulse, so that differential light shifts accumulated during the return trip cancel with the first trip. Motion is realized with a cubic profile as in ref. 7, the characteristic free-space movement time between gates is roughly 200 μs and acoustic-lensing effects from the AOD are estimated to be negligible. We pulse the 1013 laser off during motion to remove loss effects from the large light shifts. Note that the 1013-induced differential light shift on the hyperfine

qubit is only on the kHz scale but we still ensure its effects are properly echoed out.

Transferring atoms between tweezers[9] presents further challenges. We measure the infidelity of each transfer, encompassing both dephasing and loss, to be ≲0.1%. To achieve this performance, in our transfer from SLM to AOD, we ramp up the intensity of the AOD tones (with quadratic intensity profile when possible) corresponding to the appropriate sites over a time of 100–200 μs to a trap depth about two times larger than the SLM trap depth, and then move the AOD trap 1–2 μm away over a time of 50–100 μs. These timescales can probably be shortened considerably while suppressing errors using optimal control techniques. During subsequent motion, we leave the AOD trap depth at this 2× value. To transfer an atom AOD to SLM, we perform the reversed process. During these transfer processes, the differential light shifts on the transferred atoms are dynamically changing and can result in large unechoed phase shifts. As such, whenever possible, we engineer circuits such that pairs of transfers will echo with appropriately chosen π pulses. When echoing pairs of transfers is not possible, we perform one cycle of XY4 or XY8 dynamical decoupling during the transfer. Finally, we note that low-loss transfer is highly sensitive to alignment of the AOD and SLM grid. We fix small optical distortions between the AOD and SLM tweezer grids by fine adjustment of individual SLM grid tweezers, which can be arbitrarily positioned, to overlap with the AOD traps as seen on an image plane reference camera. It is important to adjust the SLM and not the AOD, as small adjustments of individual AOD tones deviating from a frequency comb causes beating and atom loss.

**Dynamical decoupling and local gates.** In our circuit design, we engineer our echo sequences to cancel out as many deleterious aspects as possible. We ensure that, in our dynamical decoupling, we have an odd number of π pulses between CZ gates (whenever possible), as this echoes out both systematic and spurious contributions to the single-qubit phase[7,8]. We apply appropriate X(π) and Z(π) rotations between local addressing with the local Raman to cancel out errors induced by the global π/2 pulses, as well as between pulses of the 420-nm laser (when used for entangling-zone single-qubit rotations[7]) to echo out small cross-talk experienced in the storage zone by the tail of the 420-nm beam. For our global decoupling pulses, we use both BB1 pulses[70] and SCROFULOUS pulses[71]. To benchmark and optimize coherence during our complex circuits, we perform a Ramsey fringe measurement encompassing the entire movement and single-qubit gate sequence and optimize the observed contrast[7]. When performing properly, our total single-qubit error is consistent with state preparation and measurement (SPAM)[8], an effective coherence time of 1–2 s and the Raman scattering error of all the Raman pulses[7,63]. We note that these measured coherence times include the movement within and between zones; although we use fewer pulses (typically one per movement) than the XY16-128 sequence used to benchmark 1.5 s coherence in ref. 7, the coherence times here are naturally longer because of further-detuned tweezers used (852 nm rather than 830 nm).

Local single-qubit gates[34,72] with the Raman AOD are realized in arbitrary positions in space on both AOD and SLM atoms. Targeted logical qubit blocks are addressed by a grid illumination of the logical block. Arbitrary patterns of rotations on the qubit grid (for example, during colour-code preparation) are realized with row-by-row serializing, with the targeted x coordinates in each row simultaneously illuminated. The duration of each row is 5–8 μs (corresponding to several tens of μs for an arbitrary pattern of rotations), which can be sped up considerably, as discussed in the next section. For simplicity, we carefully calibrate rotations on 80–160 specific sites across the array, but also perform rotations in arbitrary spots using the nearest calibrated values.

With the local single-qubit gates and entangling-zone two-qubit gates calibrated, the entire circuit is simply defined by the appropriate trapping SLM phase profile and waveforms for our several AWG channels and TTL pulse generator. These several channels then program

complex, varied circuits on hundreds of physical qubits. Animations of all of our programmed circuits are attached as Supplementary Videos.

## Programmable single-qubit gates

To enable individual single-qubit gates, we use the same Raman laser system as our global rotation scheme and illuminate only chosen atoms using a pair of crossed AODs. The focused beam waist in the plane of the atoms is 1.9 μm, which is large enough to be robust to fluctuations in atomic positions and small enough to prevent cross-talk to neighbouring atoms separated by ≳6 μm. For Raman excitation, polarization needs to be carefully considered. Unlike the global path, the local beam-propagation direction is perpendicular to the atom-quantization axis (set by the external magnetic field). Therefore, the fictitious magnetic field $\vec{B}_{\text{fict}}$ responsible for driving the transitions, as described in ref. 63, preferentially drives $\sigma^\pm$ hyperfine transitions rather than the desired π clock transition[73]. There exist two possible approaches to single-qubit gates, as illustrated in Extended Data Fig. 2a. First, off-resonant $\sigma^\pm$ dressing generates differential light shifts between qubit states, enabling fast local Z($\theta$) gates. Global π/2 rotations convert these to local X($\theta$) gates. Second, we can directly apply local X($\theta$) gates with direct π transitions by slightly rotating the quantization axis towards the local beam direction; this could be achieved with an external field but, conveniently, $\vec{B}_{\text{fict}}$ has a DC component that naturally rotates the axis. Note that, if the local beam is quickly turned on, this same fictitious DC field causes leakage out of the $m_F = 0$ subspace, therefore Gaussian-smoothed pulses are used throughout this work.

Although we realize both the π and $\sigma^\pm$ versions above, in these experiments, we use the off-resonant $\sigma^\pm$ dressing procedure because of reduced polarization sensitivity, as our polarization homogeneity was affected by the sharp wavelength edge of a dichroic after the AOD. Furthermore, as for most circuits, we perform local rotations row by row (only one Y tone at a time); this enables arbitrary fine-tuning of X coordinates and powers at each site for homogenizing and calibrating rotations (Extended Data Fig. 2b). We calibrate using the procedure in Extended Data Fig. 2c and find that these calibrations are stable on month timescales.

To quantify the fidelity, we perform randomized benchmarking using 0, 10, 20, 30, 40 and 50 local Z(π/2) rotations (per site) on 16 sites, obtaining $\mathcal{F}$ = 99.912(7)%, as shown in Extended Data Fig. 2d (note that the single-qubit gates we execute globally have fidelity closer to 99.99% (refs. 7,8)). This approaches the Raman scattering limit for our $\sigma^\pm$ scheme (error of about $7 \times 10^{-4}$ per π/2 pulse), but when not well calibrated is limited by inhomogeneity, in particular, associated with distortions of the y position of the rows. In the future, the performance can be further improved by using X($\theta$) gates, which enables robust composite sequences such as BB1 (ref. 70), has an improved Raman scattering contribution and is faster (roughly 1 μs duration).

## Mid-circuit readout and feedforward

To perform mid-circuit readout[10–15] of selected qubits without affecting the others, we use a local imaging beam focused on the readout zone that is roughly 100 μm spatially separated from the entangling zone[7,35]. The local imaging beam consists of 780-nm circularly polarized light, with a near-resonant component from $F = 2$ to $F' = 3$ and a small repump component. This beam is sent through the side of our glass vacuum cell, co-propagating with the global Raman and 1,013-nm Rydberg beams (Extended Data Fig. 1a). We use cylindrical lenses to shape the beam, with focused beam waists of 30 μm in the plane of the atom array and 80 μm out of the plane. After moving some of the atoms to this readout zone, we first perform local pushout of population in the $F = 2$ ground-state manifold (by turning off the repump laser frequency), followed by local imaging of the remaining $F = 1$ population.

As depicted in Extended Data Fig. 3a, we collect an average of about 50 photons per imaged atom. To avoid losing the atoms too quickly during mid-circuit imaging (which, unlike our global imaging scheme,

does not have multi-axis cooling), we use deep (roughly 5-mK) traps (helping retain the atoms) and stroboscopically pulse them on and off out of phase of the local imaging light to avoid deleterious effects of the deep traps, such as inhomogeneous light shifts and fluctuating dipole force heating[74] (Extended Data Fig. 3b). From a double-Gaussian fit to the two distributions in Fig. 3a, we extract an imaging fidelity of more than 99.9%. Because this fit can lead to an overestimate of the imaging fidelity (for example, owing to atom loss during imaging), we compare the total SPAM error (measured by the amplitude of the Ramsey fringe) with local imaging versus with global imaging for the same state-preparation sequence, extracting 0.14(5)% higher error with local imaging; with these considerations, we conservatively estimate a local imaging fidelity of around 99.8%.

Various design considerations facilitate local imaging in the readout zone while preserving coherence of the data qubits in the entangling zone[35] (Extended Data Fig. 3e–g). The main sources of decoherence are rescattering of photons from the locally imaged atoms, as well as beam reflections and tails of the local imaging beam hitting the data qubits. As shown in Fig. 1c, for the 500-μs mid-circuit imaging used in this work, we are able to achieve unchanged coherence (identical within the error bars) of the data qubits with the local imaging light on as without it. To understand these effects more quantitatively, we measure the error probability of the data qubits in the entangling zone while the local imaging beam is on in the readout zone for up to 20 ms and with higher intensities than used for local imaging in this work. We suppress decoherence by light shifting the 780-nm transition of the data qubits to be different from that of the locally imaged qubits by several tens of MHz, as studied in Extended Data Fig. 3f–g. Data qubit decoherence is further suppressed by the large spatial separation between the readout zone and the entangling zone, in which intensity from the Gaussian tail of the local imaging beam should theoretically fall off rapidly. Even at large separations, we find that stray beam reflections (for example, from the glass cell window and other optical elements) can hit the data qubit region. To mitigate this effect, we displace reflections away from the atom array by angling the local imaging beam as it hits the glass cell window. The estimated effects of rescattered photons from the imaged atoms, especially with the added relative detuning, is negligible. With all these considerations, we find that we are able to suppress data qubit decoherence rates to ≲0.1% per 500 μs of local imaging exposure, as illustrated in Extended Data Fig. 3h.

The full mid-circuit readout and feedforward cycle occurs in slightly less than 1 ms, including local pushout, local imaging, readout of the camera pixels, decoding of the logical qubit state on the FPGA and a local Raman pulse, which is gated on or off by a conditional trigger (Extended Data Fig. 3d). In future work, this approach to mid-circuit readout and feedforward can be considerably improved to enable mid-circuit readout close to the 100-μs scale[75]. This method can be directly extended to perform many rounds of measurement and feedforward, in which groups of ancilla atoms are consecutively brought to the readout zone throughout a deep quantum circuit.

## Correlated decoding

During transversal CNOT operations, physical CNOT gates are applied between the corresponding data qubits of two logical qubits. These physical CNOT gates propagate errors between the data qubits in a deterministic way: X errors on the control qubit are copied to the target qubit and Z errors on the target qubit are copied to the control qubit (see Extended Data Fig. 4b). As a result, the syndrome of a particular logical qubit can contain information about the errors that have occurred on another logical qubit, at the point in time in which the pair underwent a transversal CNOT operation. We can use the information about these correlations and improve the circuit fidelity by jointly decoding the logical qubits involved in the algorithm. We note that this is closely related to other recent developments in decoding entire circuits, or so-called space-time decoding[76–79]. It is also related to Steane error correction[80], for which errors are intentionally propagated from a data logical qubit onto an ancilla logical qubit, which is then projectively measured to extract the syndrome of the data logical qubit.

To perform correlated decoding, we solve the problem of finding the most likely error given the measured syndrome. We start by constructing a decoding hypergraph based on a description of the logical algorithm, which describes how each physical error mechanism (for example, a Pauli-error channel after a two-qubit gate) propagates onto the measured stabilizers[76,81]. The hypergraph vertices correspond to the stabilizer measurement results. Each edge or hyperedge corresponds to a physical error mechanism that affects the stabilizers it connects, with an edge weight related to the probability of that error. Each hyperedge can connect stabilizers both within and between logical qubit blocks (see Fig. 2b). We then run a decoding algorithm that uses this hypergraph, along with each experimental snapshot, to find the most likely physical error consistent with the measurements. This correction is then applied in software (with the exception of Fig. 4e, which is decoded in real time).

Concretely, to construct the hypergraph for a given logical circuit, we perform the following procedure. For each logical algorithm (in this section, considering only Clifford gates), we identify a set of $N$ detectors (vertices of the hypergraph) $D_i \in \{0, 1\}$ for $i = 1,...,N$, which are sensitive to physical errors occurring during the logical circuit. A detector is either on (1) or off (0) to indicate the presence of an error. For the general case, we let $D_i = 0$ if the $i$th stabilizer measurement matches the measurement of its backwards-propagated Pauli operator at a previous time and 1 otherwise (the latter indicates that an error has occurred). In particular, for our surface-code experiments, detectors in the final projective measurement are computed by comparing the final projective measurement of the stabilizers with the value of the ancilla-based stabilizer measurement that occurred before the CNOT (note that, owing to our state-preparation procedure, the initial stabilizer measurement is randomly ±1, but the detector is deterministically zero in the absence of noise). For our 2D colour-code experiments, the initial stabilizers are deterministically +1, so each detector is equal to zero if the corresponding stabilizer in the final projective measurement is +1. To construct the concrete hypergraph and hyperedge weights, we then use Stim[76] to identify the probability $p_j$ ($j = 1,..., M$) of each error mechanism $E_j$ in the circuit using a Pauli-channel noise model with approximate experimental error rates, along with the detectors that are affected by $E_j$.

To find the most likely physical error, we encode it as the optimal solution of a mixed-integer program, a canonical problem in optimization with commercial solvers readily available[82], similar to previous work in ref. 83. We associate each error mechanism $E_j$ with a binary variable that is equal to 1 if that error occurred and 0 otherwise. Our goal is then to find the error assignment $\{0, 1\}^M$ with maximum total error probability (alternatively, the error with the minimum total weight, in which the weight of error $i$ is $w_i = \log[(1 − p_i)/p_i]$), subject to the constraint that the error is consistent with the measured detectors. To be consistent with the measured detectors, the parity of the error variables for all the hyperedges connected to a given detector should match the parity of that detector. Concretely, let $f$ be a map from each detector $D_i$ to the subset of error mechanisms that flip its parity. The most likely error is then the optimal solution to the following mixed-integer program:

$$\text{Maximize} \quad \sum_{j=1}^{M} \log(p_j)E_j + \log(1 − p_j)(1 − E_j)$$

$$\text{subject to} \sum_{E_j \in f(D_i)} E_j − 2K_i = D_i \qquad \forall i = 1, ..., N$$

$$E_j \in \{0, 1\} \qquad \forall j = 1, ..., M$$

$$K_i \in \mathbb{Z}_{\geq 0} \qquad \forall i = 1, ..., N$$

The objective function evaluates to the logarithm of the probability of the assigned error configuration, and each variable $K_i$ ensures that the sum of the error variables in $f(D_i)$ matches $D_i$, modulo 2. Finally, we solve the mixed-integer program to optimality using Gurobi, a state-of-the-art solver[82], and apply the correction string associated with the error indices $j$ for which $E_j = 1$ in the optimal assignment. We explore this correlated decoding in more detail, including its consequences on error-corrected circuits and the asymptotic runtimes of different decoders (M.C. et al., manuscript in preparation). See sections 'Surface code and its implementation' and 'Correlated decoding in the surface code' for further discussion on the surface code in particular.

### Direct fidelity estimation and tomography

One challenge with logical qubit circuits is that convenient probes that are accessible with physical qubits may no longer be accessible. The GHZ state studied here provides such an example, as conventional parity-oscillation measurements cannot be performed[84]. Instead, we use a technique known as direct fidelity estimation[39], which can be understood as follows. The target state $\psi$ is the simultaneous eigenstate of the $N$ stabilizer generators $\{S_i\}$ and, so, the projector onto the target state is $|\psi\rangle\langle\psi| = \prod_i^N (S_i + 1)/2$ (which is 1 if $S_i = 1 \forall i$ and 0 otherwise). Thereby, we can directly measure fidelity by measuring the expectation values of all terms in this product, which—in other words—refers to measuring the expectation values of all elements of the stabilizer group given by the exponentially many products of all the $S_i$. The logical GHZ fidelity is defined as the average expectation value of all measured elements of the stabilizer group. With our four-qubit GHZ state, with four stabilizer generators {XXXX, ZZII, IZZI, IIZZ}, the 16-element stabilizer group is given by all possible products: {IIII, ZZII, IZZI, IIZZ, ZIIZ, IZIZ, ZIZI, ZZZZ, XXXX, XYYX, YXXY, XXYY, YYXX, YXYX, XYXY, YYYY}. We measure the expectation values of all 16 of these operators; for each element, we simply rotate each logical qubit into the appropriate logical basis and then calculate the average parity of the four logical qubits in this measurement configuration. We then directly average all 16 elements equally (with appropriate signs, as some of the stabilizer products should have −1 values) and, in this way, compute the logical GHZ state fidelity. This is an exact measurement of the logical state fidelity[39]. Scaling to larger states can be achieved by measuring elements of the stabilizer group at random[39]. To perform full tomography in Fig. 3e, we measure in all $3^4 = 81$ bases, thereby measuring the expectation values of all 256 logical Pauli strings, and reconstruct the density matrix by solving the system of equations with optimization methods.

### Sliding-scale error detection

Here we provide more information about the sliding-scale error-detection protocol applied for Figs. 3, 5 and 6. Typically, error detection refers to discarding (or postselecting) measurements in which any stabilizer errors occurred. In the context of an algorithm, however, discarding the result of an entire algorithm if just one physical qubit error occurred may be too wasteful and we may want to only discard measurements in which many physical qubits fail and the probability of algorithm success is greatly reduced. For this reason, for the algorithms here, we explore error detection on a sliding scale, for which we can set a desired 'confidence threshold' such that, on the basis of the syndrome outcomes, we determine whether to accept a given measurement. Sliding this confidence threshold enables a continuous trade-off (in data analysis) between the fidelity of the algorithm and the acceptance probability. When sliding-scale error detection is applied, in all applicable cases, we also apply error correction to return to the codespace.

We apply such a sliding-scale error detection for the colour-code logical GHZ fidelity measurements in Fig. 3d. One possible method would be to discard measurements based on the number of detected stabilizer errors. However, this is suboptimal, both because on the colour code a single physical qubit error can result from anywhere between 1 and 3 stabilizer errors and also because errors deterministically propagate between codes during the transversal CNOT gates, such that a single physical error on one code can lead to detected errors on all codes, but which are still all correctable errors. As such, we perform the sliding-scale error detection using the correlated decoding technique and set the confidence threshold as a threshold weight of the overall correction weight on the decoding hypergraph. For example, in the colour code GHZ experiment, a stabilizer error on all four logical qubits that is just consistent with a single physical qubit error that propagated to all four logical qubits is in fact a low-weight (or high-probability) error, as it corresponds to just a single physical qubit error. If the weight of hypergraph correction (inversely related to the log of the probability that a given error mechanism would have occurred leading to the observed syndrome outcome) is below the cut-off threshold weight, then the measurement is accepted; otherwise, it is rejected. For each threshold, we then calculate the average algorithm result ($y$ axis of Fig. 3d), as well as the fraction of accepted data ($x$ axis of Fig. 3d).

In Fig. 5 with [[8,3,2]] codes, for 3, 6, 24 and 48 logical qubits, we apply our sliding-scale detection simply as given by the total number of stabilizer errors detected, although—as illustrated above—this can probably be improved by considering which stabilizer error patterns are more likely to cause an algorithmic failure. For the 12 logical qubits, to have a more fine-grained sliding scale, for each of the $2^4 = 16$ possible stabilizer outcomes, we calculate the XEB to rank the likelihood that each of the observed stabilizer outcomes leads to an algorithmic failure and then use this ranking when deciding whether a given measurement is above/below the cut-off threshold. In Fig. 6b, we set the threshold by the number of stabilizer errors and in Fig. 6d, to have more fine-grained sliding-scale information, we take different subsets of stabilizer outcome events that are all below the threshold of the allowed number of stabilizer errors and calculate the $y$ axis (Pauli expectation value) and $x$ axis (purity) for all of them. Broadly, there are many ways to perform this sliding-scale error detection, and this can be useful both as continuous trade-offs between fidelity and acceptance probability, as well as for use in techniques such as zero-noise extrapolation in data analysis (Fig. 6d).

### Overview of QEC methods

Here we provide a brief overview of key QEC methods used in our work.

**Code distance, decoding and thresholds.** [[$n,k,d$]] notation describes a code with several physical qubits $n$, several logical qubits $k$ and a code distance $d$. The code distance $d$ sets how many errors a code can detect or correct. The code distance is the minimum Hamming distance between valid codewords (logical states), that is, the weight of the smallest logical operator[85]. In the case of the 2D surface and colour codes studied here, $d$ is equivalent to the linear dimension of the system[24].

Following this definition, quantum codes of distance $d$ can detect any arbitrary error of weight up to $d − 1$. Such errors cause stabilizer violations, indicating that errors occurred. Postselecting on the results with no such stabilizer violations corresponds to performing error detection, which protects the quantum information up to $d − 1$ errors at the cost of postselection overhead. Conversely, codes can correct fewer errors than they detect (but without any postselection overhead). The correction procedure brings the system back to the closest logical state (codeword); thus, if more than $d/2$ errors occur, the resulting state may be closer to a codeword different from the initial one, resulting in a logical error[85]. For this reason, codes of distance $d$ can correct any arbitrary error of weight up to $(d − 1)/2$ (rounded down if $d$ is even[24]). The process of decoding refers to analysing the observed pattern of errors and determining what correction to apply to return to the original code state and undo the physical errors created. In many cases, such as with the 2D surface and colour codes, one does not need to apply the correction in hardware (physically flipping the qubits); instead, it is sufficient to undo an unintended $X_L/Z_L$ operator that was applied by

hardware errors by simply applying a 'software' $X_L/Z_L$ operator[24], also described as Pauli frame tracking[86].

As the size of an error correcting code and the corresponding code distance is increased, so are the opportunities for errors to occur as the number of physical qubits increases. This leads to a threshold behaviour in QEC: if the density of errors $p$ is above a (possibly circuit-dependent) characteristic error rate $p_{th}$, then increasing code distance will worsen performance. However, if $p < p_{th}$, then increasing code distance will improve performance[24]. Theoretically, because we require $(d+1)/2$ errors to create a logical error, the logical error rate will be exponentially suppressed as $\propto (p/p_{th})^{(d+1)/2}$ at sufficiently low error rates[24]. The performance improvement with increasing code distance, observed for the preparation and entangling operation in Fig. 2, implies that we surpass the threshold of this circuit. We note that, in this regime, improving fidelities by, for example, a factor of 2× can then lead to an error reduction of $2^4 = 16×$ for the distance-7 code studied and further exponential suppression with increasing code distance. This rapid suppression of errors with reduced error rate and increased code distance is the theoretical basis for realizing large-scale computation. We emphasize that thresholds can be circuit-dependent, as discussed in detail in the surface-code section below.

**Fault tolerance and transversal gates.** A common definition of fault tolerance in quantum circuits[85] (which we use in this work) is that a weight-1 error (that is, an error affecting one physical qubit) cannot propagate into a weight-2 error (now affecting two physical qubits) within a logical block. This property implies that errors cannot spread within a logical block and thereby prevents a single error from growing uncontrollably and causing a logical error.

Distance-3 codes, which are of notable historical importance[3,87], can correct any weight-1 error. Fault tolerance is particularly important for these codes because otherwise a weight-1 error can lead to a weight-2 error and thereby cause a logical fault. An important characteristic of a fault-tolerant circuit that uses distance-3 codes[85] is that (in the low-error-rate regime) physical errors of probability $p$ lead to logical errors with probability $\propto p^2$. We emphasize that the notion of fault tolerance refers to circuit structuring to control propagation of errors, but a circuit can be fault-tolerant with low fidelity or non-fault-tolerant with high fidelity. For example, even if a weight-1 error can lead to a weight-2 error but the code has high distance, or if this error-propagation sequence is possible but highly unlikely, then this property may not be of practical importance (for this reason, definitions of fault tolerance may vary). In practice, the goal of QEC is to execute specific algorithms with high fidelity, and fault-tolerant structuring of a circuit is one of many tools in the design and execution of high-fidelity logical algorithms.

Transversal gates, defined here as being composed of independent gates on the qubits within the code block (that is, entangling gates are not performed between qubits within the same code block)[42], constitute a direct approach to ensure fault-tolerant structuring of a logical algorithm. Because transversal gates imply performing independent operations on the physical constituents of a code block, errors cannot spread within the block and fault tolerance is guaranteed. In this work, all logical circuits we realize (following the logical state preparation) are fault-tolerant, as all logical operations we perform are transversal. Note, in particular, that even though the transversal CNOT allows errors to propagate between code blocks, this is still fault-tolerant, as it does not lead to a higher-weight error within the block and, thereby, a single physical error can neither lead to a logical failure nor an algorithmic failure. Notably, the large family of codes referred to as Calderbank–Shor–Steane (CSS) codes all have a transversal CNOT (ref. 2), all of which can be implemented with the single-step, parallel-transport approach here.

Although all the logical circuits we implement are fault-tolerant, the logical qubit state preparation is fault-tolerant for our $d = 3$

colour code (Figs. 3 and 4) and $d = 3$ surface code (part of Fig. 2), but is non-fault-tolerant for the state preparation of our $d = 5, 7$ surface codes and [[8,3,2]] codes. Thus, all of our experiments with the $d = 3$ colour codes are fault-tolerant from beginning to end, and so the entire algorithm is fault-tolerant and theoretically has a failure probability that scales as $p^2$. However, we note that having a fault-tolerant algorithm also does not imply that errors do not build up during execution of the circuit. For this reason, deep circuits require repetitive error correction[6,88] to constantly remove errors and continuously benefit from, for example, the $p^2$ suppression.

Our logical GHZ state theoretically has a failure probability scaling as $p^2$. Nevertheless, the error build-up (increasing $p$) during the operations of the circuit and the spreading of errors through transversal gates limits our logical GHZ fidelity to 72%. This is consistent with numerical modelling. Similar to the surface-code modelling (Extended Data Fig. 4), we use empirical error rates consistent with 99.4% two-qubit gate fidelity, as well as roughly 4% data qubit decoherence error (including SPAM) over the entire circuit. We simulate the experimental circuit (including the fault-tolerant state preparation with the ancilla logical flag) and measurements of all 16 elements of the stabilizer group (see the 'Direct fidelity estimation and tomography' section), and extract a simulated logical GHZ fidelity of 79%. This is slightly higher than our measured 72% logical GHZ fidelity, possibly originating from imperfect experimental calibration. This modelling indicates that our logical GHZ fidelity is limited by residual physical errors, which will be reduced quadratically as $p^2$ with reduction in physical error rate $p$, in particular by reducing residual single-qubit errors, which were larger during this measurement and are dominating the error budget here.

**Surface code and its implementation.** In 2D planar architectures, such as those associated with superconducting qubits[6,88], stabilizer measurement is the most important building block of error-corrected circuits[24]. In such systems, stabilizers need to be constantly measured to correct qubit dephasing and increase coherence time, as demonstrated recently[6]. Logic operations are implemented by changing stabilizer measurement patterns, enabling realization of techniques such as braiding[24] and lattice surgery[89]. Similar techniques can be used to move logical degrees of freedom to implement nonlocal logical gates[23]. Owing to this gate-execution strategy, $d$ rounds of stabilizer measurement are required for each entangling gate for ensuring fault tolerance[24].

Neutral-atom quantum computers feature different challenges and opportunities. Specifically, they feature long qubit coherence times ($T_2 > 1s$), which can be further increased to the scale of tens to hundreds of seconds with well-established techniques[72]. By using the storage zone, qubits can be idly and safely stored for long periods without repeated stabilizer measurements. Hence, from a practical perspective, increasing qubit coherence by using a logical encoding does not provide immediate gains in improving quantum algorithms and the gains will be from improving the fidelity of entangling operations. Moreover, logic gates and qubit movement do not have to be performed with stabilizer measurements. Instead, they can be executed with nonlocal atom transport and transversal gates. Because such transversal gates are intrinsically fault-tolerant, they do not necessarily require $d$ rounds of correction after each operation. Even syndrome measurement may be better executed in certain cases by techniques such as Steane error correction[80] (similar to our ancilla logical flag with colour codes as used in Fig. 3), as opposed to repeated stabilizer measurement. For these reasons, the transversal CNOT is among the most important building blocks in error-corrected circuits. Hence, we focus here on improving the transversal CNOT by scaling code distance.

Specifically, we use the so-called rotated surface code[6], which has code parameters $[[d^2,1,d]]$. Our distance-7 surface codes (as drawn in Fig. 2d) are composed of 49 physical data qubits, with 24 X stabilizers (light-blue squares) and 24 Z stabilizers (dark-blue squares), and one

encoded logical qubit described by anticommuting weight-7 operators, the horizontally oriented $X_L$ and the vertically oriented $Z_L$. The X and Z stabilizers commute with the $X_L$ and $Z_L$ logical operators, allowing the measurement of the stabilizers without disturbing the underlying logical degrees of freedom. In our experiments, we prepare one surface code in $|+_L\rangle$ and one surface code in $|0_L\rangle$. In the first code, this is realized by preparing all physical data qubits in $|+\rangle$, thereby preparing an eigenstate of $X_L$ and the 24 X stabilizers, and then projectively measuring the 24 Z stabilizers with 24 ancilla qubits (Fig. 2d red dots) using four entangling-gate pulses[24]. The second code is prepared similarly but with all physical qubits initialized in $|0\rangle$, thus preparing an eigenstate of $Z_L$ and the 24 Z stabilizers, and then projectively measuring the 24 X stabilizers with 24 ancillas. The CNOT is directly transversal because these two surface-code blocks have the same orientation and does not require rotation of the lattice to implement a H. The projective measurement of the ancillas defines the values of the stabilizers. During the transversal CNOT, the values of the stabilizers are copied onto the other code as well and is tracked in software.

Because we only perform a single round of stabilizer measurement, our state-preparation scheme is nFT for the $d = 5, 7$ codes. Consider, for instance, the case when all stabilizers are defined as +1 and no errors are present in the system, but an ancilla measurement error in the middle of the surface-code lattice yields a stabilizer measurement of −1. Correction then causes a large-weight pairing of this apparent stabilizer violation to the boundary[4]. Hence, this single ancilla measurement error can lead to several data qubit errors, resulting in nFT operation. The $d = 3$ code initialization is a special case that does not suffer from this issue[38]. Higher-order considerations about fault tolerance given by gate ordering during stabilizer measurement can also be considered[6].

The effect of these nFT errors from noisy syndrome extraction is to cause X physical errors on the $|+_L\rangle$ state and Z physical errors on the $|0_L\rangle$ state. Thus, in performing just a SPAM measurement, the presence of these errors would not be directly apparent, as these errors commute with measuring the $|+_L\rangle$ in the X basis and $|0_L\rangle$ in the Z basis. As such, this circuit would not be a good benchmark of surface-code state preparation. Conversely, the transversal CNOT experiment is sensitive to the various aspects of the circuit and a good indication of performance. Because we measure the Bell state in both the $X_L^1 X_L^2$ and $Z_L^1 Z_L^2$ bases, the nFT errors in both bases will propagate through the logical CNOT and cause errors on both logical qubits in both the X and Z bases. For these reasons, unlike a surface-code SPAM measurement, this experiment is a good indication of logical performance. In fact, the effect of these nFT errors is such that, if we just apply conventional decoding within each logical block, then we find that the Bell state degrades substantially with increased code distance (Fig. 2d).

The effects of this nFT preparation are suppressed (but not entirely removed) by using the correlated decoding technique. For example, consider a nFT-induced apparent stabilizer violation to the left of the middle line in the lattice of the $d = 7$ $|+_L\rangle$ state, corresponding to a chain of three physical X errors to the boundary. These errors will propagate through the logical CNOT onto the second logical qubit and affect the independent measurement of both logical qubits in the Z basis when investigating the $Z_L^1 Z_L^2$ stabilizer. When decoded independently, if another single X error occurs on the first block after the CNOT moving the stabilizer violation to the right of the middle line, becoming a chain of four X physical errors, this will cause an incorrect pairing and lead to an independent $X_L^1$ error on this code only and thereby corrupt the $Z_L^1 Z_L^2$ stabilizer and would correspond to a total weight-6 correction between the two codes. However, when decoded jointly with correlated decoding, these errors can be effectively decoded, as they will appear on the stabilizers of both logical qubits. In this example, the lowest-weight pairing would remove this chain of three X errors from both codes and leave only the one remaining X error on the first block, which can also be decoded successfully (the total pairing weight here

is only 2). Our correlated decoding technique is thus essential to our observation of improved Bell performance with code distance.

Finally, we elaborate on our evaluation of Bell-pair error. Bell-state fidelity is given by the average of the populations and the coherences, which—for physical qubits—can be measured as the ZZ populations and the amplitude of parity oscillations. In the language of stabilizers, the parity oscillation amplitude is given by the average of $\langle XX \rangle$ and $-\langle YY \rangle$ (ref. 90). With the surface code, we cannot conveniently measure the $Y_L$ operators fault-tolerantly (and that is why we use colour codes for programmable Clifford algorithms and full tomography; see next section). For this reason, we estimate the logical coherences as $\langle X_L X_L \rangle$, which we then average with the populations for calculating the Bell-pair error. To support the validity of this analysis, we can instead calculate a lower bound on the Bell-state fidelity[90], which also shows the same improvement in performance as we increase code distance (Extended Data Fig. 4d).

**Correlated decoding in the surface code.** Following the above discussion, we provide more insights related to the correlated decoding in the case of the surface-code transversal CNOT. Consider a circuit in which perfect (noiseless) surface codes are initialized, a transversal CNOT is executed and then projective readout is performed. If errors occur before the transversal CNOT, then these errors can propagate; for example, an X physical error on the control logical qubit will propagate onto the target logical qubit and thereby double the density of errors on the target logical qubit. By multiplying the projectively measured Z stabilizers of the target logical qubit with those of the control logical qubit, the propagation is undone. Now the target logical qubit only has to decode its original density of X errors. The same considerations can be made for Z errors originating on the target logical qubit that propagate onto the control logical qubit. However, if there are errors after the transversal CNOT, then multiplying the stabilizers instead doubles the density of such errors. Thus, the optimal decoding strategy if errors are only after the transversal CNOT is to perform independent matching within both codes. The general case in which errors are present both before and after the transversal CNOT corresponds to neither case and is modelled by our decoding hypergraph that has edges and hyperedges connecting the two logical qubits, with edge weights informed by our experimental error model. Extended Data Fig. 5 explores decoding performance with different values of the scaled weights of the edges and hyperedges that connect the stabilizers of the two logical qubits. These results illustrate that the correlated decoding is robust (but not completely insensitive) to the nFT errors associated with ancilla measurement errors. This feature would also be recovered by the simpler multiplication decoder, which would be entirely insensitive to errors from ancilla measurement, but is—however—more sensitive to errors after the CNOT. Specifically, Extended Data Fig. 5c shows that our optimized decoder is not simply a 'multiplication decoder', as the ancilla measurement values indeed contribute to the correction procedure and make the correlated decoding more robust to decoder parameters. For a given logical circuit, our correlated decoding procedure generates a decoding hypergraph, which we then solve using most likely error methods, which is done here for both surface-code and colour-code experiments, and can generically be applied to any stabilizer codes and Clifford circuits[79]. More theoretical details and discussion of correlated decoding will be presented in M.C. et al, manuscript in preparation.

**2D colour codes.** 2D colour codes are topological codes that are similar to surface codes[91]. Often portrayed in a triangular geometry, the colour codes used here are a tiling of three colours of weight-4 and weight-6 stabilizers, with $X_L$ and $Z_L$ operator strings running along the boundary of the code[91]. In this work, we study 2D $d = 3$ colour codes, as portrayed in Fig. 3a, which only contain weight-4 stabilizers given by the products of X and Z on the qubits of each coloured plaquette. This $d = 3$ colour code is identical to the seven-qubit Steane code.

However, we emphasize that the techniques used here directly apply to larger-distance colour codes[92].

Although the colour codes are similar to surface codes, an important difference is that, in the colour code, the X and Z stabilizers lie directly on top of the same qubits (as opposed to being on dual lattices with respect to each other) and, similarly, the $X_L$ and $Z_L$ operators lie on top of each other (as opposed to propagating in the orthogonal directions on the surface code). In other words, the operators here are symmetric and related by a global basis transformation. This has important consequences for the allowed transversal gate set[41,93]. In particular, although the surface code technically has a transversal H that transforms $X_L \leftrightarrow Z_L$, it requires a physical 90° rotation of the code block. Although such lattice rotations are possible using atom-motion techniques, for many circuits, it is inconvenient. Conversely, in the colour code, H is transversal: it directly exchanges $X_L \leftrightarrow Z_L$ as well as the X and Z stabilizers. This difference is even more important for the transversal S gate, which is possible for the colour code. Here transversal S exchanges $X_L \leftrightarrow Y_L$ (for which $Y_L$ is given by the product of $X_L$ and $Z_L$, which lie on top of each other) as intended, and the X stabilizer of a given plaquette returns to itself by multiplying the Z stabilizer of that same plaquette. (This is in contrast to the surface code, which does not have a transversal S, for which the $Y_L$ operator is a product of horizontally propagating $X_L$ and vertically propagating $Z_L$ (ref. 24)). Because the colour code has the entire transversal gate set of {H, S, CNOT} and also does not require tracking any lattice rotations, it is well suited to exploration of programmable logical Clifford algorithms.

For fault-tolerant preparation of the $d = 3$ colour code, we use a modified version of the scheme summarized in ref. 38, in which, instead of the eight-gate encoding circuit, we use a nine-gate encoding circuit that is more conveniently mapped to specific atom movements in our system (corresponding to graph-state preparation similar to ref. 7), followed by a transversal CNOT with an ancilla logical flag. The logical SPAM fidelity is then calculated as the probability of observing $|0_L\rangle$ after decoding. We note that, in Fig. 3, we could also have made a five-qubit GHZ state but instead made a four-qubit GHZ state for simplicity of performing full tomography. In Fig. 4, when Bell-state fidelities with feedforward are reported, we estimate the logical coherences as the average of $\langle X_L X_L \rangle$ and $-\langle Y_L Y_L \rangle$, which we then average with the $Z_L Z_L$ populations (not plotted) for calculating the Bell-pair fidelity. Finally, we note that the feedforward Bell state in Fig. 4e could also be performed with a software $Z_L$ rotation on either of the two qubits, allowing correction to the appropriate Bell state, but here we perform the feedforward S on both qubits to test our feedforward capabilities; this technique is directly compatible with performing magic-state teleportation[24].

**Clifford and non-Clifford gates and universality.** 2D topological codes such as the surface and colour codes have transversal implementation of Clifford gates (for example, {H, S, CNOT}). This gate set is not universal, that is, it cannot alone be used to realize an arbitrary quantum computation and requires a non-Clifford gate such as {T, CCZ} for achieving universal computation. Moreover, circuits composed solely of stabilizer states and Clifford gates can be simulated in polynomial time because of the Gottesman–Knill theorem[44]. This can be understood as stabilizer tracking; for example, consider a three-qubit system in which a stabilizer of the state is $X \otimes I \otimes I$, such that X stabilizes the $|+\rangle$ state and I is the identity. Applying two CZ entangling gates $CZ_{1,2} \otimes CZ_{1,3}$ transforms this stabilizer to $X \otimes Z \otimes Z$ because an X flip before the CZ simply changes whether a Z flip will be applied to the other qubits. Even though Clifford circuits create superposition and entanglement between qubits, the $N$ initial stabilizers of the state can simply be tracked as they propagate through the circuit (so-called operator spreading[94]) and thereby simulation of the circuit can be easily accomplished.

The effect of non-Clifford gates, however, is far more complex. For example, passing the stabilizer $X \otimes I \otimes I$ through a CCZ maps into a superposition of Pauli strings, that is, $X \otimes I \otimes I \to 1/2(X \otimes I \otimes I + X \otimes Z \otimes I + X \otimes I \otimes Z - X \otimes Z \otimes Z)$, as an X flip now changes whether a CZ operator will be applied on the other qubits, resulting in four times more operators to track after the single CCZ. (The CZ operator matrix is simply equal to $1/2[I \otimes I + Z \otimes I + I \otimes Z - Z \otimes Z]$). This causes not only operator spreading but also so-called operator entanglement[94]. As we apply further non-Clifford gates, the number of operators to track will grow exponentially and eventually will become computationally intractable. For example, state-of-the-art Clifford + T simulators can handle roughly 16 CCZ gates[50]. This is the basis behind our complex sampling circuits, in which the 48 CCZs on the 48 logical qubits create a high degree of scrambling and magic (defined below), rendering Clifford + T simulation impractical.

### [[8,3,2]] circuit implementation

Here we provide more detail about our [[8,3,2]] circuit implementations. The [[8,3,2]] code blocks are initialized in the $|-_L, +_L, -_L\rangle$ state with the circuit in Extended Data Fig. 6, which can be understood as preparing two four-qubit GHZ states (corresponding to [[4,2,2]] codes[95]), that is, $GHZ_Z^{1,3,5,7} \otimes GHZ_X^{2,4,6,8}$, and subsequently entangling them as illustrated in Extended Data Fig. 6a (as well as applying Z gates). In our circuit implementations, for system sizes of 3–24 logical qubits for both sampling and two-copy measurements, we prepare eight blocks encoded over 64 physical qubits. For the 48-logical-qubit circuit (128 physical qubits in total), we encode eight blocks and entangle them, and then drop them into storage; then, we pick up 64 new physical qubits from storage, encode them into eight blocks in the entangling zone and entangle them. Finally, we bring the original eight blocks from storage and entangle them with the second group of eight blocks in the entangling zone (Extended Data Fig. 6) (see Supplementary Video).

The transversal gate set of the [[8,3,2]] code is enabled as follows (see also refs. 16,17,26,27). The transversal CNOT between blocks immediately follows from the fact that the [[8,3,2]] code is a CSS code. In-block CZ gates between two logical qubits $L_i$ and $L_j$ ($CZ_{Li,Lj}$) can be realized by S, $S^\dagger$ gates on the face corresponding to logical qubit $L_k$. For example, consider applying the pattern of S, $S^\dagger$ gates to the top face in Fig. 5, that is, $S_1 S_3^\dagger S_5^\dagger S_7$, which transforms $X_{L1} = X_1 X_2 X_3 X_4$ to $X'_{L1} = -Y_1 X_2 Y_3 X_4$, which is equal to $X'_{L1} = X_{L1} Z_{L2}$, and the same applies to give $X'_{L2} = X_{L2} Z_{L1}$, that is, a CZ is realized between logical qubits 1 and 2. This procedure can also be used to understand why the pattern of T, $T^\dagger$ realizes a CCZ between the three encoded qubits. CCZ gates should map $X_{L3}$ to $X_{L3} \otimes CZ_{L1,L2}$. By applying the pattern of T, $T^\dagger$ in Fig. 5a, each X face maps to itself multiplied by a pattern of S, $S^\dagger$, for example, $X_{L3} = X_1 X_3 X_5 X_7$ maps to $X'_{L3} = X_{L3} S_1 S_3^\dagger S_5^\dagger S_7$, or then $X'_{L3} = X_{L3} \otimes CZ_{L2,L3}$. This happens for all three $X_L$ faces, thereby realizing a CCZ gate. Finally, we detail the permutation CNOT, which was also developed in ref. 27. Physically permuting atoms to swap qubits $4 \leftrightarrow 8$ and $3 \leftrightarrow 7$ takes $X_{L1} = X_1 X_2 X_3 X_4$ to $X'_{L1} = X_1 X_2 X_7 X_8$ or instead $X'_{L1} = X_{L1} X_{L2}$ (also by multiplying the global X stabilizer) and, similarly, it can be seen by tracking the qubit permutations that $Z'_{L2} = Z_{L2} Z_{L1}$, that is, realizing a CNOT. Finally, although these 3D codes do not have a transversal H, because they are CSS codes, they can be initialized and measured in either the X or the Z basis, effectively allowing H gates at the beginning or end of the circuit.

In-block logical entangling gates are applied block by block and any in-block gate combination can be realized. For conceptual simplicity, we apply only two particular local Raman patterns in layers. The first is the gate combination $CCZ_{L1,L2,L3} \cdot CZ_{L1,L2} \cdot CZ_{L1,L3} \cdot CZ_{L2,L3} \cdot Z_{L1} \cdot Z_{L2} \cdot Z_{L3}$, given by applying $T^\dagger$ on the entire physical qubit block, and the second gate combination we apply is $CCZ_{L1,L2,L3} \cdot CZ_{L2,L3} \cdot CZ_{L1,L3} \cdot Z_{L3}$, given by applying T on the top row and $T^\dagger$ on the bottom row. In our circuits, we alternate layers of in-block transversal entangling gates and out-block transversal CNOTs, entangling logical blocks on up to 4D hypercubes[19,96,97] (see Extended Data Fig. 6). We keep the control and target qubits the same throughout the circuit for conceptual

simplicity, allowing the local physical H gates on the target qubits to be compiled with the in-block gate layers, but the control-target direction can also be chosen arbitrarily. We ensure that in-block logical entangling gates are applied such that they do not trivially commute through and cancel with earlier entangling-gate applications. As an experimental note, we remark that, for the Clifford states realized in the other parts of this work, stabilizers take on values of either +1 or −1 (owing to, for example, use of physical $\pi/2$ rotations instead of H), which is then simply redefined in software. Because, for our [[8,3,2]] circuits, we implement non-Cliffords on the physical level, it is important to ensure that all stabilizers are initialized and maintained as +1; for example, if a Z stabilizer is −1, then the logical CCZ implementation sends the X stabilizer expectation value to 0. This can be understood as a physical X on a single site transforming to a superposition $(X + Y)/\sqrt{2}$ by physical T, going into an equal superposition of X stabilizer being +1 and −1.

## Classically hard circuits with [[8,3,2]] codes

Our implemented circuits are equivalent to IQP circuits[98], which gives a theoretical basis for understanding why our circuits could be classically hard to simulate, for which we also provide numerical evidence of so-called anticoncentration[99,100]. IQP circuits are defined as initializing $|+\rangle^{\otimes n}$ on $n$ qubits, applying a diagonal entangling unitary such as those composed by {CCZ, CZ, Z} and then measuring in the X basis[20,98]. A uniform superposition of $2^n$ bitstrings is created, the diagonal gates apply −1 signs in a complicated fashion to the exponentially many bitstrings and then 'undoing the superposition' with the final H before measurement now results in an intricate 'speckle' interference pattern[18]. Sampling from the output distribution of this speckle pattern can be done efficiently on a quantum device that implements the circuit, but is exponentially costly on a classical device for certain choices of IQP circuits[20]. The transversal gate set of the [[8,3,2]] code, as described above, contains diagonal gates {CCZ, CZ, Z} that apply −1 signs to the bitstrings, but is made non-diagonal by the application of CNOTs, which permute bitstrings. Because this bitstring permutation does not break the IQP framework, these circuits are equivalent to an effective IQP circuit, but which is much more complex: for example, circuits with 48 CCZs and 96 CNOTs map to effective IQP circuits with roughly 1,000 CCZ gates. Nevertheless, because IQP circuits are a well-understood framework, we can discuss our circuit properties with this toolset.

We experimentally explore these circuits with the XEB[18], defined as XEB $= 2^{N_L}\Sigma_i p(x_L^i)q(x_L^i) - 1$, in which $N_L$ is the number of logical qubits, $q(x_L^i)$ is the measured probability distribution for our logical qubits and $p(x_L^i)$ is the calculated probability distribution; here we normalize the XEB by its ideal value such that the XEB for the noiseless circuit is 1. In typical cases, if noise overwhelms the circuit, the measured distribution will be uniform[18] and the measured XEB will be 0.

The IQP circuits are a good setting for quantum-advantage-type experiments, as the bitstring distribution of IQP circuits with randomly applied {CCZ, CZ, Z} gates (random degree-3 polynomials) is known to be classically hard to simulate[20,101]. In M.K. et al., manuscript in preparation, we show that the ensemble of random hypercube IQP circuits, whose instances are experimentally explored here, converges to the uniform IQP ensemble as the depth and size of the hypercube is increased. In Extended Data Fig. 8a, we show that hypercube IQP circuits with random in-block operations and randomized control-targets on the out-block CNOT layers (realizing the hypercube) anticoncentrate quickly as the dimension of the hypercube is increased, with XEB eventually reaching the uniform-IQP value of 2. We also find that the presence of non-Clifford CCZ gates, which are critical for the computational hardness here, further improves anticoncentration properties, as we observe that the ideal XEB of experimental circuits approach 2 as well, even without much randomization.

Moreover, the XEB turns out to be a better benchmark for IQP circuits than for generic random circuit sampling settings (such as Haar-random circuits)[102–104]. For IQP, the XEB is close to the many-body fidelity and the difference can be theoretically bounded under reasonable noise assumptions (M.K. et al., manuscript in preparation). Intuitively, this fact is related to the diagonal structure of the IQP circuits, which allows the XEB to capture errors in a manner closer to fidelity, despite being defined only in the computational basis. In other words, a Z error will always corrupt the X-basis measurement, and an X error (except one immediately before measurement) will create new Z errors that also corrupt the X-basis measurement. Thus, in the fully postselected regime, in which errors at the end of the circuit are well described by logical errors, we expect the XEB to be a good measure of fidelity. We further note that, as well as the efficient generation of complex IQP circuits here, the [[8,3,2]] gate set presented here can realize arbitrary IQP circuits composed of {CCZ, CZ, Z} gates[105]. The in-block {CCZ, CZ, Z} operations can be applied to any groupings of qubits by noting that combining the in-block and out-block CNOTs allow us to compose arbitrary transversal SWAP operations of targeted individual logical qubits between different blocks.

## Simulation of bitstring probabilities

To calculate the logical bitstring probabilities necessary for evaluating the XEB and benchmarking our circuits, we use a hybrid simulation approach combining wavefunction and tensor-network[106] methods. It works best only when performing all of the entangling gates of the hypercube a single time and relies on the fact that the final round of CNOTs is immediately followed by a measurement, simplifying network contraction. Concretely, for a $D$-dimensional logical hypercube, the two subsystems consisting of $2^{D-1}$ blocks are simulated independently and then the final layer of CNOTs and in-block operations is combined with the measurement outcomes (the bitstring of interest), which results in a contraction of two $8^{2^{D-1}}$ tensors (see Extended Data Fig. 8b). This is a square-root reduction in the memory requirement compared with the full wavefunction simulation, which uses $O(8^{2^D})$ space. The ideal XEB value is calculated by sampling bitstrings from the ideal output distribution and then averaging the corresponding probabilities. The bitstrings are sampled using a marginal sampling algorithm, which uses the same contraction scheme described above.

We next consider whether the finite XEB scores in this problem can be easily 'spoofed' by foregoing exact simulation of the implemented circuit and using a classical algorithm with fewer resources, similar in spirit to the algorithm introduced in ref. 102. For the circuits studied in Fig. 5, containing only a single layer of gates on the hypercube, there is only a single round of CNOTs connecting the two $2^{D-1}$-block partitions; thus, removing them from the circuit and sampling from the two independent halves might not decrease the XEB substantially while reducing the memory requirement to $8^{2^{D-2}}$. In Extended Data Fig. 8c, we study the performance of this spoofing attack and find that the obtained XEB rapidly decreases, once further gate layers are introduced, for a particular extension of our circuit.

The contraction scheme above, used for both the ideal simulation and the XEB spoofing, scales exponentially with the number of qubits. However, the exponent is substantially reduced by using the fact that the hypercube circuits can be naturally partitioned into smaller blocks, with only a single inter-partition layer of CNOTs at the end of the circuit. This simulation method therefore becomes less efficient if we introduce extra CNOT layers (within a single partition) after the inter-partition layer, as we estimate in Fig. 5d. Applying $l = \{0,...,D-1\}$ further intra-partition CNOT layers forces the CNOT tensors in Extended Data Fig. 8b to be blocked into groups of $2^l$, which results in the execution time to scale roughly as $O(8^{2^l}/2^l)$, in which the numerator comes from the tensor contraction complexity and the denominator accounts for the reduced number of contractions resulting from blocking. The explicit times quoted in Fig. 5d as a function of extra CNOT layers are

based on the above matrix-multiplication estimate and fitted such that the depth-1 hypercube time matches 1.44 s, which corresponds to our implementation. In practice, the actual runtimes might differ owing to hardware and software optimization and other factors, such as the cost of tensor permutations; however, we expect the general trend to hold. Finally, if the $2^l$-blocked tensors were to be stored directly, the memory requirement of this approach would grow as $8^{2^{l+1}}$, recovering the full $8^{2^D}$ memory complexity for $l = D - 1$.

In this work, we use these circuits and XEB results for benchmarking our logical encoding, which requires the ability to simulate these circuits. Future logical-algorithm experimentation can explore quantum-advantage[18,48,107–109] tests with encoded qubits, as will be detailed in M.K. et al., manuscript in preparation.

### Physical qubit circuit implementations

To compare our logical-qubit algorithms with analogous circuits on physical qubits, we work out a concrete implementation of our sampling/scrambling circuits on physical qubits using the same physical gate set, Clifford + T, as used in the logical circuit, which we also then attempt to realize experimentally. We replace each [[8,3,2]] block with a three-physical-qubit block, decomposing the 'in-block' CCZ gates into six CNOTs and seven {T, T†} gates and implement 'transversal' CNOTs directly between the three-qubit blocks. We note that the CZ can be compiled into the CCZ implementation, but this has a minor effect on our analysis and estimates. These physical circuits are complex: 48 qubits with 48 CCZs and 228 two-qubit gates (as realized with our logical qubits) decomposes into an effective 516 two-qubit gates (384 if the CZ gates are compiled into the CCZs). In trying to implement these circuits in practice, the build-up of coherent errors resulted in a vanishing XEB for our physical circuits. These experiments made it clear that the logical-circuit equivalent was greatly outperforming the physical circuit, thereby providing direct evidence that our logical algorithm outperforms our physical algorithm for this specific sampling circuit.

More quantitatively, with a concrete physical implementation, we calculate an upper bound by assuming optimistic performance. We assume our best-measured fidelities: SPAM of 99.4% (ref. 8), local single-qubit gate fidelity of 99.91% (Extended Data Fig. 2), two-qubit gate fidelity of 99.55% (ref. 8) and $T_2 = 2s$. We then count the total number of entangling-gate pulses for the CZ gates, the total number of compiled local single-qubit gates and the estimated circuit duration, and use these to calculate the estimate presented in Fig. 5f. We further confirm this analysis for small-scale circuit implementations. For a short three-qubit circuit, we benchmark the XEB for the physical circuit as approximately 0.87, which is below the estimated three-qubit upper bound of roughly 0.92. We note that, in Fig. 5f, we plot estimates of physical-qubit fidelity and not the XEB, but we expect the XEB and fidelity to be closely related, as discussed previously.

We note several observations made in comparing physical and logical implementations of these complex circuits. First, empirically, it seems that the logical circuit is much more tolerant to coherent errors[49,110,111], and understanding the manifestations of this is a subject of continuing investigation. Specifically, it seems that the logical circuit realizes inherently digital operation, for which the small coherent errors do not substantially shift/distort the bitstring distribution but just reduce the overall fidelity[49,110] (see, for example, the agreement in Extended Data Fig. 7a). This is in contrast to the physical implementation, in which coherent errors are seen to substantially alter the shape of the bitstring distribution, for example, changing relative amplitudes. Second, we note that we optimize our [[8,3,2]] circuits only by optimizing the stabilizer expectation values and not by optimizing the XEB or two-copy result directly. When running complex circuits, the stabilizers serve as useful intermediate fidelity benchmarks, for both optimizing circuit design and ensuring proper execution, especially in regimes in which output distributions or other observables cannot be calculated. Overall,

we find that these complex circuits seem to perform much better with logical qubits than physical qubits.

### Two-copy measurements

A powerful method to extract various quantities of interest are Bell-basis measurements between two copies of the same state[21,22,52]. First, we use these measurements to calculate the purity or entanglement entropy of the resulting state[7,21,52,112,113]. Measuring the occurrences of the singlet state $\frac{|01\rangle - |10\rangle}{\sqrt{2}}$ (|11⟩ outcome for our measurements after applying the final pairwise entangling operations) probes the eigenvalue of the SWAP operator $\hat{s}_i$ at a given pair of sites $i$. This is in turn related to the purity of the state by observing that $\mathrm{Tr}[\rho_A^2] = \mathrm{Tr}[\Pi_{i \in A} \hat{s}_i \rho_A \otimes \rho_A]$ for any subsystem $A$. Thus, the average purity can be estimated by the average parity $\mathrm{Tr}[\rho_A^2] = \langle (-1)^{\text{no. of observed singlets}} \rangle$ within $A$ and, thus, also the second-order Rényi entanglement entropy $S_2(A) = -\log_2 \mathrm{Tr}[\rho_A^2]$.

The entanglement entropy calculation only involves the singlet outcomes. By making use of the full outcome distribution, we can also evaluate the absolute value of all $4^N$ Pauli strings, from a single dataset, in which $N$ is the number of qubits involved in each copy of the state[22]. More concretely, consider a given Pauli string $O = \prod_i P_i$, in which $P_i \in \{X_i, Y_i, Z_i, I_i\}$ are individual Pauli operators on site $i$ (and the identity), and a given observed bitstring $\{\vec{a}, \vec{b}\}$, in which $\vec{a}, \vec{b}$ label the outcomes in the control and target copy. The rules of reconstructing the Pauli strings through these Bell-basis bitstrings can be worked out through considering the computational states to which the Bell states are mapped and considering which operators of XX, YY and ZZ have +1 or −1 eigenvalue for the various Bell states. We explicitly list the analysis procedure: for Pauli term $X_i$, we assign parity +1 if $a_i = 0$ and −1 otherwise; for Pauli term $Y_i$, we assign parity +1 if $a_i \neq b_i$ and −1 otherwise; for Pauli term $Z_i$, we assign parity +1 if $b_i = 0$ and −1 otherwise; for $I_i$, we assign parity +1 always. The contribution of the bitstring $\{\vec{a}, \vec{b}\}$ to $|\mathrm{tr}(O\rho)|^2$ is then given by the product of the individual parities.

We can perform the same analysis as a function of the amount of error detection applied. As shown in Extended Data Fig. 9a, as more error detection is applied, the distribution of Pauli expectation values that are expected to be zero and non-zero separate apart further. This also provides a natural method to perform error mitigation through zero-noise extrapolation: by performing sliding-scale error detection, we can extract the Pauli expectation value squared for groups of Pauli strings with the same expected value, as a function of the logical purity. We perform a linear fit of the Pauli expectation value squared versus the logical purity and extrapolate to purity $\mathrm{tr}(\rho^2) = 1$, corresponding to the case of zero noise, to estimate the error-mitigated values. The choice of a linear fit is motivated by the fact that both $|\mathrm{tr}(O\rho)|^2$ and $|\mathrm{tr}(\rho^2)|$ scale with power 2 of the density matrix. We expect that more detailed considerations of the noise model, using knowledge about the weight of each operator, as well as whether detected errors in each shot overlap with a given Pauli operator, can further improve the error-mitigation results.

We can also compute measures of distance from stabilizer states, also known as 'magic', using the additive Bell magic measure in ref. 53, which only requires O(1) number of samples and O(N) classical post-processing time. To do so, we randomly sample subsets of four measured Bell-basis bitstrings $\mathbf{r}, \mathbf{r}', \mathbf{q}, \mathbf{q}'$ and calculate their contribution to the Bell magic using the check-commute method of ref. 53: $\mathcal{B} = \sum_{\mathbf{r}, \mathbf{r}', \mathbf{q}, \mathbf{q}' \in \{0,1\}^{2N}} P(\mathbf{r})P(\mathbf{r}')P(\mathbf{q})P(\mathbf{q}') \|[\sigma_{\mathbf{r} \oplus \mathbf{r}'}, \sigma_{\mathbf{q} \oplus \mathbf{q}'}]\|_\infty$, with $\|[\sigma_{\mathbf{r} \oplus \mathbf{r}'}, \sigma_{\mathbf{q} \oplus \mathbf{q}'}]\|_\infty$ being 0 when the two Pauli strings commute and 2 otherwise. $\mathbf{r} \oplus \mathbf{r}'$ denotes bitwise XOR between the two bitstrings. $P(\mathbf{r})$ is the probability of observing bitstring $\mathbf{r}$. The Pauli string $\sigma_{\mathbf{r}}$ is of length $N$ and the $i^{\text{th}}$ element is I, X, Z or Y when the target and control qubit at site $i$ read 00, 01, 10 or 11, respectively. We convert this result to additive Bell magic through the formula $\mathcal{B}_a = -\log_2(1 - \mathcal{B})$. We use approximately $10^7$ samples to estimate the additive Bell magic for each dataset. The

results for the estimated additive Bell magic as a function of the number of non-Clifford gates applied (circuits shown in Extended Data Fig. 9f) are shown in Fig. 6c. These results also use the purity estimates in the same dataset, which are used for error mitigation as described in equations (13)–(15) of ref. 53. All additive Bell magic data shown are with full error detection applied.

The same experiments we perform here can also be interpreted as a physical Bell-basis measurement. Using this insight, in Extended Data Fig. 9c,d, we show the entanglement entropy for different subsystem sizes, when analysing the data as physical Bell-pair measurements and applying different levels of stabilizer-based postselection. Notably, the full-system parity when postselecting on all stabilizers being correct is identical when analysing the outcomes as either a physical or a logical circuit. This is because, in this limit, the results of the physical-circuit analysis can be viewed as taking the (imperfect) logical state and running a perfect encoding circuit, hence giving identical results.

## Data availability

The data that support the findings of this study are available from the corresponding author on request.

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

**Acknowledgements** We thank A. Kubica for pointing us to the connection between our transversal gate set and IQP circuits, J. Campo, S. Haney, T. Wong, T. T. Wang, P. Stroganov and especially J. Amato-Grill for contributions in the development of the FPGA technology and fast CMOS readout. We gratefully acknowledge useful discussions with B. Braverman, H. Briegel, S. Cantu, S. Choi, J. Cong, M. Devoret, H.-Y. Huang, A. Keesling, H. Levine, A. Lukin, K. V. Kirk, N. Meister, H. Pichler, H. Poulsen, J. Ramette, J. Sinclair, D. Tan and all members of the Lukin group. We acknowledge financial support from the DARPA ONISQ programme (grant number W911NF2010021), the Center for Ultracold Atoms (a NSF Physics Frontier Center), the National Science Foundation, the Army Research Office MURI (grant number W911NF-20-1-0082), IARPA and the Army Research Office, under the Entangled Logical Qubits programme (Cooperative Agreement Number W911NF-23-2-0219) and QuEra Computing. D.B. acknowledges support from the NSF Graduate Research Fellowship Program (grant DGE1745303) and the Fannie and John Hertz Foundation. S.J.E. acknowledges support from the National Defense Science and Engineering Graduate (NDSEG) fellowship. T.M. acknowledges support from the Harvard Quantum Initiative Postdoctoral Fellowship in Science and Engineering. M.C. acknowledges support from the Department of Energy Computational Science Graduate Fellowship under award number DE-SC0020347. D.H. acknowledges support from the U.S. Department of Defense through a QuICS Hartree fellowship. J.P.B.A. acknowledges support from the Generation Q G2 Fellowship and the Ramsay Centre for Western Civilisation. N.M. acknowledges support by the Department of Energy Computational Science Graduate Fellowship under award number DE-SC0021110. I.C. acknowledges support from the Alfred Spector and Rhonda Kost Fellowship of the Hertz Foundation, the Paul and Daisy Soros Fellowship and NDSEG. M.J.G. and D.H. acknowledge support from NSF QLCI (award no. OMA-2120757). The commercial equipment used in this work does not reflect endorsement by the NIST. The views and conclusions contained in this document are those of the authors and should not be interpreted as representing the official policies, either expressed or implied, of IARPA, the Army Research Office, or the US Government. The US Government is authorized to reproduce and distribute reprints for Government purposes notwithstanding any copyright notation herein.

**Author contributions** D.B., S.J.E., A.A.G., S.H.L., H.Z., T.M., S.E. and G.S. contributed to the building of the experimental setup, performed the measurements and analysed the data.

M.C., M.K., D.H., J.P.B.A., N.M., I.C. and X.G. performed theoretical analysis. P.S.R. and T.K. developed the FPGA electronics. All work was supervised by M.J.G., M.G., V.V. and M.D.L. All authors contributed to the logical processor vision, discussed the results and contributed to the manuscript.

**Competing interests** M.G., V.V. and M.D.L. are co-founders and shareholders and H.Z., P.S.R. and T.K. are employees of QuEra Computing.

**Additional information**
**Correspondence and requests for materials** should be addressed to Mikhail D. Lukin.

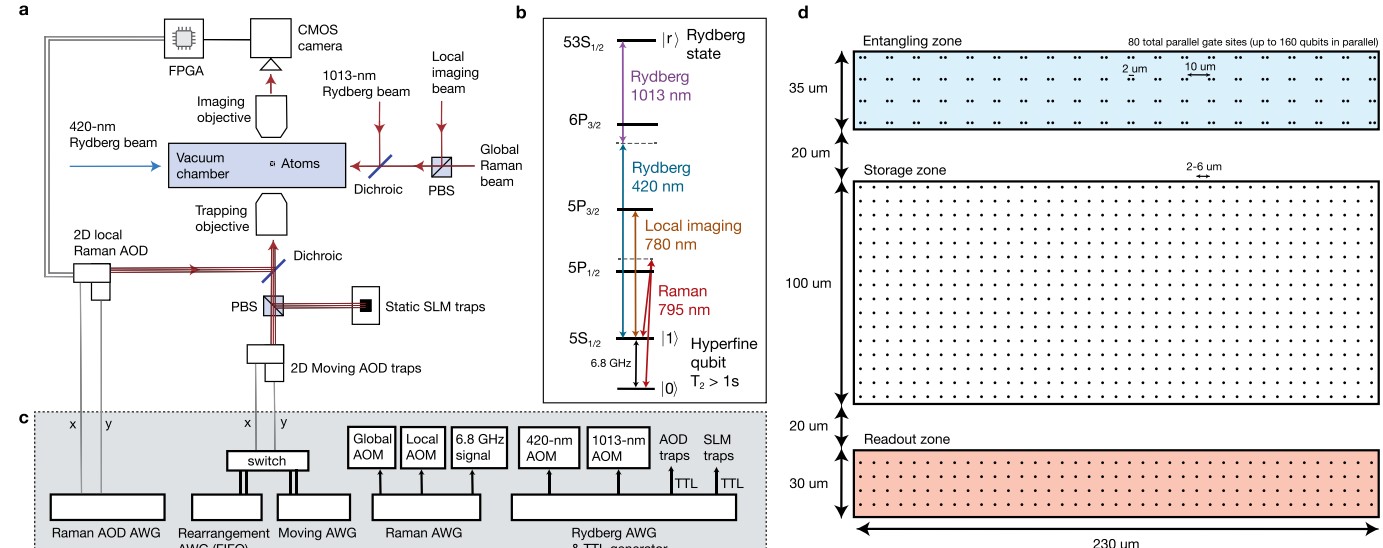

**Extended Data Fig. 1 | Neutral-atom quantum computer architecture.**
**a**, Experimental layout, featuring optical tools including static SLM and 2D moving AOD traps, global and local Raman single-qubit laser beams, 420-nm and 1,013-nm Rydberg beams and imaging system for both global and local imaging. **b**, Level structure for $^{87}$Rb atoms, with the relevant atomic transitions used in this work. **c**, Control infrastructure used for programming quantum circuits, featuring several AWGs. In particular, the moving and Raman 2D AODs are each controlled by two waveforms (one for the $x$ axis and one for the $y$ axis). An additional AWG is used in first-in-first-out (FIFO) mode for rearrangement before the circuit begins and then the moving AOD control is switched to the 'Moving AWG'. See ref. 30 for further SLM and pre-circuit rearrangement details, ref. 8 for further Rydberg AWG details and Rydberg excitation details, refs. 7,63 for further Raman laser and microwave control infrastructure details and ref. 7 for further moving AWG details. All AWGs (other than the 'Rearrangement

AWG') are synchronized to <10 ns jitter. During Rydberg gates, the traps are briefly pulsed off by a TTL. The FPGA processes images from the camera in real time and, in this work, sends control signals to the Raman 2D AOD for local single-qubit control. **d**, Example array layout featuring entangling, storage and readout zones. Zones can be directly reprogrammed and repositioned for different applications, as well as specific tweezer site locations. Tweezer beams and local Raman control are projected from out of plane. The entire objective field of view is 400 μm in diameter and, consequently, we do not expect or observe substantial tweezer deformation near the edges of our processor. During two-qubit Rydberg gates, we place atoms ≲2 μm apart within a gate site and gate sites are separated such that atoms in different gate sites are no closer than 10 μm during the gate. At our present $n = 53$ and two-photon Rabi frequency of 4.6 MHz, the blockade radius is roughly 4.3 μm, such that adjacent atoms are well within blockade and distant atoms are well outside blockade.

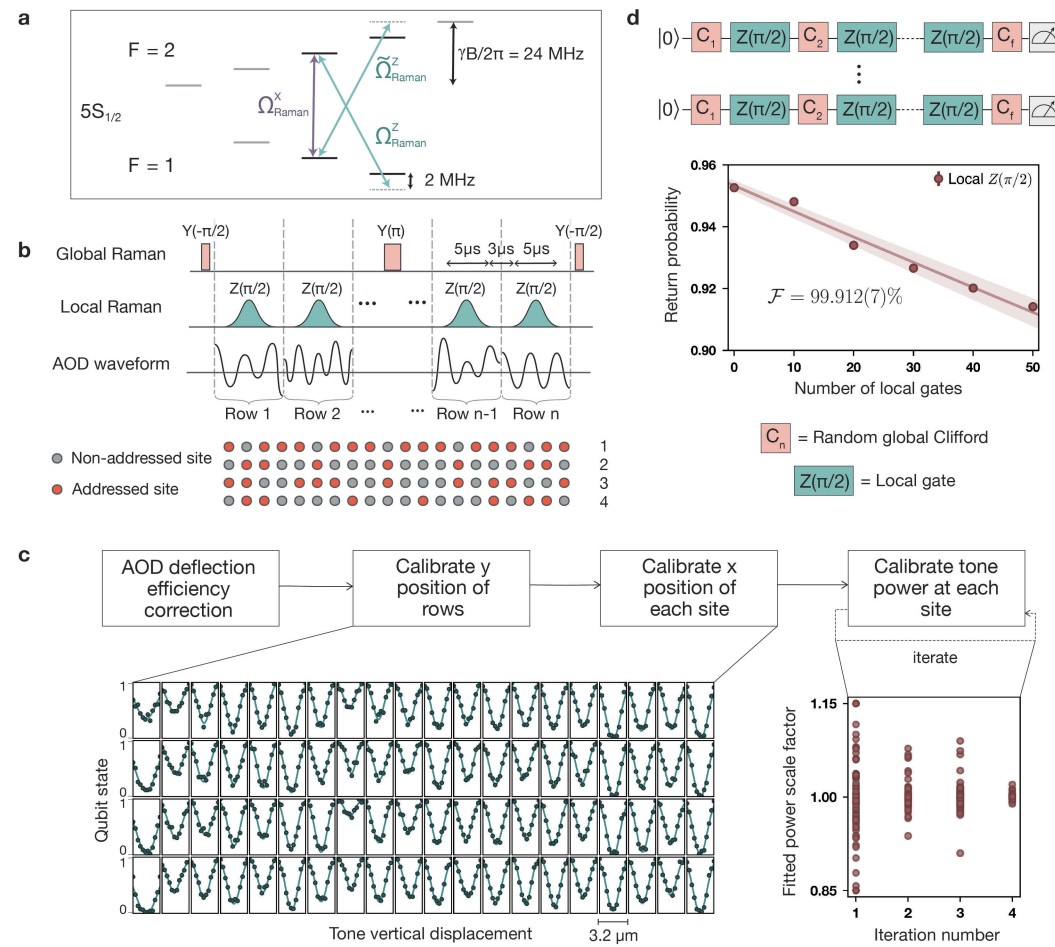

**Extended Data Fig. 2 | Single-qubit Raman addressing. a**, $5S_{1/2}$ hyperfine level diagram illustrating the two possible implementations of local single-qubit gates: resonant $X(\theta)$ (purple) and off-resonant $Z(\theta)$ (turquoise) rotations with two-photon Rabi frequencies $\Omega_{Raman}$. In this work, we use the Z rotation scheme and are blue-detuned by 2 MHz from the two-photon resonance. Owing to Clebsch–Gordan coefficients, $\widetilde{\Omega}^Z_{Raman} = -\sqrt{3}\,\Omega^Z_{Raman}$. **b**, Schematic showing the conversion of local $Z(\pi/2)$ into local $X(\pm\pi/2)$ gates, in which the pulses before (after) the central $Y(\pi)$ have positive (negative) sign, while leaving non-addressed qubit states unchanged. The Gaussian-smoothed local pulses have duration 2.5 µs for $\pi/4$ pulses and 5 µs for $\pi/2$ pulses and are performed on single rows at a time with a 3-µs gap between subsequent gates to allow the RF tones in the AODs to be changed (including this, duration is 5–8 µs per row). In this way, arbitrary patterns of qubits, such as the example drawn, can be addressed.

**c**, Calibration procedure used to homogenize the Rabi frequency over a 220 µm × 35 µm array. The position calibration is illustrated for 80 sites: approximate $X(\pi/2)$ gates are locally performed and the horizontal/vertical position of all tones is scanned in parallel such that a Gaussian fit returns the optimal alignment. After this, powers are iteratively calibrated until the fitted scale factors for the individual RF tones converge to unity. **d**, Single-qubit randomized benchmarking of local $Z(\pi/2)$ gates. The local gates are interleaved with random global single-qubit Clifford gates and the final operation $C_f$ is chosen to return to the initial state. Each data point is the average of 100 random sets of Clifford gates and fitting an exponential decay to the return probability quantifies the fidelity $\mathcal{F}$ per local gate. Note that we apply all 51 global Clifford gates for each data point, such that errors from the global Clifford gates (as well as SPAM errors) do not contribute to the fitted value.

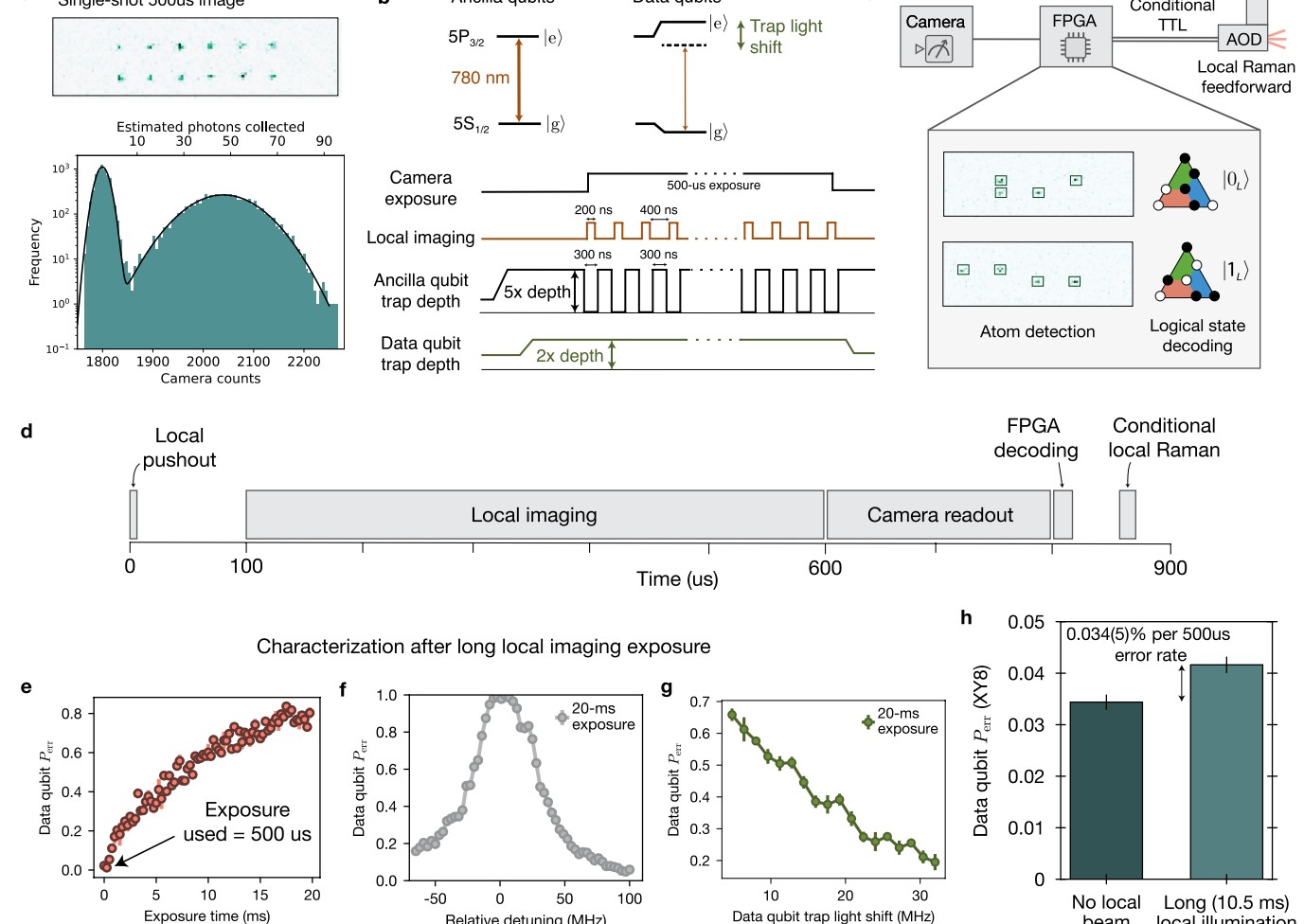

**Extended Data Fig. 3 | Mid-circuit readout and feedforward. a**, Single-shot 500-μs local image in the readout zone, in which the peak corresponds to roughly 50 photons collected by the CMOS camera. **b**, Atomic transition and pulse sequence used for local imaging of ancilla qubits. The data-qubit trap-light shift suppresses data qubit errors, as well as the large spatial separation between entangling and readout zones. We avoid quickly losing the readout-zone atoms during local imaging by using a 5× higher trap depth and we pulse the ancilla qubit traps and local imaging light to image directly on resonance while avoiding negative effects of large trap-light shifts. **c**, Diagram of components involved in mid-circuit readout and feedforward steps. Atom detection and logical-state decoding occur using the FPGA, which then outputs a conditional TTL to gate local Raman pulses performed on logical qubits in the entangling zone. **d**, Diagram of approximate timings for a mid-circuit feedforward cycle. First, the $F = 2$ population is pushed out (in 10 μs) and then the remaining $F = 1$ population is imaged locally for 500 μs. The 24 rows of pixels covering the readout zone are read out to the FPGA in 200 μs, after which processing is performed. Finally, a conditional TTL output based on the decoded state gates on or off local Raman pulses. The whole readout and

feedforward cycle takes less than 1 ms and can be sped up in the future by optimizing local imaging and camera readout. **e**–**g**, Characterization of the error probability of data qubits during local imaging. **e**, Data-qubit error probability (fraction of population depumped from $F = 2$ to $F = 1$) as a function of local imaging duration out to 20 ms to quantify the effect of the local imaging beam on data-qubit coherence for very long illumination. **f**, Data-qubit error probability after 20 ms of local imaging, as a function of detuning of the local imaging beam, showing suppression of error red-detuned or blue-detuned from the data-qubit transition. **g**, Equivalently, increasing the trap depth of the data qubits enables suppression of decoherence owing to the local imaging beam. Because qubits in the readout zone are imaged while their traps are pulsed off, any light shift of the data-qubit transition from the traps contributes directly to the relative detuning. **h**, For a long, 10.5-ms local beam illumination with optimal local imaging parameters, we observe a 0.7(1)% increase in data-qubit error during an XY8 dynamical decoupling sequence. This suggests a roughly 0.034(5)% error probability for the data qubits during the 500-μs mid-circuit readout image used in this work.

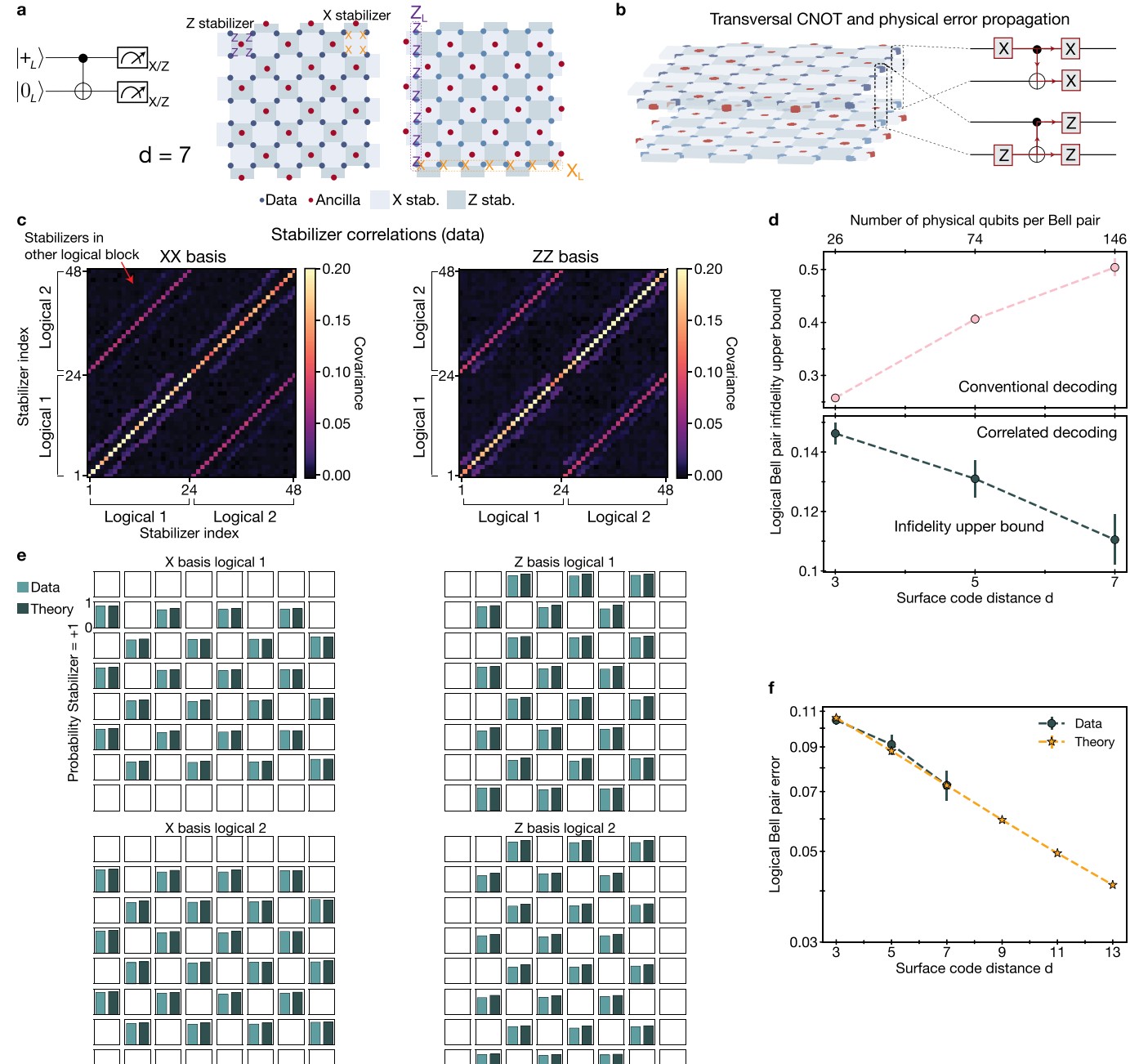

**Extended Data Fig. 4 | Further surface-code data. a**, Depiction of Bell-state circuit and $d$ = 7 surface codes. **b**, Diagram showing the transversal CNOT and physical error propagation rules. **c**, Covariance of the 48 measured stabilizers in both bases. The correlations near the diagonal corresponds to adjacent stabilizers within each block. Strong correlations are also observed with the stabilizers of the other block owing to the error propagation in the transversal CNOT. **d**, Bell-pair infidelity upper bound (as opposed to estimated Bell-pair error in Fig. 2d; see Methods), showing improvement with increasing code distance. **e**, Probability of no detected error for each of the 96 measured stabilizers, showing agreement when compared with the theoretical values from empirically chosen error rates (experiment average = 77%, theory average = 82%). Note that X-basis logical 1 and Z-basis logical 2 have higher

stabilizer error probability owing to the error propagation in the transversal CNOT (reducing expectation values relative to if the transversal CNOT is not performed). **f**, Using the empirical error rates that correspond to data-theory agreement for the measured stabilizers in **e**, our simulations for improvement in Bell-pair error, as a function of code distance, are in good agreement with experiments. The empirical error rates used are consistent with the 99.3% two-qubit gate fidelity, measured for this larger array, as well as the roughly 4% data-qubit decoherence error (integrated over the entire circuit and measured by the Ramsey method). These dephasing error rates are dominated by a complex moving sequence as we prepare the two surface codes in a serial fashion (see Supplementary Video) and would be much smaller for a repetitive error-correction experiment.

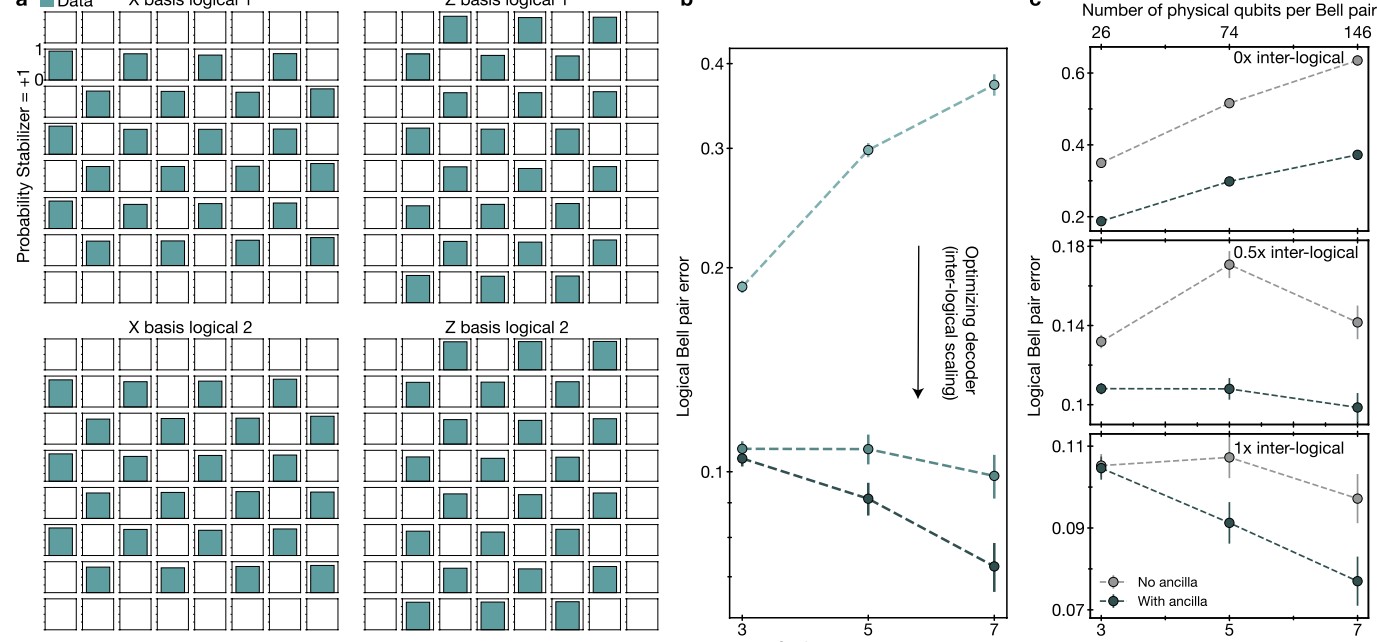

**Extended Data Fig. 5 | Surface-code preparation and decoding data.**
**a**, Surface-code stabilizers for the two independent $d$ = 7 codes following state preparation. The entire movement circuit corresponding to the transversal CNOT is implemented and the transversal entangling-gate pulse is simply turned off. The mean stabilizer probability of success across the 96 total stabilizers is 83%. The high probability of stabilizer success of the two independent codes in both the X and Z bases shows that topological surface codes were prepared (and Extended Data Fig. 4 shows that they were preserved during the transversal CNOT). We note that physical fidelities were slightly lower during this measurement because of calibration drift and, therefore, these results slightly underestimate performance relative to the data in Fig. 2 and Extended Data Fig. 4. **b**, Logical Bell-pair error while optimizing the decoder by (inversely) scaling the weights of the inter-logical edges and hyperedges that connect the stabilizers of the two logical qubits (higher values correspond to lower pairing weights). More concretely, the probability $p$ of the error mechanism corresponding to the inter-logical edges/hyperedges is scaled and the weights are calculated as $\log((1 - p)/p)$. Qualitatively, optimizing this scaling value optimizes with respect to the probability that errors are before or after the transversal CNOT, as errors before the CNOT will lead to

correlations between the two logical qubits, corresponding to the inter-logical edges. As the decoder is optimized by tuning the inter-logical scaling factor, the performance for all three code distances improves, and the larger code distances improve faster when approaching the optimal decoding configuration, as expected. These data are consistent with the decoder being properly optimized for all three code distances, consistent with the fact that our improvement with code size does not originate from suboptimal decoder performance for low distance. Note that the $y$ axis is log scale. **c**, Logical Bell-pair error when using (black) and not using (grey) the ancilla stabilizer measurement values, as a function of the scaling of the inter-logical edges and hyperedges that connect the stabilizers of the two logical qubits. The ancilla measurements contribute to the correction procedure and contribute more for smaller values of the inter-logical scaling, as they correspond to errors that happen before the transversal CNOT. 0× inter-logical scaling corresponds to conventional decoding within the two independent surface codes. For the 1× inter-logical scaling plotted here, the $d$ = 7 inter-logical scaling parameter is chosen slightly different from in Fig. 2d to have consistency across the three code distances (which produces measured values within error bars).

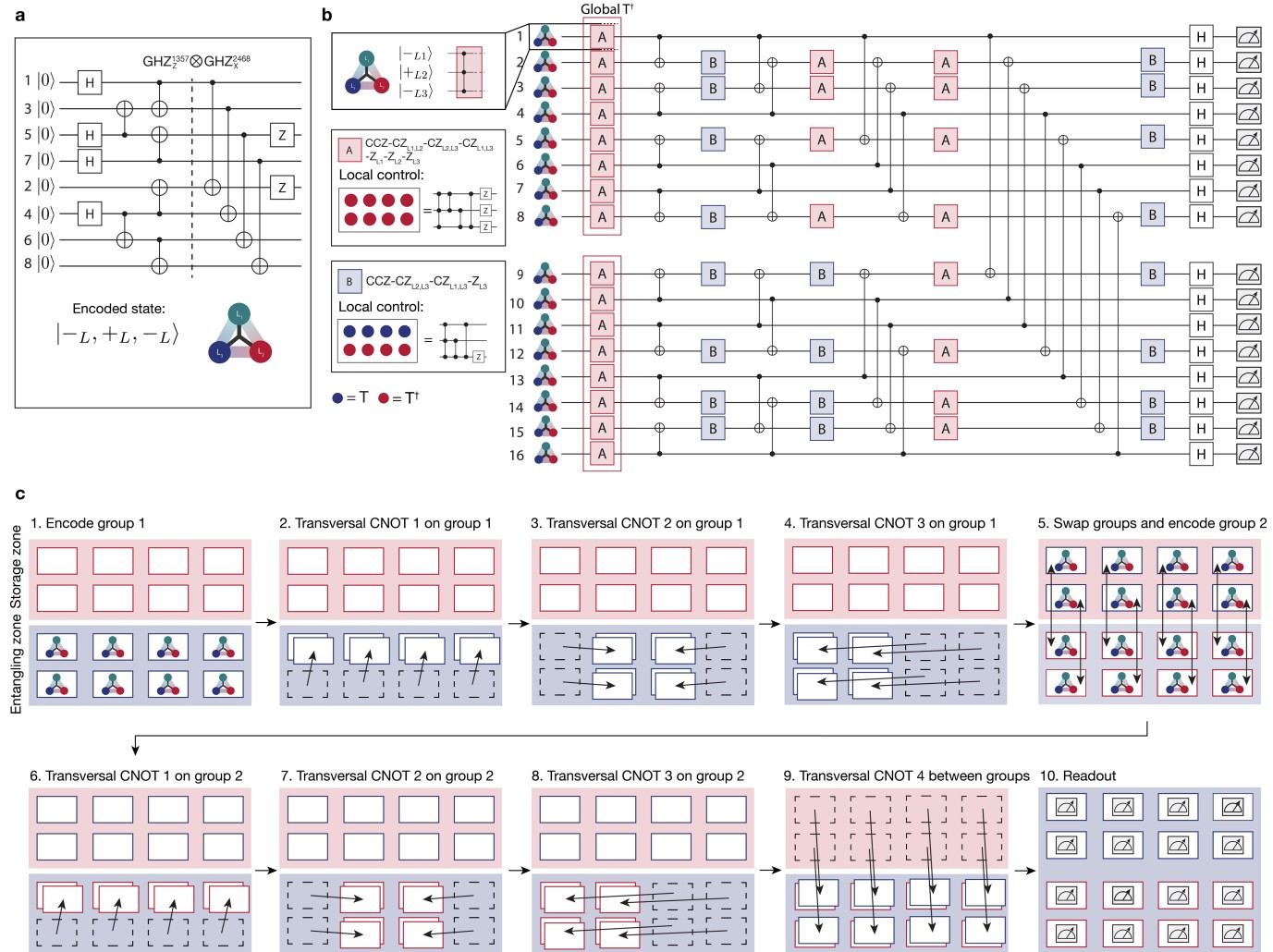

**Extended Data Fig. 6 | [[8,3,2]] and hypercube encoding. a**, State-preparation circuit for the [[8,3,2]] code, in which two four-qubit GHZ states are simultaneously prepared and subsequently entangled. This initializes an [[8,3,2]] code with logical states $|-_{L1},+_{L2},-_{L3}\rangle$. **b**, 4D hypercube circuit performed on 48 logical qubits (128 physical qubits). The circuit is drawn on the block level, in which each block consists of three logical qubits and eight physical qubits. The first in-block gate layer is performed with a global $T^\dagger$. The local gate patterns, and the corresponding logical gates they execute within each code block, are illustrated in the inset. **c**, Diagram illustrating the code-block movements and use of the processor's zoned architecture throughout the circuit. Initially, eight [[8,3,2]] code blocks are prepared in the entangling zone and atoms for later state preparation of eight additional code blocks are loaded in the storage zone. The code blocks in the entangling zone are then picked up and interlaced with adjacent blocks to perform three transversal CNOT layers. The two groups of eight code blocks are then swapped and the same procedure is repeated with the second group of code blocks. The first group of code blocks is then moved back into the entangling zone and interleaved with the atoms of the first group to perform a final parallel transversal CNOT. The layers of CNOT gates connect the code blocks such that a 4D hypercube on 16 blocks of [[8,3,2]] codes is constructed. See also Supplementary Video.

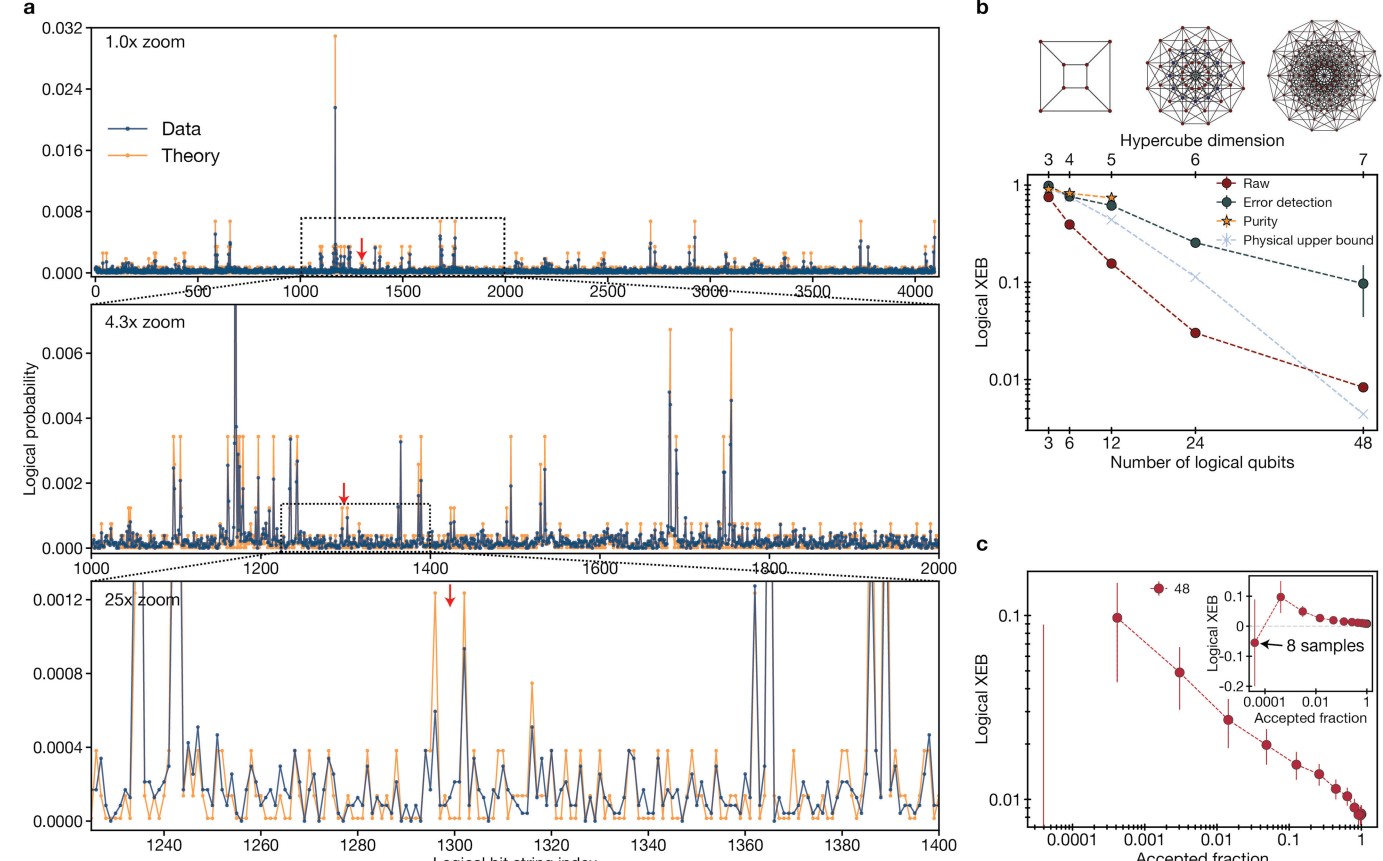

**Extended Data Fig. 7 | Further [[8,3,2]] circuit sampling data. a**, Overlap of error-detected 12-qubit sampling data with the theoretical distribution (same data as fully error-detected case in Fig. 5b). Progressive zoom-ins show the agreement between theory and experiment, down to the level of $10^{-4}$ probability per bitstring. This error-detected dataset is composed of 23,545 shots (raw dataset is 138,626 shots). Note that we simultaneously measure on two groups of 12 logical qubits; plotted here is only one of the two 12-logical groups with an XEB of 0.69(1), whereas in plots Fig. 5e,f and Extended Data Fig. 7b, we average the two logical groups, which gives a measured XEB of 0.616(7). **b**, Same data as Fig. 5f but with purity (orange), as measured by two-copy measurement, also plotted. The measured XEB is slightly below the measured purity, providing evidence that the XEB is a faithful fidelity proxy. We further note that, under error detection, the logical XEB for these IQP circuits should be a good fidelity proxy. Notably, the behaviour can be different

for the raw, uncorrected data, as the circuit we apply on the physical level is not IQP. Without applying error detection, not all errors are logical errors and, therefore, the circuit differs from IQP behaviour and can lend itself to a different scaling. For systems of 3, 6 and 12 logical qubits, several systems are measured in parallel and their results are averaged. We note that, although our preparation of [[8,3,2]] code states makes these states on a cube, it does not have CNOTs between two pairs of qubits in the first step and, therefore, does not have the full gate connectivity of a cube. Instead, we can interpret these CNOTs as having been included but then compiled away as they commute with the state. We neglect this in plotting our physical-qubit connectivity, which is derived from entangling 3D cubes on a 4D hypercube connectivity, realizing a 7D hypercube. **c**, 48-qubit XEB sliding-scale error-detection data. The point with full postselection on all stabilizers being perfect returned only eight samples, so we omit this point from the plot in the main text for clarity.

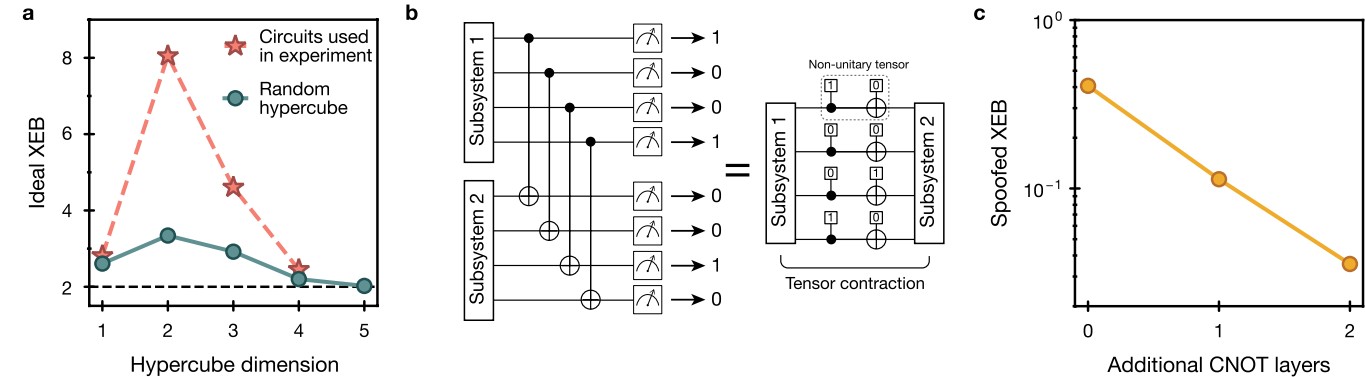

**Extended Data Fig. 8 | Theoretical exploration of hypercube IQP circuits.**
**a**, Anticoncentration property of our circuits. The circuit is said to be anticoncentrated if its output distribution is spread almost uniformly among all outcomes, without the probability being concentrated on a subset of bitstrings. This property is crucial for many proofs of classical hardness[20,100] and, thus, it is desired for our sampling circuits to anticoncentrate. The plot shows that the output distribution of random hypercube circuits (randomized in-block operations and randomized control/target in out-block CNOT layers) anticoncentrates as the dimension of the hypercube is increased and the XEB (which captures the output collision probability) converges to the uniform IQP value of 2 (here using Clifford circuits; that is, circuits comprising random CZ and Z only)[20]. This suggests that sampling from the ideal output distribution can be classically hard. In general, the hypercube IQP circuit ensemble converges to the uniform IQP ensemble in total variation distance as the depth and hypercube dimension are increased (M.K. et al., manuscript in preparation). The specific circuit instances implemented in the experiment also anticoncentrate quickly with increasing hypercube dimension. **b**, A single layer of the hypercube circuit admits an efficient tensor-network contraction scheme, which allows us to evaluate the ideal and experimental XEB values. The final out-block CNOT layer is immediately followed by the measurement, which can be incorporated into a non-unitary tensor that is contracted between the two halves of the

system (controls and targets of the final CNOT layer). This contraction scheme reduces the memory requirements to half the system size, which enables bitstring amplitude evaluation for the 48-qubit experiment. This simulation approach can be made much more expensive by applying further out-block operations within the two subsystems, forcing the blocking of the intra-partition tensors, which increases the memory and runtime requirements (Fig. 5d).
**c**, To understand the effects of finite XEB on required classical simulation time, we explore whether our circuit families can be 'spoofed' with a cheaper, approximate simulation that achieves moderately high XEB scores[102], studied here for a 24-qubit system with full state-vector simulation. The spoofing algorithm works by independently sampling from the two halves of the system (two groups of 12 qubits), effectively removing the final layer of CNOTs. This further reduces the simulation complexity, as each of the halves can, in principle, be independently simulated with the efficient approach from **b**. The plot shows that the spoofed XEB for the 24-qubit non-Clifford circuit can be exponentially reduced by extending the circuit with further gate layers (similar to the approach used to decrease the performance of the efficient hypercube contraction), for a particular extension of our circuit. This result shows that future work can consider adding extra CNOT layers into these circuits to demonstrate quantum advantage (in the presence of finite experimental noise).

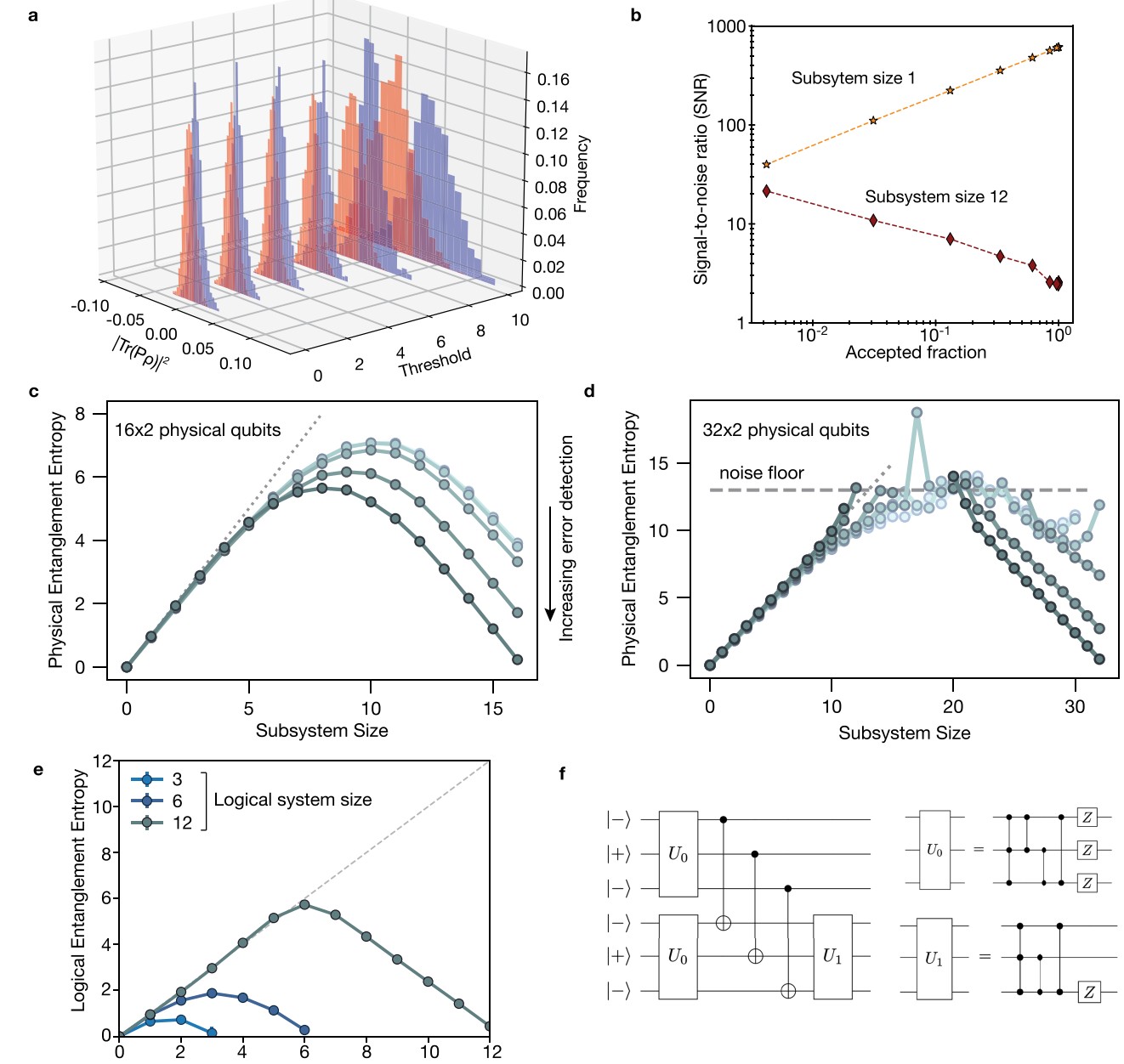

**Extended Data Fig. 9 | Further Bell-basis measurement results. a**, Histogram of $|\mathrm{tr}(P\rho)|^2$ for all $4^6$ Pauli strings $P$ in the six-logical-qubit circuit, as a function of stabilizer postselection threshold (that is, the number of correct stabilizers across the $6 \times 2$ logical qubits). Blue (red) indicate Pauli strings that are expected to have $|\mathrm{tr}(P\rho)|^2 = 0.0625\,(0)$. The separation between the histograms improves as more postselection is applied. **b**, Signal to noise (purity divided by statistical uncertainty of purity) as a function of sliding-scale error detection (converted into accepted fraction) for the 12-logical-qubit two-copy measurements, in which subsystem size 1 indicates a single logical qubit in one copy and subsystem size 12 indicates all logical qubits. For subsystem size 1, the signal-to-noise ratio gets worse as data are discarded, as the signal does not change (maximally mixed) but the number of repetitions decreases. By contrast, for the global purity, the signal to noise increases, as near-unity purities are faster to measure[113].

**c**,**d**, Entanglement entropy when analysing the circuit as a physical Bell-basis measurement as opposed to a logical Bell-basis measurement. For logical entanglement entropy calculations, we average over all possible subsystems of that given subsystem size, which we find behaves very similarly to, for example, contiguous subsystems owing to the high-dimensional hypercube connectivity. In the physical qubit entanglement entropy calculations, we randomly choose from the possible subsystems, as there are many. **c**, Six logical (16 physical) qubits per copy. **d**, 12 logical (32 physical) qubits per copy. The finite sampling imposes a noise floor for very high entanglement entropy values. **e**, Entanglement entropy measurements, as in Fig. 6b, but as a function of logical subsystem size. **f**, Logical circuits used for benchmarking magic. For one CCZ, we include $U_1$ and omit $U_0$; for two CCZs, we include $U_0$ and omit $U_1$; for the three CCZs, we include both $U_0$ and $U_1$.