## [Peer Review File · Nature]

Manuscript Title: Logical quantum processor based on reconfigurable atom arrays

Reviewer Comments & Author Rebuttals

Reviewer Reports on the Initial Version:

Referees' comments:

Referee #1:

Remarks to the Author:

The manuscript by Bluvstein and coworkers demonstrates a quantum processor utilizing logical encoding through reconfigurable atom arrays consisting of hundreds of physical qubits. Employing both $[[7,1,3]]$ and $[[8,3,2]]$ encodings, the authors successfully achieve transversal CNOT gates, prepare logical GHZ states, implement mid-circuit feedforward, demonstrate XEB up to 48 logical qubits, and measure the entanglement entropy in scrambling circuits at the logical level. This work represents an impressive and timely experimental demonstration, highlighting the key advantages of reconfigurable atom arrays for parallel non-local gates, which are crucial for fault-tolerant quantum computing.

Before I can fully recommend publication of this work, I have some questions and suggestions for the authors:

1. The authors claim that "the logical Bell pair improves with larger code distance," as shown in Figure 2d. An alternative explanation might be that the correlated decoder (rather than the underlying surface code encoding) performs better for larger distances. It is important to rule out such an alternative explanation to assert that the measured logical Bell pair error truly reflects the logical errors associated with the underlying physical system.
2. It would be helpful if the authors could clarify why the logical GHZ state fidelity is only 72%, as depicted in Figure 3.
 - a. First, providing an explicit definition of logical fidelity (in the Methods or supplementary materials) would help avoid ambiguity.
 - b. Considering that the entire procedure to prepare the logical GHZ state (including ancilla-assisted FT initialization, transversal CNOT, and Pauli measurement) is fault-tolerant, we would expect the logical error to be suppressed to the 2nd order of the physical error rate, $O(\epsilon^2)$, for $[[7,1,3]]$ encoding.
 - c. However, the fidelity of 72% for the logical GHZ state seems to be limited by the first-order physical error. What prevents the logical GHZ state from achieving an even higher fidelity? It would be helpful if the authors could provide an estimated error budget to explain the limited fidelity of the logical GHZ state.
3. It would be beneficial if the authors could provide more information on the protocol for the slide-scale error detection, as this is an important technique used to evaluate the performance of the atom array quantum processor.

Referee #2:

Remarks to the Author:

The authors present results from several quantum error correction experiments using an array of neutral atoms. Among these experiments the authors claim that they (1) - demonstrate threshold-like behaviour for a pair of surface codes undergoing a transversal entangling gate (2) - prepare an array of color code qubits in a GHZ state. The authors extend this experiment to produce 40

logical color code qubits. (3) - I have differing opinions for all of these experiments. (4) - The authors measure the entanglement entropy of a scrambled system using Bell measurements. Among these experiments there are some very impressive results. However, given my concerns with (1) I am not willing to recommend acceptance. I give my comments to these experiments below.

(1) - I have concerns about the claims of experiment 1. I can give much simpler explanation for the limited number of results that are presented in the manuscript without preparing any surface code states. I have attached a note pointing to some of my concerns.

(2) - I find this experiment very interesting and well carried out. The authors are much more thorough with the analysis of the results here as compared with (1).

I am curious to know how the state preparation fidelity is calculated. Ref. 2305.13581 for instance uses two different tomographic experiments to verify they prepare the state they claim, to argue for break even, for the magic state they prepare as compared with preparation using physical qubits. The authors could be more clear about how fidelity is calculated for their logical qubits.

The demonstration of feedforward here is perhaps not the most impressive, as the feedforward could be implemented in software without post selection. After all, one can correct the 'correct' Bell pair with a simple Z rotation on one of the two qubits. The XX and YY expectations therefore do not need to vanish without the use of feedforward or post selection, with a straight forward use of classical updates in software. Nevertheless, I am impressed by the four-qubit encoded GHZ state.

The authors make use of the correlated decoder in this experiment. Given my criticisms of this decoder in experiment (1), it would be good for the authors to comment on the use of the correlated decoder here. It is my impression these two experiments are using a correlated decoder in a different way, but I am interested in the authors' comment on their contrast here.

Experiments (3) and (4) run transversal non-Clifford circuits between distance 2 codes. These are interesting applications, and it is very exciting the authors can produce such a large number of logical qubits, and run circuits between them. The authors demonstrate improvements in their experimental results by demanding a smaller fraction of errors are detected. I think this is a very nice early demonstration of error correction used in 'small' logical circuits.

One question I have, there is a comment on the decisions about post selection following error detection earlier in the paper around experiment (2). As the authors mention in a throw away comment that some errors are more likely to have introduced a logical error than others. Could the authors elaborate on this point? I am curious to know how this calculations is made.

Minor comment - in contrast to the authors' comment early on in this section of the paper, one can perform non-Clifford gates with the surface code, although I agree it is perhaps not straight forward in comparison to a transversal gate. See Sci. Adv. 6, eaay4929 (2020). In any case, magic states are not strictly required for a 2D architecture, as the authors write.

A comment on the surface-code experiment in “Logical quantum processor based on reconfigurable atom arrays”

The authors claim to produce two copies of the surface code and entangle them with a transversal CNOT gate. They then collapse the system with single-qubit measurements to make a fault-tolerant readout operation. Let me put forward an alternative explanation for the below-threshold behaviour shown in Fig. 2(d). I will argue that one could reproduce these results using d^2 independent Bell pairs, and that in this sense the results presented by the authors do not show persuasively that they have produced a surface code or any kind of long-range order at all. As such, I would suggest the authors should reconsider their methodology on this particular experiment in order to justify their claims. To reach my point, I will simplify the notion of the ‘correlated decoder’. I believe, with a careful look at the system preparation, the notion of a correlated decoder hides the fact that the system they have produced might be a much more simple correlated system, rather than two distinct surface codes with an entangled logical code space, as is the goal of this demonstration. My argument is consistent with the results the authors have presented using an uncorrelated decoder. Additionally, if my argument below is reasonable, error correction will only work successfully with specific choices of an initial state and readout. This is inconsistent with the function of a transversal gate, that should act correctly between arbitrary choices of input state and readout. Concluding from my observations that are summarised in this note, I would say that the authors would need a stronger signature that they have really demonstrated an entangling gate between two surface codes. Furthermore, it would be good to see evidence that they have even produced surface codes on either code block, beyond what is presented in the current manuscript.

I. SUMMARY OF THE EXPERIMENT

Let me put forward some notation. We have two identical code blocks with qubits indexed q and lattice vertices and plaquettes indexed v and p . I will distinguish between the two code blocks with superscript indices 1 and 2 such that I have Pauli operators X_q^1 and Z_q^1 acting on qubit q of the first copy and X_q^2 and Z_q^2 acting on the second copy. The qubits are aligned such that when we apply the transversal CNOT gate U , we apply physical CNOT gates between equivalent qubits q of the two copies. To be concrete let us suppose that U is such that the copy with superscript index 1 is the control qubit and the copy with superscript index 2 Likewise we have the standard star and plaquette operators A_v^1 and A_v^2 , and B_p^1 and B_p^2 acting on the two copies. The vertex and plaquette indices of the two blocks are also identified such that $UA_v^1U = A_v^1A_v^2$ and $UB_p^2U = B_p^1B_p^2$. Similarly, let us also write down some representative logical operators \bar{X}^1 , \bar{X}^2 , \bar{Z}^1 and \bar{Z}^2 where \bar{X}^1 and \bar{X}^2 , and \bar{Z}^1 and \bar{Z}^2 have the same support on their respective blocks.

With the logical CNOT gate defined as above the authors perform the following operations.

1. They prepare qubits in the first surface code block as $|+\rangle_q^1$ and the second as $|0\rangle_q^2$.
2. They measure the operators B_p^1 and A_v^2 once. These operators give random outcomes, and may give rise to measurement errors.
3. The operator U is applied, thereby entangling the two code blocks.
4. All qubits are read out in either the Pauli-X or Pauli-Z basis. This infers the values of the stabilizers, and the logical operators, enabling us to perform error correction. Let us just suppose we measure all qubits in the Z-basis, but an equivalent argument holds for Pauli-X measurements.
5. The syndrome data is interpreted, to determine if a decoder can recover $\bar{Z}^1\bar{Z}^2$.

II. THE CORRELATED SYSTEM

Let me begin by arguing that the decoding problem that is being solved by the correlated decoder is more simple than decoding two copies of the surface code, thereby enabling below threshold like behaviour to emerge. Rather, I can argue that a correlated system is produced, that can be decoded as a single-copy of the surface code. I stress that I will use the term ‘correlated system’ and not ‘surface codes’, because I do not believe the authors have demonstrated they have produced the latter. This argument is more than semantics. It may be that the results can be reproduced by independent Bell pairs, and not by any long-range entangled codes.

Roughly speaking, over steps 1-3, stabilizers are prepared, either by initialisation and entangling operations, or by measurements. At steps 4 and 5, a collapse measurement is made to infer the values of the stabilizers we have prepared, and decoding is performed. My argument will hold with a very simple noise model, where the only errors that occur, occur during readout at step 4. I will also expand on the argument the authors have presented, that the standard uncorrelated decoder that does not use information from both code blocks in parallel cannot work.

The key to my argument here is to concentrate on the stabilizers that are produced at steps 1-3 above. As we concentrate on the case where we collapse the qubits in the Pauli-Z basis, we will concentrate on the Pauli-Z type stabilizers. At the collapse step we infer the value of some Pauli-Z stabilizers, and logical-Z operators. We assume some readout errors at this stage that must be corrected for. In order to do that, we must know what reliable stabilizer information we have available. Let us work carefully through the first three steps to see what stabilizer information we have available at readout. At step 1, the qubit preparation is such that all of the B_p^2 stabilizers are prepared in their +1 eigenstate. We can simplify my argument here and assume there is no noise occurring here.

As the authors find, we can solve the problem at hand by taking syndrome data from both code blocks into account simultaneously. Importantly, we do not need to correct two independent logical operators. Rather, we only aim to recover the value of $\bar{Z}^1\bar{Z}^2$. This can be achieved with stabilizers we have prepared reliably, namely, $UB_p^2U = B_p^1B_p^2$. In fact, this simplifies things greatly. All of these operators commute with error operator $X_q^1X_q^2$. Rather, we are only interested in errors X_q^1 and X_q^2 . In fact, these act equivalently on the system. These errors alter the sign of the logical operator $\bar{Z}^1\bar{Z}^2$, or any stabilizer $UB_p^2U = B_p^1B_p^2$, if q is in the support of any of these operators.

If you prefer, you can think of this as a correlated system with a two qubit code at site q (consisting of the qubit at site q from each block) where the local inner code has stabilizer $X_q^1X_q^2$. Now errors are the single qubit $X_q = X_q^1$ (or equivalently $X_q = X_q^2$), and the logical operator $\bar{Z} = \bar{Z}^1\bar{Z}^2$ and the stabilizers $B_p = B_p^1B_p^2$ of the original system are composed of the logical and the Pauli-Z operators $Z_q = Z_q^1Z_q^2$ of these two qubit codes at site q in this concatenated system. Importantly, in this coarse grained picture, we are only decoding a single copy of the surface code.

To summarise my criticism of this work to this point, I do not believe that it is a special type of decoder that allows below-threshold like behaviour. Rather, the system they have produced, (that is produced in a very similar way to which a transversal gate is applied between two surface codes), can be interpreted as a single surface code that can be decoded reliably, built on small two-qubit error detecting codes.

I would argue the below threshold behaviour can come from the correlated system using an ordinary minimum-weight perfect-matching decoder, and treating the correlated system as a surface code. This decoder has a threshold of around 10%, as in Ref 4 of the manuscript. Back of the envelope calculations show us that this system should be below threshold by calculating the error of a site. I will conservatively assume gates have a 1% error rate. Each site has two qubits, and each qubit is involved in four operations (three entangling gates and one measurement). We have then that $2 \times 4 \times 1\% \approx 8\% < 10\%$ is comfortably below threshold.

III. THE CORRELATED SYSTEM AS BELL PAIRS

Now to point to a bigger problem. It is not even clear that it was necessary to produce two surface codes to obtain the below-threshold plot the authors show in Fig. 2(d). Notably, my decoding strategy did not even use the measurement outcomes obtained at step 2. In fact, if I skip step 2, and only run through steps 1 and 3, I have produced a system with a stabilizer subgroup of the stabilizers I needed in my discussion above. Furthermore, the system I have produced would be a system of d^2 Bell pairs that do not interact. As such, I could even reproduce this result with two qubits, where I prepare d^2 Bell pairs in series. Of course, there is an additional challenge to preparing these Bell states in parallel, as they may experience correlated errors between the Bell pairs, so I will concede my last comment is a little unfair. However:

Given the results of the authors can be reproduced with a system of d^2 Bell pairs, it is not clear to me that the authors can argue persuasively that they have produced a (pair of) surface codes.

Indeed, step 2, which the authors claim to be a stabilizer readout step, could be any operation on the two code blocks that commutes with stabilizers $B_p = B_p^1B_p^2$ and logical operator $\bar{Z} = \bar{Z}^1\bar{Z}^2$, and the same results can still be obtained. For instance, it could easily be an identity operation, or some kind of noise simulator.

IV. MEASUREMENT ERRORS

I have given a simple picture for how the results the authors present could be obtained, without using any parity measurements. In my argument I directly correlated the values of B_p^1 and B_p^2 by multiplying them together. A black box decoder that takes both blocks of syndrome information into account could well be doing this. Let me make a second argument, now paying a little more attention to measurement errors that occur during step 2, where I now assume that parity measurements were performed as described. Indeed, I could imagine an improved decoder beyond the basic one I proposed that takes the distinct readings of B_p^1 and B_p^2 into account. However, it is strange to me that the transversal gate is also copying errors between the two codes to support successful error correction. This is not really in the spirit of a transversal gate, which is only designed to interact the two logical spaces of the codes, while leaving the rest of the system invariant.

First I will argue that random measurement errors cannot be corrected. This is consistent with the authors' observation that the standard uncorrelated decoder does not lead to below threshold behaviour. This leads me to assume that the measurement errors must be correlated between the two code blocks in order to decode successfully. I will then explain how measurement errors are copied. (This argument is a little different from that given by the authors in the Correlated Decoders section of Methods, where they only talk about how data qubit errors are transferred.) Afterwards I will argue that the correlated decoder they do use will only work because the transversal gates correlates the measurement errors between the two code blocks. Once again, it is a little telling that this decoding procedure really relies on U in order to work reliably.

Let me argue here that the stabilizers prepared by measurement at round 2 cannot be used for error correction. In fact, uncorrelated measurement errors are likely only a hinderance to obtaining a threshold. The authors make this argument by appealing to their definition of fault tolerance, but I can make the same argument in another way. At step 2, we prepare the B_p^1 stabilizers by measuring them. These give random outcomes $\pm_p B_p^1$ and do not reflect any errors that have occurred at this stage. I have used the \pm_p sign here to indicate that the outcomes over all the plaquettes p here is completely random on the surface code with boundaries. In principle though, although these outcomes are random, the uncorrelated decoder can try to use these random outcomes by comparing them to the read out at stage 4. However, measurement errors mean that these signatures cannot distinguish a single Pauli error from one or two ancilla measurement errors.

Indeed, fundamentally, the decoder here is trying to identify all of the Pauli errors that occur on the support of a logical operator. A single Pauli error supported on the logical operator at readout, that must be corrected, will give rise to two syndrome defects assuming defects assuming no other errors occur. I assume this readout Pauli error occurs with probability p_r . This same error signature is indistinguishable from two measurement errors (or only one at the boundary) acting on the ancilla qubits. Assuming these errors occur with constant probability p_a , a Pauli error is misdiagnosed with constant probability p_a^2 (or p_a at the boundary). Given that with increasing code distance, the support of the logical operator must grow, the likelihood of one of these misdiagnosis errors occurring at some location on the logical operator only increases with code distance. To leading order this explains the result of the standard decoder acting on the separate code blocks.

So then, uncorrelated measurement errors in fact are likely to lead the decoder to failure. How then, can a decoder that takes syndrome data from both code blocks work? Another way of presenting my argument above, that the two code blocks are correlated, goes as follows. Let us look at the action of U on these stabilizers that are prepared by measurement at step 2. This operation causes the issues introduced by readout errors when measuring stabilizers of code block 1 to spread to code block 2. We have that

$$U(\pm_p m_p B_p^1)U = \pm_p m_p B_p^1 \quad \text{and} \quad UB_p^2U = B_p^1 B_p^2$$

Let me stress that the error free value of B_p^2 is no longer known on application of U . We only know the exact value of $B_p^1 B_p^2$. In order to attempt to learn the value of B_p^2 , we can only multiply it by (our best knowledge of) the operator $\pm_p m_p B_p^1$ (where $m_p = -1$ if a measurement error has occurred, thereby flipping the value of the actual stabilizer measurement outcome). We therefore observe a system with observed stabilizers like

$$\pm_p m_p B_p^1 \quad \text{and} \quad \pm_p m_p B_p^2.$$

From here, one can imagine a decoder that can take this correlated data from both code blocks, and decode them in unison, such as the decoder used in the paper. In my earlier example, I simply multiplied the stabilizers at a given p together to cancel measurement errors and random outcomes. To reiterate, it is not really in the spirit of a transversal gate, that measurement error data is copied in its application in this way.

V. INITIAL STATES

Finally, it is also worth mentioning, that this trick for decoding successfully only works for specific logical input states. Specifically, the stabilizers only correlate as described above if the qubits are prepared and measured exactly as described in the paper. There are other configurations of input and readout that could be considered, but in the general case I would not expect to see threshold-like behaviour. Nevertheless, to argue for genuine logical entangling operations, one might want to conduct process tomography with many input and output configurations, to really verify the operation is as promised. This too leads me further to believe that the claims of the authors, that a transversal gate is performed between two surface codes, overstate the results of the experimental data presented.

Referee #3:

Remarks to the Author:

The manuscript “Logical Quantum Processor Based on Reconfigurable Atom Arrays” by D. Bluvstein et al. presents the experimental realization of a quantum processor that uses logical error-corrected qubits. The physical qubits that comprise the system are neutral atoms trapped in optical tweezers. This platform has significantly accelerated in the past few years and is now considered a leading contender for quantum computational platforms, as evidenced by the results in this manuscript. Enabling this technology are reconfigurable arrays of neutral atoms combined with local single-qubit gate operations and Rydberg-mediated entangling gates. The quantum processor is split into three zones for qubit storage, entangling gate operations, and readout (final and mid-circuit). The robust experimental control platform enables any-to-any connectivity between physical qubits and parallel gates on physical qubits, leading to fault-tolerant logical qubit operation. The authors test the capabilities of the experimental system by performing a CNOT gate between two logical qubits encoded in surface codes of varying code distance ($d=3,5,7$). The operational performance of the surface code corrected qubits is compared for a conventional and correlated error-decoded system made of the same number of physical qubits, indicating the superior performance of the surface codes using correlated errors. The fault-tolerant algorithmic performance of logical qubits encoded using a $d=3$ color code (like surface codes) is probed using state preparation infidelities and measuring the infidelity of a logical GHZ state.

The authors correctly point out that more complex algorithms cannot be implemented with the 2D codes demonstrated in the first part of the manuscript and consequently expand to a 3D code. The 3D code in the manuscript enacts a fast-scrambling circuit where the theoretical bitstring outcome is compared to the experimental outcome. The authors show that the outcome converges to the theoretical value as they progressively increase the error detection.

Finally, the authors investigate the logical qubit system’s ability to simulate quantum states by encoding two copies of a quantum state, performing a fast-scrambling circuit, and conducting a Bell basis measurement on the outcome. The extracted fidelity is higher than that extracted from the XEB measurements of the previous section, indicating at the least that the XEB metric is a reasonable measurement of the state purity. The entanglement entropy is extracted vs. system size, showing the expected improvement with increased error detection.

In complex quantum simulations, non-Clifford operations are required to construct the relevant quantum state. The authors measure the “magic resource” of the system or the additive Bell magic, quantifying the amount of non-Clifford operations needed to generate a quantum state. Overall, the manuscript represents a significant step forward in encoding logical qubits using various depth and complexity error-correcting codes. The logical qubits are used to generate entangling operations and perform quantum circuits, which puts the platform along the path toward more complex quantum computation and quantum simulation with logical qubit states.

The impressive experimental achievement in the manuscript warrants publication in Nature, provided the authors can address the questions and concerns outlined below.

1-The statement, “..., free from gate errors and ...” in the first paragraph of the first section (Logical processor based on atom arrays) is unclear, and the authors should qualify what they mean by this statement. Single qubit gates performed in these zones on the physical qubits would indeed register

errors, and the stored qubits would be subject to idle bitflip errors from scattering of tweezer light. I assume they are referring to logical qubit errors, but it is unclear. Generally, the authors do a good job throughout the text to distinguish between logical and physical qubits and errors; however, in a few cases, it is more ambiguous.

2-The authors quote their coherence in the paper as " $T_2 > 1s$." this number requires more context within the main text as it results from the coherence extracted after an XY16 128 pi pulse sequence. The T_2^* for this system is significantly less at 4ms, as quoted in Reference 7. While the XY16 sequence could be continuously run to maintain coherence in the storage zone, as the atoms move between additional zones, maintaining the sequence to preserve coherence for each atom throughout these zones is challenging.

3-Fig. 1 should include scale bars. These dimensions are provided in ED Fig 1; however, it would be easy to have them in the main text unless they are excluded for a particular reason. If Fig. 1a is the entire field of view of the objective. If so, the authors should comment on tweezer deformation on the perimeter. Additionally, denoting the Rydberg blockade radius in the text or on the figure can give some additional context to the reader.

4-As the Fault tolerance of these operations hinges on the parallel realization of single and two-qubit gate operations, have the authors performed similar measurements to Fig1b in the "entangling zone" for single atoms (i.e., the non-dimer case) to ensure that there is no crosstalk between "gate sites?"

5-In the second section (improving entangling gate with code distance), the authors indicate the presence of correlated errors, and this discussion is expanded upon in the ED. However, from the ED, it would seem as though the correlated errors are introduced to the system intentionally. In contrast, in the main text, it is less clear if they result as an intentional experimental choice.

6-Fault-tolerant is used often throughout the manuscript, and for the average Nature reader not directly in the field of quantum computing, this may need to be clarified. A sentence or two geared towards these readers would help contextualize the results' importance.

7-In the section "Complex logical circuits using 3D codes," the authors claim that their XEB score is almost an order of magnitude greater than previous work in superconducting qubits. This requires more context since the superconducting circuit case is not based on logical qubits, as is the case for this work. This does not diminish the results, but this is not an apples-to-comparison.

8-The Authors should consider adding more context and explanation surrounding the additive Bell magic measurements. This is relatively new in the literature, and readers would benefit from more context.

9-In the ED portion of the manuscript, under the section "Programmable single-qubit gates," The randomized benchmarking sequence discussion in the last paragraph is confusing. At first read, it appears that the randomized benchmarking is performed using only 50 pi/2 rotations. A study of the ED Figure 2 shows that this is not the case, and the authors should clarify this in the text.

10-In the ED portion of the manuscript, under the section "Direct fidelity estimation and tomography," I think it would be helpful to include a sample calculation for one of these elements that details the relevant assumptions and measurements that come into the calculation.

These results are a significant advancement for the field, and I commend the authors on this impressive work.

Author Rebuttals to Initial Comments:

Referee responses and summary of revisions

Nature Manuscript number: 2023-10-18750 Bluvstein

November 16, 2023

We would like to thank all Referees for their careful reading of our manuscript and many useful comments and suggestions. In what follows we address all comments point by point and indicate revisions when appropriate. The blue text are the referee comments, the black text are responses, the red text are revisions, underlined text indicates text additions, and strike-throughs indicates text removals. At the editor's request we have made efforts to not lengthen our manuscript, and in fact have made additional revisions that have reduced the main text by roughly 100 words.

Reviewer: 1

The manuscript by Bluvstein and coworkers demonstrates a quantum processor utilizing logical encoding through reconfigurable atom arrays consisting of hundreds of physical qubits. Employing both $[[7,1,3]]$ and $[[8,3,2]]$ encodings, the authors successfully achieve transversal CNOT gates, prepare logical GHZ states, implement mid-circuit feedforward, demonstrate XEB up to 48 logical qubits, and measure the entanglement entropy in scrambling circuits at the logical level. This work represents an impressive and timely experimental demonstration, highlighting the key advantages of reconfigurable atom arrays for parallel non-local gates, which are crucial for fault-tolerant quantum computing.

We thank the referee for their positive evaluation of our manuscript.

Before I can fully recommend publication of this work, I have some questions and suggestions for the authors:

1. The authors claim that “the logical Bell pair improves with larger code distance,” as shown in Figure 2d. An alternative explanation might be that the correlated decoder (rather than the underlying surface code encoding) performs better for larger distances. It is important to rule out such an alternative explanation to assert that the measured logical Bell pair error truly reflects the logical errors associated with the underlying physical system.

We thank the referee for this insightful comment. In order to obtain the best performance of any given logical circuit, the decoder must be optimized for each specific circuit (and code) used, and the combined performance of both the decoder and the physical encoding will set the overall performance (and typically, if the decoder is performing suboptimally, this will preferentially worsen performance for larger code distances, as seen in Ref. 6). In our surface code experiments, the decoder has been properly optimized for each code distance, ensuring that the observed improvement with code distance does not simply originate e.g. from suboptimal decoder performance for $d = 3$. Specifically we optimize the decoder for each code distance by re-scaling the edgeweights of the edges / hyperedges that connect the two logical qubits (which we find is the most relevant parameter in optimizing decoding performance). We optimize this for each code distance (finding similar values in all cases), thus ensuring the decoder is performing correctly for all code distances. We note that the decoding hypergraph for each code distance is generated using an approximate error model for the system and thus the starting point for the decoding graph is already near-optimal, and we then solve the decoding problem (finding the most likely error) exactly using Gurobi as described in the **Correlated decoding** section. Similar analysis and decoder optimization is done in Ref. 6 (Google surface code scaling paper).

Revision: To emphasize this point that the decoder is properly optimized for all code distances, we have added an additional methods figure ED Fig. 5b where we show different choices of values for this inter-logical edge-rescaling. Notably, as we approach the optimal values, all three code distances improve, and the larger code distances improve faster due to their more sensitive error scaling. This data is consistent with our decoder performing correctly for all code distances, and thus that our improvement with code distance is not originating from improper decoder performance at low code distances.

2. It would be helpful if the authors could clarify why the logical GHZ state fidelity is only 72%, as depicted in Figure 3.

a. First, providing an explicit definition of logical fidelity (in the Methods or supplementary materials) would help avoid ambiguity.

Revision: We have added the following description to the Methods, in the section “Direct fidelity estimation and tomography”.

“The target state ψ is the simultaneous eigenstate of the N stabilizer generators $\{S_i\}$, and so the projector onto the target state is $|\psi\rangle\langle\psi| = \prod_i^N (S_i + 1)/2$ (which is 1 if $S_i = 1 \forall i$, and 0 otherwise). One can therefore directly measure fidelity by measuring the expectation values of all terms in this product, which corresponds the expectation

values of all elements of the stabilizer group given by the exponentially many products of all the S_i . The logical GHZ fidelity is defined as the average expectation value of all measured elements of the stabilizer group. With our 4-qubit GHZ state, with 4 stabilizer generators $\{XXXX, ZZII, IZZI, IIZZ\}$, the 16-element stabilizer group is given by all possible products: $\{IIII, ZZII, IZZI, IIZZ, ZIIZ, IZIZ, ZIZI, ZZZZ, XXXX, XYYX, YXXY, XYYI, YYXX, YXYX, XYXY, YYYYY\}$. We measure the expectation values of all 16 of these operators; for each element, we rotate each logical qubit into the appropriate logical basis and then calculate the average parity of the four logical qubits in this measurement configuration. We then directly average all 16 elements equally (with appropriate signs, as some of the stabilizer products should have -1 values), and in this way compute the logical GHZ state fidelity. This corresponds to an exact measurement of the logical state fidelity.”

b. Considering that the entire procedure to prepare the logical GHZ state (including ancilla-assisted FT initialization, transversal CNOT, and Pauli measurement) is fault-tolerant, we would expect the logical error to be suppressed to the 2nd order of the physical error rate, $O(\epsilon^2)$, for $[[7,1,3]]$ encoding.

This is correct, we do expect the logical error to scale as ϵ^2 . However, the circuit is sufficiently long and sufficiently many errors have propagated through the transversal CNOTs, such that ϵ is no longer very small, as ϵ corresponds to the physical error accumulated over the entire circuit, since we do not perform repeated correction during the circuit. We have added text to the Methods elaborating on this (see next question).

c. However, the fidelity of 72% for the logical GHZ state seems to be limited by the first-order physical error. What prevents the logical GHZ state from achieving an even higher fidelity? It would be helpful if the authors could provide an estimated error budget to explain the limited fidelity of the logical GHZ state.

Our measurements are consistent with estimated error budgeting (similar to that done for the surface code in ED Fig. 4). Similar to the surface code modeling, using empirically estimated error rates (consistent with 99.4% two-qubit gate fidelity as well as roughly 4% data qubit decoherence error over the entire circuit) we simulate the experimental circuit (including the FT state preparation with the ancilla logical flag) and measurements of all 16 elements of the stabilizer group, and estimate a simulated logical GHZ fidelity of 79%. This is slightly higher than our measured 72% logical GHZ fidelity, possibly orig-

inating from imperfect experimental calibration. But this modeling indicates that our logical GHZ fidelity is limited by residual physical errors, which will be reduced quadratically with improvement in physical error rate, in particular by reducing residual single-qubit errors which were larger during this measurement and are dominating the error budget here. Note that performing error detection improves the logical GHZ fidelity to $99.85^{+0.1}_{-1.0}\%$ because of the higher noise tolerance and effective distance associated with performing error detection.

Revision: To address comments of b and c, we have added the following to the Methods section *Fault-tolerance and transversal gates*:

“Thus, all of our experiments with the $d = 3$ color codes are fault-tolerant from beginning to end, and so the entire algorithm is fault-tolerant and theoretically has a failure probability that scales as p^2 . However, we note that having a fault-tolerant algorithm also does not imply that errors do not build up during execution of the circuit. For this reason deep circuits require repetitive error correction to constantly remove errors and continuously benefit from the, e.g., p^2 suppression.

“Our logical GHZ state theoretically has a failure probability scaling as p^2 . Nevertheless, the error build-up (increasing p) during the operations of the circuit and the spreading of errors through transversal gates, limits our logical GHZ fidelity to 72%. This is consistent with numerical modeling. Similar to the surface code modeling (ED Fig. 4) we use empirical error rates consistent with 99.4% two-qubit gate fidelity as well as roughly 4% data qubit decoherence error (including SPAM) over the entire circuit. We simulate the experimental circuit (including the FT state preparation with the ancilla logical flag) and measurements of all 16 elements of the stabilizer group (see direct fidelity estimation section), and extract a simulated logical GHZ fidelity of 79%. This is slightly higher than our measured 72% logical GHZ fidelity, possibly originating from imperfect experimental calibration. This modeling indicates that our logical GHZ fidelity is limited by residual physical errors, which will be reduced quadratically as p^2 with reduction in physical error rate p , in particular by reducing residual single-qubit errors which were larger during this measurement and are dominating the error budget here.”

3. It would be beneficial if the authors could provide more information on the protocol for the slide-scale error detection, as this is an important technique used to evaluate the performance of the atom array quantum processor.

Revision: We have added the following section to the Methods, titled:

Sliding-scale error detection.

We again thank the referee for their useful questions and suggestions.

Reviewer: 2

The authors present results from several quantum error correction experiments using an array of neutral atoms. Among these experiments the authors claim that they (1) - demonstrate threshold-like behaviour for a pair of surface codes undergoing a transversal entangling gate (2) - prepare an array of color code qubits in a GHZ state. The authors extend this experiment to produce 40 logical color code qubits. (3) - I have differing opinions for all of these experiments. (4) - The authors measure the entanglement entropy of a scrambled system using Bell measurements. Among these experiments there are some very impressive results. However, given my concerns with (1) I am not willing to recommend acceptance. I give my comments to these experiments below.

We thank the referee for their careful evaluation of our work, many detailed questions and insights and engaging discussion. In what follows, we provide additional explanations and data regarding point (1), addressing all aspects of the referee's comments. We believe these added data and explanations further strengthen the conclusions of our work.

(1) - I have concerns about the claims of experiment 1. I can give much simpler explanation for the limited number of results that are presented in the manuscript without preparing any surface code states.

The fact that we prepare surface code states and not d^2 independent Bell pairs is demonstrated in ED Fig. 4e which reports all measured stabilizers, in both bases, of the two independent codes after the CNOT. Independent Bell pairs would give stabilizer expectation values of 0 for the stabilizers of the two independent codes. The plotted stabilizers for both independent codes measured in both bases show both that surface code states were prepared and that they were preserved during the transversal CNOT. To help further clarify, we have added an additional figure to the Methods (new ED Fig. 5) showing that we indeed prepare surface code states by now also plotting all surface code stabilizers immediately following state preparation (i.e. not including the transversal CNOT).

We also note that our data shows that even without any correlated decoding, the conventional decoding procedure (involving independent minimum-weight-perfect-matching within the two codes) produces logical Bell states with fidelities $>50\%$ (Fig. 2d top panel). The stabilizers (both in the existing ED Fig. 4e and the new ED Fig. 5) show we prepare surface code states, and the $> 50\%$ logical Bell state fidelities (with both conventional decoding and with correlated decoding) show we entangle these logical states, thereby showing that an entangling gate was performed between the two surface codes.

Although these data are already present in ED Fig. 4 (and now also ED Fig. 5), we realize our paper is very dense. To ensure readers do not miss these key pieces of data and discussion, we have added the following references to the

Figure 1: XX and ZZ parity of each physical qubit pair of the $d = 7$ surface codes. Bell state parities are consistent with zero (dominated by statistical fluctuations). Recall that maximum parity is 1, and the low XX and ZZ parities here clearly indicate that the atoms in the two surface codes are not just d^2 independent Bell pairs. This is the same data that is used for the $d = 7$ data in Fig. 2d, which also allows one to calculate the physical parities of each physical Bell pair. Clearly our data is not described by d^2 independent Bell pairs. This is also implied by the high values of the stabilizers of the two independent codes, and monogamy of entanglement.

surface code ED figures and Methods discussions and included a new Methods section devoted to correlated decoding in the surface code:

Revision: At the end of the Fig. 2 caption, we have added:

“See ED Figs. 4,5 for additional surface code data.”

And at the end of the **Correlated decoding** section we have added

“See Methods sections *Surface code and its implementation* and *Correlated decoding in the surface code* for additional discussion on the surface code in particular”

We hope these references will make this Methods data more readily apparent.

I have attached a note pointing to some of my concerns.

Our detailed response to the note is provided in the Appendix to this Referee response. To summarize, (a) the new ED Figure 5 with additional data requested by the referee (along other data in Fig. 2 and ED Fig. 4) shows we clearly prepare and entangle surface code states with certifiable entangle-

ment within each code, (b) our measurements shown in the Figure 1 above in the referee response are clearly inconsistent with d^2 independent Bell pairs (all Bell pair expectation values are 0), and (c) the ED figure also shows that ancilla measurements clearly contribute to the error correction procedure. We are aware of the subtleties described in the referee’s note. While some portions of this discussion (e.g. how measurement errors affect the standard decoder and how they are copied over / corrected by the correlated decoder) have already been detailed in the Methods section *Surface code and its implementation*, we appreciate the insights and nice intuitions in the Referee’s comment. For these reasons, we have created a new section titled *Correlated decoding in the surface code*, where we discuss various aspects of correlated decoding, including the intuitive explanation and contrast to the described “product decoder”. Note that this useful intuition reproduces our correlated decoder only in the case where all errors are before the transversal CNOT. Moreover, we note that we use the same correlated decoding procedure for both our surface code and color code experiments (to address the referee’s comments about experiment (2)).

More broadly, we note the referee’s comment that without additional data (which as detailed above, clearly show encoded qubits are prepared and entangled in our experiments) these results could hypothetically have been reproduced with d^2 independent Bell pairs is a generic feature of studying Clifford circuits with stabilizer codes, and is not unique to our approach here. E.g. the referee’s argument also hypothetically applies to, for example, making independent 4-qubit GHZ states and measuring the parities XXXX and ZZZZ. These operators should always be +1. In principle, rather than preparing encoded logical qubits, one can prepare d^2 such GHZ states and then perform classical error correction in both bases. As such, this is a generic property of studying Clifford circuits with stabilizer codes, and is not unique to our approach.

In the Appendix at the end of the referee response we provide section-by-section responses through the referee’s note, addressing all comments.

Revision: In response to these questions, we added:

ED Figure 5

and

Section *Correlated decoding in the surface code*

elaborating on these aspects.

(2) - I find this experiment very interesting and well carried out. The authors are much more thorough with the analysis of the results here as compared with (1).

We thank the referee for their positive evaluation of experiment (2). To ensure our thorough analysis of the surface code experiment is more readily seen, as mentioned above we have added references to the appropriate sections in the Methods, and we hope that the added data and discussion in the above even further strengthens the analysis of experiment (1).

I am curious to know how the state preparation fidelity is calculated. Ref. 2305.13581 for instance uses two different tomographic experiments to verify they prepare the state they claim, to argue for break even, for the magic state they prepare as compared with preparation using physical qubits. The authors could be more clear about how fidelity is calculated for their logical qubits.

We thank the referee for this comment. We will answer for both the logical SPAM and logical GHZ fidelity. The state preparation and measurement (SPAM) error is calculated by measuring the probability that $|0_L\rangle$ was measured. This is the typical way to measure and report logical SPAM of these distance-3 codes, as is done in Refs. 5,39,40 and also more works. Tomographic measurements can provide more information but is not required for reporting logical SPAM, as is done in these other works as well. To measure the logical GHZ state fidelity, we use direct fidelity estimation.

Revision: To make it clear how fidelity is calculated here, we have added the following to the Methods section **Direct fidelity estimation and tomography**:

“The target state ψ is the simultaneous eigenstate of the N stabilizer generators $\{S_i\}$, and so the projector onto the target state is $|\psi\rangle\langle\psi| = \prod_i^N (S_i + 1)/2$ (which is 1 if $S_i = 1 \forall i$, and 0 otherwise). One can thereby directly measure fidelity by measuring the expectation values of all terms in this product, which in other words refers to measuring the expectation values of all elements of the stabilizer group given by the exponentially many products of all the S_i . The logical GHZ fidelity is defined as the average expectation value of all measured elements of the stabilizer group. With our 4-qubit GHZ state, with 4 stabilizer generators $\{XXXX, ZZII, IZZI, IIZZ\}$, the 16-element stabilizer group is given by all possible products: $\{IIII, ZZII, IZZI, IIZZ, ZIIZ, IZIZ, ZIZI, ZZZZ, XXXX, XYYX, YXXY, XYYX, YYYX, YXXY, XYXY, YYYX\}$. We measure the expectation values of all 16 of these operators; for each element, we simply rotate each logical qubit into the appropriate logical basis and then calculate the average parity of the four logical qubits in this measurement configuration. We then

directly average all 16 elements equally (with appropriate signs, as some of the stabilizer products should have -1 values), and in this way compute the logical GHZ state fidelity. This is an exact measurement of the logical state fidelity.”

And have added the following to the end of the Methods section *2D color codes*:

“For fault-tolerant preparation of the $d = 3$ color code we use a modified version of the scheme summarized in Ref. 37, where instead of the 8-gate encoding circuit, we use a 9-gate encoding circuit that is more conveniently mapped to specific atom movements in our system (corresponding to graph state preparation similar to Ref. 7), followed by a transversal CNOT with an ancilla logical flag. The logical SPAM fidelity is then calculated as the probability of observing $|0_L\rangle$ after decoding.”

The demonstration of feedforward here is perhaps not the most impressive, as the feedforward could be implemented in software without post selection. After all, one can correct the ‘correct’ Bell pair with a simple Z rotation on one of the two qubits. The XX and YY expectations therefore do not need to vanish without the use of feedforward or post selection, with a straight forward use of classical updates in software. Nevertheless, I am impressed by the four-qubit encoded GHZ state.

We agree that feedforward could have been implemented in software, and this was in fact pointed out at the end of the Methods section *2D color codes* where we already state “Finally, we note that the feedforward Bell state in Fig. 4e could also be performed with a software Z_L rotation on one of the two qubits, but here we do the feedforward S on both qubits to test our feedforward capabilities; this technique is directly compatible with performing magic state teleportation.”.

Although magic state teleportation indeed would be a more interesting application, our goal here is to demonstrate all of the requisite tools to accomplish such non-trivial operations. The realization of the magic state teleportation is indeed an exciting future direction with many interesting features and subtleties. It is simply out of the scope of the present already very long manuscript.

Revision: We modified the corresponding section in the Methods to read:

“Finally, we note that the feedforward Bell state in Fig. 4e could also be performed with a software Z_L rotation on either one of the two qubits allowing one to correct to the appropriate Bell state, but here we do the feedforward S on both qubits to test our feedforward capabilities; this technique is directly compatible with performing magic

state teleportation.”

The authors make use of the correlated decoder in this experiment. Given my criticisms of this decoder in experiment (1), it would be good for the authors to comment on the use of the correlated decoder here. It is my impression these two experiments are using a correlated decoder in a different way, but I am interested in the authors’ comment on their contrast here.

See above discussion and Appendix. Indeed both experiments are using the same type of correlated decoder and the same infrastructure for calculating the decoding hypergraph. The edges and edgeweights are different, however, as they are calculated for the circuit and error model of each given experiment. For example, in the surface code experiment, the largest hyperedges affect up to 4 stabilizers at a time (originating from a physical Pauli error on one logical qubit affecting two stabilizers of both logical qubits), whereas in the GHZ color code experiment, the largest hyperedges affect up to 12 stabilizers at a time, since a single physical qubit error (which can affect up to 3 stabilizers in a single color code) can propagate onto all 4 color codes and affect all 12 stabilizers. For a given logical circuit, our correlated decoding procedure generates a decoding hypergraph which we then solve using most likely error methods, which is done here for both surface code and color code experiments, and can generically be applied to any stabilizer codes and Clifford circuits.

Experiments (3) and (4) run transversal non-Clifford circuits between distance 2 codes. These are interesting applications, and it is very exciting the authors can produce such a large number of logical qubits, and run circuits between them. The authors demonstrate improvements in their experimental results by demanding a smaller fraction of errors are detected. I think this is a very nice early demonstration of error correction used in ‘small’ logical circuits.

We thank the referee for their positive evaluation of these experiments.

One question I have, there is a comment on the decisions about post selection following error detection earlier in the paper around experiment (2). As the authors mention in a throw away comment that some errors are more likely to have introduced a logical error than others. Could the authors elaborate on this point? I am curious to know how this calculations is made.

We thank the referee for this question. We have added the new section to the Methods titled “Sliding-scale error detection”.

Revision: See new section on Sliding-scale error detection.

Minor comment - in contrast to the authors' comment early on in this section of the paper, one can perform non-Clifford gates with the surface code, although I agree it is perhaps not straight forward in comparison to a transversal gate. See Sci. Adv. 6, eaay4929 (2020). In any case, magic states are not strictly required for a 2D architecture, as the authors write.

We agree with the Referee that magic states are not strictly required for a 2D architecture. Our intent was to simply say that one cannot perform non-Cliffords easily.

Revision: To make it even more explicit that we do not mean that magic states are the only way to produce non-Cliffords on surface codes (and to address the editor's request to shorten our manuscript), we have added the above reference and modified the text to read:

“For instance, when using 2D codes such as the surface code, non-Clifford operations cannot be easily performed, and relatively expensive techniques ~~such as magic state distillation~~ are required for non-trivial computation [Ref. 36 and Sci. Adv. 6, eaay4929 (2020)] as Clifford circuits can be easily simulated.”

We once again thank the referee for their detailed comments, and provide additional discussion in the Appendix of this response.

Reviewer: 3

The manuscript “Logical Quantum Processor Based on Reconfigurable Atom Arrays” by D. Bluvstein et al. presents the experimental realization of a quantum processor that uses logical error-corrected qubits. The physical qubits that comprise the system are neutral atoms trapped in optical tweezers. This platform has significantly accelerated in the past few years and is now considered a leading contender for quantum computational platforms, as evidenced by the results in this manuscript. Enabling this technology are reconfigurable arrays of neutral atoms combined with local single-qubit gate operations and Rydberg-mediated entangling gates. The quantum processor is split into three zones for qubit storage, entangling gate operations, and readout (final and mid-circuit). The robust experimental control platform enables any-to-any connectivity between physical qubits and parallel gates on physical qubits, leading to fault-tolerant logical qubit operation. The authors test the capabilities of the experimental system by performing a CNOT gate between two logical qubits encoded in surface codes of varying code distance ($d=3,5,7$). The operational performance of the surface code corrected qubits is compared for a conventional and correlated error-decoded system made of the same number of physical qubits, indicating the superior performance of the surface codes using correlated errors. The fault-tolerant algorithmic performance of logical qubits encoded using a $d=3$ color code (like surface codes) is probed using state preparation infidelities and measuring the infidelity of a logical GHZ state.

The authors correctly point out that more complex algorithms cannot be implemented with the 2D codes demonstrated in the first part of the manuscript and consequently expand to a 3D code. The 3D code in the manuscript enacts a fast-scrambling circuit where the theoretical bitstring outcome is compared to the experimental outcome. The authors show that the outcome converges to the theoretical value as they progressively increase the error detection.

Finally, the authors investigate the logical qubit system’s ability to simulate quantum states by encoding two copies of a quantum state, performing a fast-scrambling circuit, and conducting a Bell basis measurement on the outcome. The extracted fidelity is higher than that extracted from the XEB measurements of the previous section, indicating at the least that the XEB metric is a reasonable measurement of the state purity. The entanglement entropy is extracted vs. system size, showing the expected improvement with increased error detection.

In complex quantum simulations, non-Clifford operations are required to construct the relevant quantum state. The authors measure the “magic resource” of the system or the additive Bell magic, quantifying the amount of non-Clifford operations needed to generate a quantum state. Overall, the manuscript represents a significant step forward in encoding logical qubits using various depth and complexity error-correcting codes. The logical qubits are used to generate entangling operations and perform quantum circuits, which puts the platform along the path toward more complex quantum computation and quantum simulation with logical qubit states.

The impressive experimental achievement in the manuscript warrants publica-

tion in Nature, provided the authors can address the questions and concerns outlined below.

We thank the referee for their positive evaluation of our manuscript.

1 - The statement, "... , free from gate errors and ... " in the first paragraph of the first section (Logical processor based on atom arrays) is unclear, and the authors should qualify what they mean by this statement. Single qubit gates performed in these zones on the physical qubits would indeed register errors, and the stored qubits would be subject to idle bitflip errors from scattering of tweezer light. I assume they are referring to logical qubit errors, but it is unclear. Generally, the authors do a good job throughout the text to distinguish between logical and physical qubits and errors; however, in a few cases, it is more ambiguous.

We agree this point was unclear in our original presentation.

Revision: To clarify that we mean that atoms in the storage zone are free from entangling gate errors.

"The storage zone is used for dense qubit storage, free from entangling gate errors and featuring long coherence times."

In aim of addressing the final sentence of the referee's comment above, we have studied our manuscript to see if there are other places we could more clearly distinguish between logical and physical qubits.

Revision: We have added the word "logical" in the following places:

"Subsequently four computation logicals are used to prepare the GHZ state, and logical Clifford rotations are used at the end of the circuit for direct fidelity estimation and full logical state tomography."

"As an example, Fig. 6d shows the measured absolute expectation values of all 4^{12} logical Pauli strings with sliding-scale error detection."

2 - The authors quote their coherence in the paper as " $T_2 > 1s$." this number requires more context within the main text as it results from the coherence extracted after an XY16 128 pi pulse sequence. The T_2^* for this system is significantly less at 4ms, as quoted in Reference 7. While the XY16 sequence could be continuously run to maintain coherence in the storage zone, as the

atoms move between additional zones, maintaining the sequence to preserve coherence for each atom throughout these zones is challenging.

We commend the referee on their careful observation in comparison to Ref. 7. Indeed, Ref 7 was using many pulses. In the present work however, we perform less dynamical decoupling than the 128 pulses, but our atomic temperature is lower and also we use further-detuned tweezers (852 nm instead of 830 nm), which improves coherence without using as substantial of dynamical decoupling. Consequently, for example, the T_2^* in our measured present conditions is roughly 8 ms, and our dynamically decoupled coherence times are > 1 s with fewer decoupling pulses than in Ref 7. We clarify however that in the present work the dynamical decoupling is performed using a global Raman beam that illuminates all three zones, and we indeed use this global laser for dynamical decoupling to preserve coherence as the qubits move between the various zones. In practice, with our typical dynamical decoupling cadence here, we regularly benchmark coherence times on the scale of 2s in the system. This is both when atoms are fixed in static traps and when they are moved between traps or between zones.

Revision: We have added additional text to the methods to clarify this, in the first paragraph of *Dynamical decoupling and local gates*:

“To benchmark and optimize coherence during our complex circuits, we perform a Ramsey fringe measurement encompassing the entire movement and single-qubit gate sequence and optimize the observed contrast. When performing properly, our total single-qubit error is consistent with SPAM, an effective coherence time of 1-2 s, and the Raman scattering error of all the Raman pulses. We note that these measured coherence time include the movement within and between zones; although we use fewer pulses (typically 1 per movement) than the XY16-128 sequence used to benchmark 1.5s coherence in Ref. 7, the coherence times here are naturally longer due to further-detuned tweezers used (852 nm rather than 830 nm).”

3 - Fig. 1 should include scale bars. These dimensions are provided in ED Fig 1; however, it would be easy to have them in the main text unless they are excluded for a particular reason. If Fig. 1a is the entire field of view of the objective. If so, the authors should comment on tweezer deformation on the perimeter. Additionally, denoting the Rydberg blockade radius in the text or on the figure can give some additional context to the reader.

We thank the referee for these useful comments. We intend Fig. 1 to just be a generic cartoon and not a detailed technical diagram.

Revision: We have updated the text in Fig. 1 caption to read:

“Diagram Schematic of the logical processor, segmented into three zones: storage, entangling, and readout (see ED Fig. 1 for detailed layout). ”

The region used in this work is still within the field of view of the objective and consequently we do not observe significant tweezer deformation on the edges of the range. We also thank the referee for commenting on the inclusion of the blockade radius.

Revision: We have added the following to the ED Fig 1 caption:

“The entire objective field-of-view is 400- μm diameter, and consequently we do not expect or observe substantial tweezer deformation near the edges of our processor. During two-qubit Rydberg gates, we place atoms $\lesssim 2 \mu\text{m}$ apart within a gate site, and gate sites are separated such that atoms in different gate sites are no closer than 10 μm during the gate. At our present $n = 53$ and two-photon Rabi frequency of 4.6 MHz, the blockade radius is roughly 4.3 μm , such that adjacent atoms are well-within blockade and distant atoms are well-outside blockade.”

4 - As the Fault tolerance of these operations hinges on the parallel realization of single and two-qubit gate operations, have the authors performed similar measurements to Fig1b in the “entangling zone” for single atoms (i.e., the non-dimer case) to ensure that there is no crosstalk between “gate sites?”

We indeed performed extensive characterization of two-qubit gate cross-talk within the entangling zone in ED Fig. 7 of our recent high-fidelity gates paper, Evered et al Nature 2023 (reference 8).

Revision: We have added the following discussion to the end of the third paragraph of the “Programming circuits” section of the methods.

“We find minimal two-qubit cross-talk between gate sites, as probed with long benchmarking sequences in Ref. 8. Although Ref. 8 appears to find some small cross-talk seemingly originating from decay into Rydberg P states, this should be considerably suppressed in the practical operation here due to the $\sim 200 \mu\text{s}$ duration between gates, during which time Rydberg atoms should either fly away or decay back to the ground state. ”

5 - In the second section (improving entangling gate with code distance), the authors indicate the presence of correlated errors, and this discussion is expanded upon in the ED. However, from the ED, it would seem as though the correlated errors are introduced to the system intentionally. In contrast, in the main text, it is less clear if they result as an intentional experimental choice.

We thank the referee for this comment. We clarify that the correlations between the two surface codes are indeed “correlated errors”, but these are not correlated errors in the sense of e.g. cross-talk or “cosmic-ray-type” events. Instead, these correlations are due to the deterministic propagation of Pauli errors between physical qubits of different blocks during the transversal CNOT. As such they are not intentionally introduced, but arise naturally from the transversal structure of the circuit. By accounting for this deterministic propagation, we can improve our decoding performance.

Revision: We have added the following to the caption of Fig. 2 to clarify this point:

“b, The concept of correlated decoding. Physical errors propagate between physical qubit pairs during transversal CNOT gates, creating correlations that can be utilized for improved decoding. We account for these correlations, arising from deterministic error propagation (as opposed to correlated error events), by adding edges and hyperedges that connect the decoding graphs of the two logical qubits.”

6 - Fault-tolerant is used often throughout the manuscript, and for the average Nature reader not directly in the field of quantum computing, this may need to be clarified. A sentence or two geared towards these readers would help contextualize the results’ importance.

Revision: In the last paragraph of the “Logical processor based on atom arrays” section, we have added the following:

“For example, to realize a logical single-qubit gate, we use the Raman 2D AOD (Fig. 1b) to create a grid of light beams and simultaneously illuminate the physical qubits of the logical block with the same instruction. Such a gate is transversal, meaning that operations act on physical qubits of the code block independently. This transversal property further implies the gate is inherently fault-tolerant, meaning that errors cannot spread within the code block (see Methods), thereby preventing a physical error from spreading into a logical fault. Crucially, a similar approach can realize logical entangling gates. Specifically, we use the grids generated by our moving 2D AOD

to pick up two logical qubits, interlace them in the entangling zone, and then pulse our single global Rydberg excitation laser to realize a physical entangling gate on each twin pair of the blocks (Figs. 1a,2a). This process realizes a high-fidelity, fault-tolerant transversal CNOT in a single parallel step. ”

7 - In the section “Complex logical circuits using 3D codes,” the authors claim that their XEB score is almost an order of magnitude greater than previous work in superconducting qubits. This requires more context since the superconducting circuit case is not based on logical qubits, as is the case for this work. This does not diminish the results, but this is not an apples-to-comparison.

Indeed, our score is higher due to the use of a logical encoding, which is the main advantage of our approach. The downside is the cost of increased measurement time due to the use of error detection here, but we feel that is stated clearly.

Revision: To provide clearer context, we have added the following:

“The finite XEB indicates successful sampling, and the improvement with error detection shows the benefit of using logical qubits. While this improvement comes at the cost of measurement time due to error detection, improving the sample quality cannot be replaced by simply generating more samples. Thus, improving the XEB score yields significant practical gains. We obtain an XEB of ≈ 0.1 for 48 logical qubits and hundreds of nonlocal logical entangling gates, up to roughly an order of magnitude higher than previous physical qubit implementations of similar complexity, showing the benefits of a logical encoding for this application.”

8 - The Authors should consider adding more context and explanation surrounding the additive Bell magic measurements. This is relatively new in the literature, and readers would benefit from more context.

Revision: We have modified the text as follows:

“Using this property and an analysis technique known as Bell difference sampling we experimentally evaluate and directly verify the amount of additive Bell magic ~~associated with non-Clifford operations (see Methods for definition)~~ in our circuits as a function of number of applied logical CCZs (Fig. 6c). This measurement of magic, associated with non-Clifford operations, quantifies the number of T gates”

(assuming decomposition into T) required to realize the quantum state by observing the probability that sampled Pauli strings commute with each other (see Methods)."

9 - In the ED portion of the manuscript, under the section "Programmable single-qubit gates," The randomized benchmarking sequence discussion in the last paragraph is confusing. At first read, it appears that the randomized benchmarking is performed using only 50 $\pi/2$ rotations. A study of the ED Figure 2 shows that this is not the case, and the authors should clarify this in the text.

We agree.

Revision: We have modified the methods text to read:

"To quantify the fidelity, we perform randomized benchmarking using 0, 10, 20, 30, 40, and 50 local $Z(\pi/2)$ rotations (per site) on 16 sites, ..."

10 - In the ED portion of the manuscript, under the section "Direct fidelity estimation and tomography," I think it would be helpful to include a sample calculation for one of these elements that details the relevant assumptions and measurements that come into the calculation.

Revision: We have added the following text to the methods section:

"We measure the expectation values of all 16 of these operators; for each element, we simply rotate each logical qubit into the appropriate logical basis and then calculate the average parity of the four logical qubits in this measurement configuration. We then directly average all 16 elements equally (with appropriate signs, as some of the stabilizer products should have -1 values), and in this way compute the logical GHZ state fidelity. This is an exact measurement of the logical state fidelity."

These results are a significant advancement for the field, and I commend the authors on this impressive work.

We thank the referee again for their positive evaluation of our manuscript and useful comments and suggestions.

Appendix: response to the note of Referee 2

Here we provide detailed response to the comments summarized in the additional note by the Referee 2.

Abstract:

The authors claim to produce two copies of the surface code and entangle them with a transversal CNOT gate. They then collapse the system with single-qubit measurements to make a fault-tolerant readout operation. Let me put forward an alternative explanation for the below-threshold behaviour shown in Fig. 2(d). I will argue that one could reproduce these results using d^2 independent Bell pairs, and that in this sense the results presented by the authors do not show persuasively that they have produced a surface code or any kind of long-range order at all. As such, I would suggest the authors should reconsider their methodology on this particular experiment in order to justify their claims.

As mentioned by the main body of our Response, the new ED Fig. 5 shows the stabilizers for the two independent surface codes before the transversal CNOT, showing that surface codes were successfully prepared. To provide further evidence, in Figure 1 of the Response we present the Bell state expectation values of all d^2 pairs measured in the XX and ZZ bases. Clearly there is a vanishing expectation value and our data is not described by d^2 independent Bell pairs. These data unambiguously show that we do not prepare d^2 Bell pairs, and that we do prepare two surface codes. We note again that the surface code stabilizers after the transversal CNOT are presented in ED Fig. 4.

To reach my point, I will simplify the notion of the ‘correlated decoder’. I believe, with a careful look at the system preparation, the notion of a correlated decoder hides the fact that the system they have produced might be a much more simple correlated system, rather than two distinct surface codes with an entangled logical code space, as is the goal of this demonstration.

See above. These experimental data clearly show that we do indeed prepare two surface codes. That the surface code logical qubits are entangled is also unambiguous as even with conventional decoding, the logical Bell state fidelities are $> 50\%$. In Section IV we will explain that our correlated decoder is not simply just a product of the stabilizer pairs (e.g., the “product decoder” is more sensitive to post-CNOT errors).

My argument is consistent with the results the authors have presented using an uncorrelated decoder. Additionally, if my argument below is reasonably, error

correction will only work successfully with specific choices of an initial state and readout. This is inconsistent with the function of a transversal gate, that should act correctly between arbitrary choices of input state and readout.

Our procedure to realize the transversal CNOT and decoding works independent of the input logical states and readout bases. In fact, with the $\{|+_L\rangle, |0_L\rangle\}$ input states and $\{X_L, Z_L\}$ measurement bases directly accessible to the surface code, the entangling case we study here is the most stringent. It is true that our nFT state preparation (for $d = 5, 7$, whereas $d = 3$ is FT), however, would cause additional error if the logical circuit was non-Clifford. We clarify that this is a limitation of the nFT state preparation and not of the transversal CNOT or correction procedure. Despite of the fact that our state preparation is nFT, the correlated decoding enables seeing the error suppression of the transversal CNOT because of the tolerance to measurement errors, which is already clearly detailed in the section *Surface code and its implementation*. We provide additional discussion in section V.

Concluding from my observations that are summarised in this note, I would say that the authors would need a stronger signature that they have really demonstrated an entangling gate between two surface codes. Furthermore, it would be good to see evidence that they have even produced surface codes on either code block, beyond what is presented in the current manuscript.

As stated above, we present additional data (supplementing those already presented in ED Fig. 4e) clearly showing that we prepare two surface codes, and the logical states are clearly entangled as measured by the high values of the logical Bell state stabilizers both with and without correlated decoding.

Section II:

Addressing the first paragraph, as described above, both our existing and new data show that we prepared surface codes, and that we do not prepare independent Bell pairs.

We agree with the spirit of the reviewer’s back-of-the-envelope threshold estimate. It is accurate that our experiment benefits from the higher thresholds associated with perfect syndrome extraction, i.e. the final projective measurement. This can be understood either as having one “joint” surface code with a high perfect syndrome extraction threshold, or two “disjoint” surface codes that both have high perfect syndrome extraction threshold and are decoded either together or separately. The benefit of perfect syndrome extraction is utilized in almost all existing error correction experiments including e.g. Refs. 5, 39, 40.

The fact that the threshold will worsen for deeper circuits that include more transversal CNOTs and more rounds of stabilizer measurement is clearly stated in the main text of our paper.

However, we note that decoding these two as a single surface code, or the “product decoding” discussed later, is not the optimal strategy if there are errors both before and after the transversal CNOT. Our correlated decoding accounts for the error model of the system and attempts to perform the optimal decoding. We note that this is very similar in spirit to the decoding hypergraphs constructed for single surface codes with repeated correction in Ref. 6 (Google scaling paper). Note that while we do not seek to over-emphasize the novelty of our decoding procedure, it is clearly a very useful tool, especially for transversal gate operations.

Revision: To address any concerns in the spirit of Referee remark, to not overstate the novelty of the decoder (and also to shorten our manuscript), we have made the following changes to the main text:

~~“In contrast, we note that when conventional decoding, i.e. independent minimum-weight perfect matching within both codes, is used, the fidelity decreases with code distance. This is in part due to the nFT state preparation, whose effect is partially mitigated by the correlated decoding (Methods). These measurements establish the advantage of the transport-based transversal CNOT, demonstrate a key QEC property, and highlight the utility of this correlated decoding technique in logical circuits.”~~

As well as the newly added Methods section *Correlated decoding in the surface code.*

Section III:

We show extensively in the above that our data is clearly not d^2 independent Bell pairs and that we clearly produce a pair of surface codes. This is shown both with new data and with the already existing data in ED Fig. 4e.

Section IV:

We agree with most of the discussion about the measurement errors, their effect, and how they are corrected by a correlated decoder. We point the referee to the Methods section *Surface code and its implementation* which already provides this discussion about measurement errors, how they affect the standard decoder, how they are copied over during the transversal CNOT, and how this is fixed

by the correlated decoder and enables us to observe improvement with code distance despite the nFT preparation. As mentioned above, we have provided additional references to this section to ensure it is clearly seen by readers.

However, we clarify that our decoder does not simply disregard the ancilla measurement results, and thus can not be viewed as the “product decoder”. This is borne out by the new ED Fig. 5 where we show that the ancilla measurement results improve performance. While for the conventional decoder the ancilla measurements make the most substantial difference, even for our correlated decoder the ancilla measurements improve the result, and the use of the ancilla measurements makes the decoder much less sensitive to decoding parameters. We note that in fact the $d = 3$ surface code appears to be the least impacted by the ancilla result (for the correlated decoding) even though its state preparation is fault-tolerant (as will be explained in the next section). This indicates that while it is correct that our correlated decoding procedure is more robust (but not completely insensitive) to ancilla measurement error, this is not a flaw but rather a feature of our approach. (in other words, even if we were to fault-tolerantly initialize the $d = 5, 7$ codes by repeating the stabilizer measurements, this robustness to measurement error would only improve our results.)

There are important implications of being able to keep track of these correlations and how measurement errors propagate. In particular, in a conventional approach, one requires performing d rounds of stabilizer measurement between gates. However, the ability to build these decoding hypergraphs now allows one to perform only ~ 1 round of stabilizer measurement between transversal CNOTs. Although each ~ 1 round is not sufficient to be robust to measurement errors, by keeping track of the various correlations, one can still perform additional transversal entangling gates with interleaved rounds of stabilizer measurements, and the subsequent rounds can still be used to verify earlier ancilla measurement results. Without keeping track of such correlations and propagation of measurement errors, each time a transversal gate is executed, one must have already robustly determined the value of each stabilizer and thus limits the cadence at which transversal gates can be executed. These features will be discussed in our upcoming theoretical manuscript that is currently in preparation (Ref 80) and is closely related to the concept of spacetime decoding (Refs 72-75). These features elucidate that keeping track of these measurement errors and their propagation through transversal gates, is a feature which can actually reduce space-time costs of transversal circuits.

Finally we clarify that our decoder is not simply just the described “product decoder”, as is also elucidated by our data showing that we indeed benefit from using the ancilla measurement result.

This product decoding is indeed the optimal correlated decoding strategy in the case when all errors are before the transversal CNOT. In the case that all errors in the circuit are before the transversal CNOT, then X physical errors on the control will deterministically lead to correlated errors between the same two sites (or stabilizers) of the two logical qubits and double the density of X errors on the target logical qubit. When decoding the target logical qubit however, by multiplying by the stabilizers of the control logical qubit, this propagation

is undone. Now the target logical qubit only has to decode its original density of X errors and one recovers the original surface code threshold. The same considerations can be made for Z errors originating on the target logical qubit that propagate onto the control logical qubit.

However, if there are errors *after* the transversal CNOT, then now multiplying the stabilizers of both logical qubits *double* the errors that the logical qubits experience. In fact, if there are errors only after the transversal CNOT, then the optimal decoding strategy is to decode the two logical qubits independently. The general case where there are errors both *before* and *after* the transversal CNOT, neither corresponds to independent matching nor “product decoding”. This general situation is modeled by our decoding hypergraph with edgeweights informed by our experimental error model and is how we model and decode both the surface code and color code experiments. These considerations will be further described in our upcoming theory work [Ref 80].

Section V:

As stated above, our procedure of the transversal CNOT and decoding works independent of the input states and readout bases. Our state preparation is nFT for $d=5,7$, and that the correlated decoding suppresses (but does not completely remove) the effects of this nFT state preparation due to its robustness to measurement errors is detailed in *Surface code and its implementation*. At the same time, our $d = 3$ surface code state preparation is, in fact, fault-tolerant (originating from the fact that e.g. 2 X errors on state $|+_L\rangle$ is equivalent to 1 X error up to an X_L which does nothing to $|+_L\rangle$, see Ref. 37). Importantly, for both the FT $d = 3$ codes and the nFT $d = 5, 7$ codes, the robustness to ancilla measurement error is not unique to the initial states and readout bases we choose, but instead is unique to stabilizer states and Pauli measurement bases, which we explain here. First consider the Bell state we produce here. There are no other initial states or measurement bases we can directly prepare and measure in this way that will lead to a different conclusion or not observing below-threshold behavior. Indeed, consider preparing $|+_L\rangle$ and $|0_L\rangle$ and performing a transversal CNOT between them as done here, but then instead measuring in the $X_L Z_L$ basis. This basis does not measure either of the two logical stabilizers of the logical Bell state, and therefore the logical expectation value will be 0, which does not provide useful information for benchmarking the transversal CNOT. Consider instead preparing the two logical qubits in $|1_L\rangle$ and $|1_L\rangle$, performing the transversal CNOT, and then measuring in the Z bases. This measurement can provide some information but is actually significantly simpler than the entangling case. To measure a nonzero logical expectation value, we must measure these states in the Z basis. However, this is entirely insensitive to the ancilla measurement result - i.e. in this basis the ancillas were used only for measuring the X stabilizers of the two codes, and the X stabilizers are completely unused in the final analysis of the logical state. This is unlike the entangling case studied

in our work, where although the correlated decoding is more robust to ancilla measurement errors, they are still clearly used and contribute to the correction procedure. The same conclusions will hold for e.g. performing transversal CNOTs between the $|+_L\rangle$ and $|+_L\rangle$ states prepared by Z stabilizer measurement. Thereby it follows that, in fact, the entangling case studied here and measurements in the $X_L X_L$ and $Z_L Z_L$ bases, are the only configurations with $\{|+_L\rangle, |0_L\rangle\}$ input states and $\{X_L, Z_L\}$ measurement bases that both produce nonzero logical expectation values and whose analysis actually depend on the ancilla measurement results at all.

For these reasons, we conclude that the entangling case is thus the most stringent case we can directly probe with our surface codes. Note that this conclusion is fully consistent with a number of recent experiments demonstrating quantum operations between logical qubits. While creating Bell states is a very common way of probing entangling gates, we specifically note Refs 5,40 that perform transversal CNOTs between logical qubits and do study different input and output states. In both references the authors find that the entangling case, creating the logical Bell state, is by far the lowest fidelity and the most sensitive to imperfections. In addition to the arguments in the previous paragraph, the Bell state case is clearly the most stringent and sensitive case here and thus is well-suited to our probing of the transversal gate.

In summary, in addition to the existing data in the Methods, we have provided further evidence that (a) that we have clearly prepared two surface codes and performed an entangling transversal CNOT between them, (b) that our data is not d^2 independent Bell pairs, (c) that the ancilla measurements contribute to the error correction procedure, (d) that our decoder is more complex than just the “product decoder” (although related), and (e) that the entangling case chosen is the clearest demonstration of a transversal CNOT that we can study with $\{|+_L\rangle, |0_L\rangle\}$ input states and $\{X_L, Z_L\}$ measurement bases for the surface code prepared in this way. We thank the referee for this engaging discussion.

Revision: we have added:

ED Figure 5

and

Section *Correlated decoding in the surface code*

elaborating on these aspects.

Reviewer Reports on the First Revision:

Referees' comments:

Referee #1 (Remarks to the Author):

The authors have addressed all my questions. I would like to recommend publishing this work, which represents a significant advance in quantum error correction, in Nature.

Referee #2 (Remarks to the Author):

I thank the authors for their response to my discussion. Indeed, I must confess that I overlooked ED figure 4e during the expedited review process, but I am glad the authors have added a comment making it clearer to check this data. Furthermore I think the work benefits from the addition of ED figure 5. This has reassured me that reliable stabilizer readings do not depend on the transversal operation to the extent that I could have postulated.

I think the work also benefits from the addition of the correlated decoding for the surface code. As I mentioned, I was mainly concerned with the effect of ancilla measurement errors for the syndrome between the two code blocks. I am now reasonably convinced that the logical Bell state preparation presented here is robust to such errors.

I am also pleased to see an extended discussion on sliding scale error detection. As far as I know this is a new method that is probably deserved of additional scrutiny. However, this work would go beyond the scope of this paper.

Overall, with the changes made by the authors, I am happy to recommend this work for publication.

Referee #3 (Remarks to the Author):

I thank the authors for their thoughtful responses to my comments and those from the other referees, and I am satisfied that they have addressed all the concerns raised. Based on this, I recommend the publication of the revised manuscript.

Author Rebuttals to First Revision:

Referee responses and summary of revisions 2

Nature Manuscript number: 2023-10-18750 Bluvstein

November 28, 2023

Reviewer: 1

The authors have addressed all my questions. I would like to recommend publishing this work, which represents a significant advance in quantum error correction, in Nature.

We thank the referee again for their positive evaluation of the manuscript and their useful suggestions.

Reviewer: 2

I thank the authors for their response to my discussion. Indeed, I must confess that I overlooked ED figure 4e during the expedited review process, but I am glad the authors have added a comment making it clearer to check this data. Furthermore I think the work benefits from the addition of ED figure 5. This has reassured me that reliable stabilizer readings do not depend on the transversal operation to the extent that I could have postulated.

I think the work also benefits from the addition of the correlated decoding for the surface code. As I mentioned, I was mainly concerned with the effect of ancilla measurement errors for the syndrome between the two code blocks. I am now reasonably convinced that the logical Bell state preparation presented here is robust to such errors.

I am also pleased to see an extended discussion on sliding scale error detection. As far as I know this is a new method that is probably deserved of additional scrutiny. However, this work would go beyond the scope of this paper.

Overall, with the changes made by the authors, I am happy to recommend this work for publication.

We thank the referee for their positive evaluation of the manuscript and recommending publication. We agree that the work benefits from these additional data and discussion, and we again thank the referee for the engaging discussion and their thoughtful insights.

Reviewer: 3

I thank the authors for their thoughtful responses to my comments and those from the other referees, and I am satisfied that they have addressed all the concerns raised. Based on this, I recommend the publication of the revised manuscript.

We thank the referee again for their positive evaluation of the manuscript and their useful suggestions.